# When Should We Prefer Offline Reinforcement Learning Over Behavioral Cloning?

**Aviral Kumar**[*,1,2]**, Joey Hong**[*,1]**, Anikait Singh**[1]**, Sergey Levine**[1,2]
[1]Department of EECS, UC Berkeley      [2]Google Research          (*Equal Contribution)
{aviralk, joey_hong}@berkeley.edu

## Abstract

Offline reinforcement learning (RL) algorithms can acquire effective policies by utilizing previously collected experience, without any online interaction. It is widely understood that offline RL is able to extract good policies even from highly suboptimal data, a scenario where imitation learning finds suboptimal solutions that do not improve over the demonstrator that generated the dataset. However, another common use case for practitioners is to learn from data that resembles demonstrations. In this case, one can choose to apply offline RL, but can also use behavioral cloning (BC) algorithms, which mimic a subset of the dataset via supervised learning. Therefore, it seems natural to ask: when can an offline RL method outperform BC with an equal amount of expert data, even when BC is a natural choice? To answer this question, we characterize the properties of environments that allow offline RL methods to perform better than BC methods, even when only provided with expert data. Additionally, we show that policies trained on sufficiently noisy suboptimal data can attain better performance than even BC algorithms with expert data, especially on long-horizon problems. We validate our theoretical results via extensive experiments on both diagnostic and high-dimensional domains including robotic manipulation, maze navigation, and Atari games, with a variety of data distributions. We observe that, under specific but common conditions such as sparse rewards or noisy data sources, modern offline RL methods can significantly outperform BC.

## 1 Introduction

Offline reinforcement learning (RL) algorithms aim to leverage large existing datasets of previously collected data to produce effective policies that generalize across a wide range of scenarios, without the need for costly active data collection. Many recent offline RL algorithms [16, 26, 66, 27, 69, 61, 25] can work well even when provided with highly suboptimal data, and a number of these approaches have been studied theoretically [64, 71, 52, 20]. While it is clear that offline RL algorithms are a good choice when the available data is either random or highly suboptimal, it is less clear if such methods are useful when the dataset consists of demonstration that come from expert or near-expert demonstrations. In these cases, imitation learning algorithms, such as behavioral cloning (BC), can be used to train policies via supervised learning. It then seems natural to ask: *When should we prefer to use offline RL over imitation learning?*

To our knowledge, there has not been a rigorous characterization of when offline RL perform better than imitation learning. Existing *empirical* studies comparing offline RL to imitation learning have come to mixed conclusions. Some works show that offline RL methods appear to greatly outperform imitation learning, specifically in environments that require "stitching" parts of suboptimal trajectories [14]. In contrast, a number of recent works have argued that BC performs better than offline RL on both expert and suboptimal demonstration data over a variety of tasks [38, 12, 17]. This makes it confusing for practitioners to understand whether to use offline RL or simply run BC on collected demonstrations. Thus, in this work we aim to understand if there are conditions on the environment or the dataset under which an offline RL algorithm might outperform BC for a given task, even when BC is provided with expert data or is allowed to use rewards as side information. Our findings can inform a practitioner in determining whether offline RL is a good choice in their domain, even when expert or near-expert data is available and BC might appear to be a natural choice.

Our contribution in this paper is a theoretical and empirical characterization of certain conditions when offline RL outperforms BC. Theoretically, our work presents conditions on the MDP and dataset that are sufficient for offline RL to achieve better worst-case guarantees than even the *best-case*

*lower-bound* for BC using the same amount of *expert demonstrations*. These conditions are grounded in practical problems, and provide guidance to the practitioner as to whether they should use RL or BC. Concretely, we show that in the case of expert data, the error incurred by offline RL algorithms can scale significantly more favorably when the MDP enjoys some structure, which includes horizon-independent returns (*i.e.*, sparse rewards) or a low volume of states where it is "critical" to take the same action as the expert (Section 4.2). Meanwhile, in the case of sufficiently noisy data, we show that offline RL again enjoys better guarantees on long-horizon tasks (Section 4.3). Finally, since BC methods ignore rewards, we consider generalized BC methods that use the observed rewards to inform learning, and show that it is still preferable to perform offline RL (Section 4.4).

Empirically, we validate our theoretical conclusions on diagnostic gridworld domains [13] and large-scale benchmark problems in robotic manipulation and navigation and Atari games, using human data [14], scripted data [60], and data generated from RL policies [3]. We verify that in multiple long-horizon problems where the conditions we propose are likely to be satisfied, practical offline RL methods can outperform BC and generalized BC methods. We show that using careful offline tuning practices, we show that it is possible for offline RL to outperform cloning an expert dataset for the same task, given equal amounts of data. We also highlight open questions for hyperparameter tuning that have the potential to make offline RL methods work better in practice.

## 2 RELATED WORK

Offline RL [32, 34] has shown promise in domains such as robotic manipulation [22, 37, 60, 23], NLP [19] and healthcare [58, 63]. The major challenge in offline RL is distribution shift [16, 26], where the learned policy might execute out-of-distribution actions. Prior offline RL methods can broadly be characterized into two categories: **(1)** *policy-constraint* methods that regularize the learned policy to be "close" to the behavior policy either explicitly [16, 26, 36, 66, 15] or implicitly [59, 47, 45], or via importance sampling [35, 62, 43], and **(2)** *conservative* methods that learn a conservative, estimate of return and optimize the policy against it [27, 25, 24, 69, 70]. Our goal is not to devise a new algorithm, but to understand when existing offline RL methods can outperform BC.

*When do offline RL methods outperform BC?* Rashidinejad et al. [52] derive a conservative offline RL algorithm based on lower-confidence bounds (LCB) that provably outperforms BC in the simpler contextual bandits (CB) setting, but do not extend it to MDPs. While this CB result signals the possibility that offline RL can outperform BC in theory, this generalization is not trivial, as RL suffers from compounding errors [41, 42, 64]. Laroche et al. [33], Nadjahi et al. [44], Kumar et al. [27], Liu et al. [36], Xie et al. [68] present safe policy improvement bounds expressed as improvements over the behavior policy, which imitation aims to recover, but these bounds do not clearly indicate when offline RL is better or worse. Empirically, Fu et al. [14] show that offline RL considerably outperforms BC for tasks that require "stitching" trajectory segments to devise an optimal policy. In contrast, Mandlekar et al. [38], Brandfonbrener et al. [6], Chen et al. [8], Hahn et al. [17] suggest that BC or filtered BC using the top fraction of the data performs better on other tasks. While the performance results in D4RL [14], especially on the Adroit domains, show that offline RL outperforms BC even on expert data, Florence et al. [12] reported superior BC results, making it unclear if the discrepancy arises from different hyperparameter tuning practices. Kurenkov & Kolesnikov [31] emphasize the importance of the online evaluation budget for offline RL methods and show that BC is more favorable in a limited budget. While these prior works discussed above primarily attempt to show that BC can be better than offline RL, we attempt to highlight when offline RL is expected to be better by providing a characterization of scenarios where we would expect offline RL to be better than BC, and empirical results verifying that offline RL indeed performs better on such problems [14, 60, 4].

Our theoretical analysis combines tools from a number of prior works. We analyze the total error incurred by RL via an error propagation analysis [41, 42, 11, 7, 67, 36], which gives rise to bounds with *concentrability coefficients* that bound the total distributional shift between the learned policy and the data distribution [67, 36]. We use tools from Ren et al. [53], which provide horizon-free bounds for standard (non-conservative) offline Q-learning but relax their strict coverage assumptions. While our analysis studies a LCB-style algorithm similar to Rashidinejad et al. [52], Jin et al. [20], we modify it to use tighter Bernstein bonuses [73, 1], which is key to improving the guarantee.

## 3 PROBLEM SETUP AND PRELIMINARIES

The goal in reinforcement learning is to learn a policy $\pi(\cdot|\mathbf{s})$ that maximizes the expected cumulative discounted reward in a Markov decision process (MDP), which is defined by a tuple $(\mathcal{S}, \mathcal{A}, P, r, \gamma)$.

$\mathcal{S}, \mathcal{A}$ represent state and action spaces, $P(\mathbf{s}'|\mathbf{s}, \mathbf{a})$ and $r(\mathbf{s}, \mathbf{a})$ represent the dynamics and mean reward function, and $\gamma \in (0, 1)$ represents the discount factor. The effective horizon of the MDP is given by $H = 1/(1 - \gamma)$. The Q-function, $Q^\pi(\mathbf{s}, \mathbf{a})$ for a given policy $\pi$ is equal to the discounted long-term reward attained by executing $\mathbf{a}$ at the state $\mathbf{s}$ and then following policy $\pi$ thereafter. $Q^\pi$ satisfies the recursion: $\forall \mathbf{s}, \mathbf{a} \in \mathcal{S} \times \mathcal{A}, Q^\pi(\mathbf{s}, \mathbf{a}) = r(\mathbf{s}, \mathbf{a}) + \gamma \mathbb{E}_{\mathbf{s}' \sim P(\cdot|\mathbf{s}, \mathbf{a}), \mathbf{a}' \sim \pi(\cdot|\mathbf{s}')} [Q(\mathbf{s}', \mathbf{a}')]$. The value function $V^\pi$ considers the expectation of the Q-function over the policy $V^\pi(\mathbf{s}) = \mathbb{E}_{\mathbf{a} \sim \pi(\cdot|\mathbf{s})} [Q^\pi(\mathbf{s}, \mathbf{a})]$. Meanwhile, the Q-function of the optimal policy, $Q^*$, satisfies the recursion: $Q^*(\mathbf{s}, \mathbf{a}) = r(\mathbf{s}, \mathbf{a}) + \mathbb{E}_{\mathbf{s}' \sim P(\cdot|\mathbf{s}, \mathbf{a})} [\max_{\mathbf{a}'} Q^*(\mathbf{s}', \mathbf{a}')]$, and the optimal value function is given by $V^*(\mathbf{s}) = \max_{\mathbf{a}} Q^*(\mathbf{s}, \mathbf{a})$. Finally, the expected cumulative discounted reward is given by $J(\pi) = \mathbb{E}_{\mathbf{s}_0 \sim \rho} [V^\pi(\mathbf{s}_0)]$.

In offline RL, we are provided with a dataset $\mathcal{D}$ of transitions, $\mathcal{D} = \{(\mathbf{s}_i, \mathbf{a}_i, r_i, \mathbf{s}'_i)\}_{i=1}^N$ of size $|\mathcal{D}| = N$. We assume that the dataset $\mathcal{D}$ is generated i.i.d. from a distribution $\mu(\mathbf{s}, \mathbf{a})$ that specifies the effective behavior policy $\pi_\beta(\mathbf{a}|\mathbf{s}) := \mu(\mathbf{s}, \mathbf{a})/\sum_{\mathbf{a}} \mu(\mathbf{s}, \mathbf{a})$. Note that this holds even if the data itself is generated by running a non-Markovian policy $\pi_\beta$ [48]. Let $n(\mathbf{s}, \mathbf{a})$ be the number of times $(\mathbf{s}, \mathbf{a})$ appear in $\mathcal{D}$, and $\widehat{P}(\cdot|\mathbf{s}, \mathbf{a})$ and $\widehat{r}(\mathbf{s}, \mathbf{a})$ denote the empirical dynamics and reward distributions in $\mathcal{D}$, which may be different from $P$ and $r$ due to stochasticity. Following Rashidinejad et al. [52], the goal is to minimize the suboptimality of the learned policy $\widehat{\pi}$:

$$\mathsf{SubOpt}(\widehat{\pi}) = \mathbb{E}_{\mathcal{D} \sim \mu} [J(\pi^*) - J(\widehat{\pi})] = \mathbb{E}_{\mathcal{D}} \left[ \mathbb{E}_{\mathbf{s}_0 \sim \rho} \left[ V^*(\mathbf{s}_0) - V^{\widehat{\pi}}(\mathbf{s}_0) \right] \right]. \tag{1}$$

We will now define some conditions on the offline dataset and MDP structure that we will use in our analysis. The first characterizes the distribution shift between the data distribution $\mu(\mathbf{s}, \mathbf{a})$ and the normalized state-action marginal of $\pi^*$, given by $d^*(\mathbf{s}, \mathbf{a}) = (1 - \gamma) \sum_{t=0}^{\infty} \gamma^t \mathbb{P}(\mathbf{s}_t = s, \mathbf{a}_t = a; \pi^*)$, via a *concentrability coefficient* $C^*$.

**Condition 3.1** (Rashidinejad et al. [52], Concentrability of the data distribution). *Define $C^*$ to be the smallest, finite constant that satisfies:* $d^*(\mathbf{s}, \mathbf{a})/\mu(\mathbf{s}, \mathbf{a}) \leq C^* \ \ \forall \mathbf{s} \in \mathcal{S}, \mathbf{a} \in \mathcal{A}.$

Intuitively, the coefficient $C^*$ formalizes how well the data distribution $\mu(\mathbf{s}, \mathbf{a})$ covers the state-action pairs visited under the optimal $\pi^*$, where $C^* = 1$ corresponds to data from $\pi^*$. If $\mu(\mathbf{s}, \mathbf{a})$ primarily covers state-action pairs that are not visited by $\pi^*$, $C^*$ would be large. The next condition is that the return for any trajectory in the MDP is bounded by a constant, which w.l.o.g., we assume to be 1.

**Condition 3.2** (Ren et al. [53], the value of any trajectory is bounded by 1). *The infinite-horizon discounted return for any trajectory* $\tau = (\mathbf{s}_0, \mathbf{a}_0, r_0, \mathbf{s}_1, \cdots)$ *is bounded as* $\sum_{t=0}^{\infty} \gamma^t r_t \leq 1.$

This condition holds in sparse-reward tasks, particularly those where an agent succeeds or fails at its task once per episode. This is common in domains such as robotics [60, 22] and games [4], where the agent receives a signal upon succeeding a task or winning. This condition also appears in prior work deriving suboptimality bounds for RL algorithms [53, 73]. Let $n \wedge 1 = \max\{n, 1\}$. Denote $\iota = \mathrm{polylog}(|\mathcal{S}|, H, N)$. We let $\iota$ be a polylogarithmic quantity, changing with context. For $d$-dimensional vectors $\mathbf{x}, \mathbf{y}, \mathbf{x}(i)$ denotes its $i$-th entry, and define $\mathbb{V}(\mathbf{x}, \mathbf{y}) = \sum_i \mathbf{x}(i) \mathbf{y}(i)^2 - (\sum_i \mathbf{x}(i) \mathbf{y}(i))^2$.

## 4 THEORETICAL COMPARISON OF BC AND OFFLINE RL

In this section, we present performance guarantees for BC and offline RL, and characterize scenarios where offline RL algorithms will outperform BC. We first present general upper bounds for both algorithms in Section 4.1, by extending prior work to account for the conditions discussed in Section 3. Then, we compare the performance of BC and RL when provided with the same data generated by an expert in Section 4.2 and when RL is given noisy, suboptimal data in Section 4.3. Our goal is to characterize the conditions on the environment and offline dataset where RL can outperform BC. Furthermore, we provide intuition for when they are likely to hold in Appendix D.

### 4.1 IMPROVED PERFORMANCE GUARANTEES OF BC AND OFFLINE RL

Our goal is to understand if there exist offline RL methods that can outperform BC for a given task. As our aim is to provide a proof of existence, we analyze representative offline RL and BC algorithms that achieve optimal suboptimality guarantees. For brevity, we only consider a conservative offline RL algorithm (as defined in Section 2) in the main paper and defer analysis of a representative policy-constraint method to Appendix C. Both algorithms are described in Algorithms 1 and 2.

**Guarantees for BC.** For analysis purposes, we consider a BC algorithm that matches the empirical behavior policy on states in the offline dataset, and takes uniform random actions outside the support of the dataset. This BC algorithm was also analyzed in prior work [49], and is no worse than other

schemes for acting at out-of-support states in general. Denoting the learned BC policy as $\widehat{\pi}_\beta$, we have $\forall \mathbf{s} \in \mathcal{D}, \widehat{\pi}_\beta(\mathbf{a}|\mathbf{s}) \leftarrow n(\mathbf{s}, \mathbf{a})/n(\mathbf{s})$, and $\forall \mathbf{s} \notin \mathcal{D}, \widehat{\pi}_\beta(\mathbf{a}|\mathbf{s}) \leftarrow 1/|\mathcal{A}|$. We adapt the results presented by Rajaraman et al. [49] to the setting with Conditions 3.1 and 3.2. BC can only incur a non-zero asymptotic suboptimality (*i.e.*, does not decrease to 0 as $N \to \infty$) in scenarios where $C^* = 1$, as it aims to match the data distribution $\mu(\mathbf{s}, \mathbf{a})$, and a non-expert dataset will inhibit the cloned policy from matching the expert $\pi^*$. The performance for BC is bounded in Theorem 4.1.

**Theorem 4.1** (Performance of BC). *Under Conditions 3.1 and 3.2, the suboptimality of BC satisfies*

$$\mathsf{SubOpt}(\widehat{\pi}_\beta) \lesssim \frac{(C^* - 1)H}{2} + \frac{|\mathcal{S}|H\iota}{N}.$$

A proof of Theorem 4.1 is presented in Appendix B.1. The first term is the additional suboptimality incurred due to discrepancy between the behavior and optimal policies. The second term in this bound is derived by bounding the expected visitation frequency of the learned policy $\widehat{\pi}_\beta$ onto states not observed in the dataset. The analysis is similar to that for existing bounds for imitation learning [54, 49]. We achieve $\tilde{\mathcal{O}}(H)$ suboptimality rather than $\tilde{\mathcal{O}}(H^2)$ due to Condition 3.2, since the worst-case suboptimality of any trajectory is 1 rather than $H$.

**Guarantees for conservative offline RL.** We consider guarantees for a class of offline RL algorithms that maintain conservative value estimator such that the estimated value lower-bounds the true one, *i.e.*, $\widehat{V}^\pi \leq V^\pi$ for policy $\pi$. Existing offline RL algorithms achieve this by subtracting a penalty from the reward, either explicitly [69, 24] or implicitly [27]. We only analyze one such algorithm that does the former, but we believe the algorithm can serve as a theoretical model for general conservative offline RL methods, where similar algorithms can be analyzed using the same technique. While the algorithm we consider is similar to VI-LCB , which was proposed by Rashidinejad et al. [52] and subtracts a penalty $b(\mathbf{s}, \mathbf{a})$ from the reward during value iteration, we use a different penalty that results in a tighter bound. The estimated Q-values are obtained by iteratively solving the following Bellman backup: $\widehat{Q}(\mathbf{s}, \mathbf{a}) \leftarrow \widehat{r}(\mathbf{s}, \mathbf{a}) - b(\mathbf{s}, \mathbf{a}) + \sum_{\mathbf{s}'} \widehat{P}(\mathbf{s}'|\mathbf{s}, \mathbf{a}) \max_{\mathbf{a}'} \widehat{Q}(\mathbf{s}', \mathbf{a}')$. The learned policy is then given by $\widehat{\pi}^*(\mathbf{s}) \leftarrow \arg\max_{\mathbf{a}} \widehat{Q}(\mathbf{s}, \mathbf{a})$. The specific $b(\mathbf{s}, \mathbf{a})$ is derived using Bernstein's inequality:

$$b(\mathbf{s}, \mathbf{a}) \leftarrow \sqrt{\frac{\mathbb{V}(\widehat{P}(\cdot|\mathbf{s}, \mathbf{a}), \widehat{V})\iota}{(n(\mathbf{s}, \mathbf{a}) \wedge 1)}} + \sqrt{\frac{\widehat{r}(\mathbf{s}, \mathbf{a})\iota}{(n(\mathbf{s}, \mathbf{a}) \wedge 1)}} + \frac{\iota}{(n(\mathbf{s}, \mathbf{a}) \wedge 1)}. \tag{2}$$

The performance of the learned policy $\widehat{\pi}^*$ can then be bounded as:

**Theorem 4.2** (Performance of conservative offline RL). *Under Conditions 3.1 and 3.2, the policy $\widehat{\pi}^*$ found by conservative offline RL algorithm can satisfy*

$$\mathsf{SubOpt}(\widehat{\pi}^*) \lesssim \sqrt{\frac{C^*|\mathcal{S}|H\iota}{N}} + \frac{C^*|\mathcal{S}|H\iota}{N}.$$

We defer a proof for Theorem 4.2 to Appendix B.2. At a high level, we first show that our algorithm is always conservative, *i.e.*, $\forall \mathbf{s}, \widehat{V}(\mathbf{s}) \leq V^{\widehat{\pi}^*}(\mathbf{s})$, and then bound the total suboptimality incurred as a result of being conservative. Our bound in Theorem 4.2 improves on existing bounds several ways: **(1)** by considering pessimistic value estimates, we are able to remove the strict coverage assumptions used by Ren et al. [53], **(2)** by using variance recursion techniques on the Bernstein bonuses, with Condition 3.2, we save a $O(H^2)$ factor over VI-LCB in Rashidinejad et al. [52], and **(3)** we shave a $|\mathcal{S}|$ factor by introducing $s$-absorbing MDPs for each state as in Agarwal et al. [1]. Building on this analysis of the representative offline RL method, we will now compare BC and offline RL under specific conditions on the MDP and dataset.

## 4.2 COMPARISON UNDER EXPERT DATA

We first compare the performance bounds from Section 4.1 when the offline dataset is generated from the expert. In relation to Condition 3.1, this corresponds to small $C^*$. Specifically, we consider $C^* \in [1, 1 + \tilde{\mathcal{O}}(1/N)]$ and in this case, the suboptimality of BC in Theorem 4.1 scales as $\tilde{\mathcal{O}}(|\mathcal{S}|H/N)$. In this regime, we perform a nuanced comparison by analyzing specific scenarios where RL may outperform BC. We consider the case of $C^* = 1$ and $C^* = 1 + \tilde{\mathcal{O}}(1/N)$ separately.

**What happens when $C^* = 1$?** In this case, we derive a lower-bound of $|\mathcal{S}|H/N$ for any offline algorithm, by utilizing the analysis of Rajaraman et al. [49] and factoring in Condition 3.2.

**Theorem 4.3** (Information-theoretic lower-bound for offline learning with $C^* = 1$). *For any learner $\widehat{\pi}$, there exists an MDP $\mathcal{M}$ satisfying Assumption 3.2, and a deterministic expert $\pi^*$, such that the expected suboptimality of the learner is lower-bounded:*

$$\sup_{\mathcal{M},\pi^*} \mathsf{SubOpt}(\widehat{\pi}) \gtrsim \frac{|\mathcal{S}|H}{N}$$

The proof of Theorem 4.3 uses the same hard instance from Theorem 6.1 of Rajaraman et al. [49], except that one factor of $H$ is dropped due to Condition 3.2. The other factor of $H$ arises from the performance difference lemma and is retained. In this case, where BC achieves the lower bound up to logarithmic factors, we argue that we cannot improve over BC. This is because the suboptimality of BC is entirely due to encountering states that do not appear in the dataset; without additional assumptions on the ability to generalize to unseen states, offline RL must incur the same suboptimality, as both methods would choose actions uniformly at random.

> **Practical Observation 4.1.** *When no assumptions are made on the environment structure, both offline RL and BC perform equally poorly with trajectories from an expert demonstrator.*

However, we argue that even with expert demonstrations as data, $C^*$ may not exactly be 1. Naïvely, it seems plausible that the expert that collected the dataset did not perform optimally at every transition; this is often true for humans, or stochastic experts such as $\epsilon$-greedy or maximum-entropy policies. In addition, there are scenarios where $C^* > 1$ even though the expert behaves optimally: for example, when the environment is stochastic or in the presence of small distribution shifts in the environment. One practical example of this is when the initial state distribution changes between dataset collection and evaluation (*e.g.*, in robotics [60], or self-driving [5]). Since the normalized state-action marginals $d^*(\mathbf{s}, \mathbf{a}), \mu(\mathbf{s}, \mathbf{a})$ depend on $\rho(\mathbf{s})$, this would lead to $C^* > 1$ even when the data collection policy acts exactly as the expert $\pi^*$ at each state.

**What happens when $C^* = 1 + \tilde{\mathcal{O}}(1/N)$?** Here $C^*$ is small enough that BC still achieves the same optimal $\tilde{\mathcal{O}}(|\mathcal{S}|H/N)$ performance guarantee. However, there is suboptimality incurred by BC even for states that appear in the dataset due to distribution shift, which allows us to argue about structural properties of MDPs that allow offline RL to perform better across those states, particularly for problems with large effective horizon $H$. We discuss one such structure below.

In several practical problem domains, the return of any trajectory can mostly be explained by the actions taken at a small fraction of states, which we call *critical states*. This can occur when there exist a large proportion of states in a trajectory where it is not costly to recover after deviating from the optimal trajectory, or when identifying an optimal trajectory is easy: this could be simply because there exists a large volume of near-optimal trajectories, or because at all but a few states, the volume of "good-enough" actions is large. Therefore, we would expect that offline RL can quickly identify and master such good-enough actions at a majority of the states while reward-agnostic BC would be unable to do so.

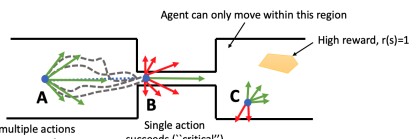

Figure 1: **Illustration showing the intuition behind critical states.** The agent is supposed to navigate to a high-reward region marked as the yellow polygon, without crashing into the walls. For different states, **A**, **B** and **C** that we consider, the agent has a high volume of actions that allow it to reach the goal at states **A** and **C**, but only few actions that allow it to do so at state **B**. States around **A** and **C** are not critical, and so this task has only a small volume of critical states (i.e., those in the thin tunnel).

Two examples are in robotic manipulation and navigation. In manipulation tasks such as grasping, if the robot is not near the object, it can take many different actions and still pick up the object by the end; this is because unless the object breaks, actions taken by the robot are typically reversible [21]. In this phase, the robot just needs to avoid performing actions that do not at all move it closer to the object since it will fail to grasp it, but it needs to know no more than a general sense of how to approach the object, which is easily identifiable from the reward information. It is only at a few "critical states" when the robot grasps the object, where the robot should be careful to not drop and break the object.

In navigation, as we pictorially illustrate in Figure 1, there may exist multiple paths that end at the same goal, particularly in large, unobstructed areas [55]. For example, while navigating through a wide tunnel, the exact direction the agent takes may not matter so much as multiple directions will take the agent through the tunnel, and identifying these good-enough actions using reward

information is easy. However, there are "critical states" like narrow doorways where taking a specific action is important. Domains that do not satisfy this structure include cliffwalk environments, where a single incorrect action at any state will cause the agent to fall off the cliff. We can formally define one notion of critical states as follows:

**Definition 4.1** (Non-critical states). *A state $\mathbf{s}$ is said to be non-critical (i.e., $\mathbf{s} \notin \mathcal{C}$) if there exists a large subset $\mathcal{G}(\mathbf{s})$ of $\varepsilon$-good actions, such that,*

$$\forall \mathbf{a} \in \mathcal{G}(\mathbf{s}), \quad \max_{\mathbf{a}'} Q^*(\mathbf{s}, \mathbf{a}') - Q^*(\mathbf{s}, \mathbf{a}) \leq \frac{\varepsilon}{H}, \quad and \quad \forall \mathbf{a} \in \mathcal{A} \setminus \mathcal{G}(\mathbf{s}), \quad \max_{\mathbf{a}'} Q^*(\mathbf{s}, \mathbf{a}') - Q^*(\mathbf{s}, \mathbf{a}) \simeq \Delta(\mathbf{s}).$$

Now the structural condition we consider is that for any policy, the total density of critical states, $\mathbf{s} \in \mathcal{C}$ is bounded by $p_c$, and we will control $p_c$ for our bound.

**Condition 4.1** (Occupancy of critical states is small). *Let $\exists p_c \in (0, 1)$ such that for any policy $\pi$ the average occupancy of critical states is bounded above by $p_c$: $\sum_{\mathbf{s} \in \mathcal{C}} d^\pi(\mathbf{s}) \leq p_c$.*

We can show that, if the MDP satisfies the above condition with a small enough $p_c$ and and the number of good actions $|\mathcal{G}(\mathbf{s})|$ are large enough, then by controlling the gap $\Delta(\mathbf{s})$ between suboptimality of good and bad actions some offline RL algorithms can outperform BC. We perform this analysis under a simplified setting where the state-marginal distribution in the dataset matches that of the optimal policy and find that a policy constraint offline RL algorithm can outperform BC. We describe the informal statement below and discuss the details and a proof in Appendix B.3.

**Corollary 4.1** (Offline RL vs BC with critical states). *Let the dataset distribution be such that $\forall \mathbf{s}, d^{\pi^*}(\mathbf{s}) = \mu(\mathbf{s})$ and all for state-action pairs $\forall (\mathbf{s}, \mathbf{a}), n(\mathbf{s}, \mathbf{a}) \geq n_0$. Then, under Conditions 4.1 with $p_c = 1/H$, 3.1 and 3.2 a policy $\widehat{\pi}^*$ found by policy constraint offline RL (Equation 10) satisfies*

$$\mathsf{SubOpt}(\widehat{\pi}^*) \lesssim \mathsf{SubOpt}(\widehat{\pi}_{BC}).$$

The above statement indicates that when we encounter $\mathcal{O}(1/H)$ critical states on average in any trajectory, then we can achieve better performance via offline RL than BC. In this simplified setting of full coverage over all states, the proof relies on showing that the particular offline RL algorithm we consider can collapse to good-enough actions at non-critical states quickly, whereas BC is unable to do so since it simply mimics the data distribution. On the other hand, BC and RL perform equally well at critical states. We can control the convergence rate of RL by adjusting $\Delta$, and $|\mathcal{G}(\mathbf{s})|$, whereas BC does not benefit from this and incurs a suboptimality that matches the information-theoretic lower bound (Theorem 4.3). While this proof only considers one offline RL algorithm, we conjecture that it should be possible to generalize such an argument to the general family of offline RL algorithms.

> **Practical Observation 4.2.** *Offline RL can be preferred over BC, even with expert or near-expert data, when either the initial state distribution changes during deployment, or when the environment has a few "critical" states.*

## 4.3 COMPARISON UNDER NOISY DATA

In practice, it is often much more tractable to obtain suboptimal demonstrations rather than expert ones. For example, suboptimal demonstrations can be obtained by running a scripted policy. From Theorem 4.1, we see that for $C^* = 1 + \Omega(1/\sqrt{N})$, BC will incur suboptimality that is worse asymptotically than offline RL. In contrast, from Theorem 4.2, we note that offline RL does not scale nearly as poorly with increasing $C^*$. Since offline RL is not as reliant on the performance of the behavior policy, we hypothesize that RL can actually benefit from suboptimal data when this improves coverage. In this section, we aim to answer the following question: *Can offline RL with suboptimal data outperform BC with expert data?*

We show in Corollary 4.2 that, if the suboptimal dataset $\mathcal{D}$ satisfies an additional coverage condition, then running conservative offline RL can attain $\tilde{\mathcal{O}}(\sqrt{H})$ suboptimality

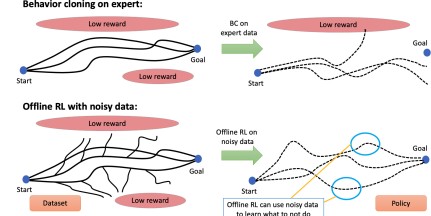

Figure 2: **Illustration showing the intuition behind noisy data.** BC trained on expert data (data composition is shown on the left) may diverge away from the expert and find a poor policy that does not solve the task. If offline RL is provided with noisy expert data that sometimes ventures away from the expert distribution, RL can use this data to learn to stay on the course to the goal.

in the horizon. This implies, perhaps surprisingly, that offline RL with suboptimal data can actually outperform BC, *even when the latter is provided expert data.* Formally:

**Condition 4.2** (Coverage of the optimal policy). $\exists b \in [\log H/N, 1)$ *such that* $\mu$ *satisfies:* $\forall(\mathbf{s}, \mathbf{a}) \in \mathcal{S} \times \mathcal{A}$ *where* $d^*(\mathbf{s}, \mathbf{a}) \geq b/H$, *we have* $\mu(\mathbf{s}, \mathbf{a}) \geq b$.

Intuitively, this means that the data distribution puts sufficient mass on states that have non-negligible density in the optimal policy distribution. Note that this is a weaker condition than prior works that require **(1)** full coverage of the state-action space, and **(2)** enforce a constraint on the empirical state-action visitations $\widehat{\mu}(\mathbf{s}, \mathbf{a})$ instead of $\mu(\mathbf{s}, \mathbf{a})$ [53, 73]. This condition is reasonable when the dataset is collected by $\epsilon$-greedy or a maximum-entropy expert, which is a standard assumption in MaxEnt IRL [74]. Even if the expert is not noisy, we argue that in several real-world applications, creating noisy-expert data is feasible. In many robotics applications, it is practical to augment expert demonstrations with counterfactual data by using scripted exploration policies [21, 23]. Existing work has also simulated trajectories to increase coverage, as was done in self-driving by perturbing the vehicle location [5], or in robotics using learned simulators [51]. For intuition, we provide an illustrative example of the noisy data that can help offline RL in Figure 2.

**Corollary 4.2** (Performance of conservative offline RL with noisy data). *If* $\mu$ *satisfies Condition 4.2, and under Conditions 3.1 and 3.2, the policy* $\widehat{\pi}^*$ *found by conservative offline RL can satisfy:*

$$\mathsf{SubOpt}(\widehat{\pi}^*) \lesssim \sqrt{\frac{H\iota}{bN}} + \frac{H\iota}{bN} + \sqrt{b\iota} + \frac{C^*|\mathcal{S}|\iota}{N}.$$

*If we take* $b \sim \sqrt{H}/N$, *then* $\mathsf{SubOpt}(\widehat{\pi}^*) \lesssim \tilde{\mathcal{O}}(\sqrt{H})$.

The bound in Corollary 4.2 has $\tilde{\mathcal{O}}(\sqrt{H})$ scaling compared to $\tilde{\mathcal{O}}(H)$ for BC. Thus, when the data satisfies the practical coverage conditions (more discussion in Appendix D), offline RL performs better in long-horizon tasks compared to BC with the same amount of expert data.

> **Practical Observation 4.3.** *Offline RL outperforms BC on expert data on long-horizon tasks, when provided with noisy-expert data. Thus, if noisy-expert data is easy to collect (e.g., through scripted policies or by first running standard behavioral cloning, and then storing data from evaluations of the behavior-cloned policy), doing so and then using offline RL can lead to better results.*

### 4.4 COMPARISON OF GENERALIZED BC METHODS AND OFFLINE RL

So far we have studied scenarios where offline RL can outperform naïve BC. One might now wonder how offline RL methods perform relative to generalized BC methods that additionally use reward information to inform learning. We study two such approaches: **(1)** filtered BC [8], which only fits to the top $k$-percentage of trajectories in $\mathcal{D}$, measured by the total reward, and **(2)** BC with one-step policy improvement [6], which fits a Q-function for the behavior policy, then uses the values to perform one-step of policy improvement over the behavior policy. In this section, we aim to answer how these methods perform relative to RL.

**Filtered BC.** In expectation, this algorithm uses $\alpha N$ samples of the offline dataset $\mathcal{D}$ for $\alpha \in [0, 1]$ to perform BC on. This means that the upper bound (Theorem 4.1) will have worse scaling in $N$. For $C^* = 1$, this leads to a strictly worse bound than regular BC. However, for suboptimal data, the filtering step could decrease $C^*$ by filtering out suboptimal trajectories, allowing filtered BC to outperform traditional BC. Nevertheless, from our analysis in Section 4.3, offline RL is still preferred to filtered BC because RL can leverage the noisy data and potentially achieve $O(\sqrt{H})$ suboptimality, whereas even filtered BC would always incur a worse $O(H)$ suboptimality.

**BC with policy improvement.** This algorithm utilizes the entire dataset to estimate the Q-value of the behavior policy, $\widehat{Q}^{\widehat{\pi}_\beta}$, and performs one step of policy improvement using the estimated Q-function, typically via an advantage-weighted update: $\widehat{\pi}^1(\mathbf{a}|\mathbf{s}) = \widehat{\pi}_\beta(\mathbf{a}|\mathbf{s}) \exp(\eta H \widehat{A}^{\widehat{\pi}_\beta}(\mathbf{s}, \mathbf{a}))/\mathbb{Z}_1(\mathbf{s})$. *When would this algorithm perform poorly compared to offline RL?* Intuitively, this would happen when multiple steps of policy improvement are needed to effectively discover high-advantage actions under the behavior policy. This is the case when the the behavior policy puts low density on high-advantage transitions. In Theorem 4.4, we show that more than one step of policy improvement can improve the policy under Condition 4.2, for the special case of the softmax policy parameterization [2].

**Theorem 4.4** (One-step is worse than $k$-step policy improvement). *Assume that the learned policies are represented via a softmax parameterization (Equation 3, Agarwal et al. [2]). Let* $\widehat{\pi}^k$ *denote the*

*policy obtained after k-steps of policy improvement using exponentiated advantage weights. Then, under Condition 4.2, the performance difference between $\widehat{\pi}^k$ and $\widehat{\pi}^1$ is lower-bounded by:*

$$J(\widehat{\pi}^k) - J(\widehat{\pi}^1) \gtrsim \frac{k}{H\eta}\mathbb{E}_{\mathbf{s}\sim\mu}\left[\frac{1}{k}\sum_{t=1}^{k}\log\mathbb{Z}_t(\mathbf{s})\right] - \sqrt{\frac{C^*H\iota}{N}}.$$

A proof of Theorem 4.4 is provided in Appendix B.5. This result implies that when the average exponentiated empirical advantage $1/k\sum_{i=1}^{k}\log\mathbb{Z}_t(\mathbf{s})$ is large enough (*i.e.*, $\geq c_0$ for some universal constant), which is usually the case when the behavior policy is highly suboptimal, then for $k = \mathcal{O}(H)$, multiple steps of policy improvement will improve performance, *i.e.*, $J(\widehat{\pi}^k) - J(\widehat{\pi}^1) = \tilde{\mathcal{O}}(H - \sqrt{H/N})$, where the gap increases with a longer horizon. This is typically the case when the structure of the MDP allow for stitching parts of poor-performing trajectories. One example is in navigation, where trajectories that fail may still contain segments of a successful trajectory.

> **Practical Observation 4.4.** *Using multiple policy improvement steps (i.e., full offline RL) can lead to greatly improved performance on long-horizon tasks, particularly when parts of various trajectories can be concatenated or stitched together to give better performance.*

## 5 EMPIRICAL EVALUATION OF BC AND OFFLINE RL

Having characterized scenarios where offline RL methods can outperform BC in theory, we now validate our results empirically. Concretely, we aim to answer the following questions: **(1)** Does an existing offline RL method trained on expert data outperform BC on expert data in MDPs with few critical points?, **(2)** Can offline RL trained on noisy data outperform BC on expert data?, and **(3)** How does full offline RL compare to the reward-aware BC methods studied in Section 4.4? We first validate our findings on high-dimensional domains below, but we provide some diagnostic experiments on a gridworld domain in Appendix H.

**Evaluation in high-dimensional tasks.** Next, we turn to high-dimensional problems. We consider a diverse set of domains and behavior policies that are representative of practical scenarios: multi-stage robotic manipulation tasks from state (Adroit domains from Fu et al. [14]) and image observations [60], antmaze navigation [14], and 7 Atari games [3]. We use the scripted expert provided by Fu et al. [14] for antmaze and those provided by Singh et al. [60] for manipulation, an RL-trained expert for Atari, and human expert for Adroit [50]. We obtain suboptimal data using failed attempts from a noisy expert policy (*i.e.*, previous policies in the replay buffer for Atari, and noisy scripted experts for antmaze and manipulation). All these tasks utilize sparse rewards such that the return of any trajectory is bounded by a constant much smaller than the horizon. We use CQL [27] as a representative RL method, and utilize Brandfonbrener et al. [6] as a representative BC-PI method.

**Tuning offline RL and BC.** Naïvely running offline RL can lead to poor performance, as noted by prior works [38, 12]. This is also true for BC, but, some solutions such as early stopping based on validation losses, can help improve performance. We claim that an offline tuning strategy is also crucial for offline RL. In our experiments we utilize the offline workflow proposed by Kumar et al. [30] to perform policy selection, and address overfitting and underfitting, purely offline. While this workflow does not fully address all the tuning issues, we find that it is sufficient to improve performance. When the Q-values learned by CQL are extremely negative (typically on the Adroit domains), we utilize a capacity-decreasing dropout regularization with probability $0.4$ on the layers of the Q-function to combat overfitting. On the other hand, when the Q-values exhibit a relatively stable trend (*e.g.*, in Antmaze or Atari), we utilize the DR3 regularizer [29] to increase capacity. Consistent with prior work, we find that naïve offline RL generally performs worse than BC without offline tuning, but we find that *tuned* offline RL can work well. For tuning BC and BC-PI, we applied regularizers such as dropout on the policy to prevent overfitting in Adroit, and utilized a larger ResNet [18] architecture for the robotic manipulation tasks and Atari domains. For BC, we report the performance of the *best* checkpoint found during training, giving BC an unfair advantage, but we still find that *offline-tuned* offline RL can do better better. More details about tuning each algorithm can be found in Appendix F.

**Answers to questions (1) to (3).** For **(1)**, we run CQL and BC on expert data in each task, and present the comparison in Table 1 and Figure 3. While naïve CQL performs comparable or worse than BC in this case, after offline tuning, CQL outperforms BC. This tuning does not require any additional online rollouts. Note that while BC performs better or comparable to RL for antmaze (large) with

| Domain / Behavior Policy | Task/Data Quality | BC | Naïve CQL | Tuned CQL |
|---|---|---|---|---|
| **AntMaze (scripted)** | Medium, Expert | 53.2%±8.7% | 20.8% ± 1.0% | 55.9% ± 3.2% |
| | Large, Expert | **4.83%**±0.8% | 0.0% ± 0.0% | 0.0% ± 0.0% |
| | Medium, Expert w/ diverse initial | 55.2%±6.7% | 19.0% ± 5.2% | **67.0%** ± 7.3% |
| | Large, Expert w/ diverse initial | 1.3%±0.5% | 0.0% ± 0.0 | **5.1%** ± 6.9% |
| **Manipulation (scripted)** | pick-place-open-grasp, Expert | 14.5%±1.8% | 12.3%±5.3% | **23.5%**±6.0% |
| | close-open-grasp, Expert | 17.4%±3.1% | 20.0%±6.0% | **49.7%**±5.4% |
| | open-grasp, Expert | 33.2%±8.1% | 22.8%±5.3% | **51.9%**±6.8% |
| **Adroit (Human)** | hammer-human, Expert | 71.0% ± 9.3% | 62.5% ± 39.0% | **78.1%** ± 6.7% |
| | door-human, Expert | **86.3%** ± 6.5% | 70.3% ± 27.2% | 79.1% ± 4.7% |
| | pen-human, Expert | **73.0** % ± 9.1% | 64.0% ± 6.9% | **74.1%** ± 6.1% |
| | relocate-human, Expert | 0.0% ± 0.0% | 0.0% ± 0.0% | 0.0% ± 0.0% |

Table 1: Offline CQL vs. BC with expert dataset compositions averaged over 3 seeds. While naïve offline CQL often performs comparable or worse than BC, the performance of offline RL improves drastically after offline tuning. Also note that offline RL can improve when provided with diverse initial states in the Antmaze domain. Additionally, note that offline-tuned offline RL outperforms BC significantly in the manipulation domains.

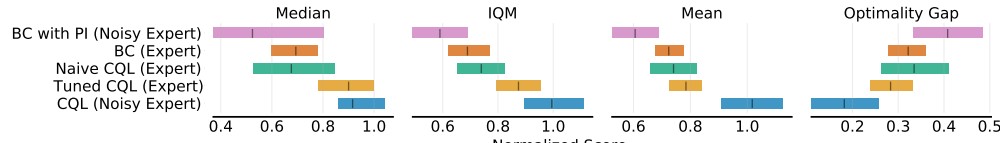

Figure 3: IQM performance of various algorithms evaluated on 7 Atari games under various dataset compositions (per game scores in Table 3). Note that offline-tuned CQL with expert data ("Tuned CQL") outperforms cloning the expert data ("BC (Expert)"), even though naïve CQL is comparable to BC in this setting. When CQL is provided with noisy-expert data, it significantly outperforms cloning the expert policy.

expert data, it performs worse than RL when the data admits a more diverse initial state distribution such that $C^* \neq 1$, even though the behavior policy matches the expert.

For **(2)**, we compare offline RL trained on noisy-expert data with BC trained on on an equal amount of expert data, on domains where noisy-expert data is easy to generate: (a) manipulation domains (Table 2) and (b) Atari games (Figure 3). Observe that CQL outperforms BC and also improves over only using expert data. The

| Task | BC (Expert) | CQL (Noisy Expert) |
|---|---|---|
| pick-place-open-grasp | 14.5% ± 1.8% | **85.7%** ± 3.1% |
| close-open-grasp | 17.4% ± 3.1% | **90.3%** ± 2.3% |
| open-grasp | 33.2% ± 8.1% | **92.4%** ± 4.9% |

Table 2: CQL with noisy-expert data vs BC with expert data with equal dataset size on manipulation tasks. CQL outperforms BC as well as CQL with only expert data.

performance gap also increases with $H$, *i.e.*, open-grasp ($H = 40$) vs pick-place-open-grasp ($H = 80$) vs Atari domains ($H = 27000$). This validates that some form of offline RL with noisy-expert data can outperform BC with expert data, particularly on long-horizon tasks.

Finally, for **(3)**, we compare CQL to a representative BC-PI method [6] trained using noisy-expert data on Atari domains, which present multiple stitching opportunities. The BC-PI method estimates the Q-function of the behavior policy using SARSA and then performs one-step of policy improvement. The results in Figure 3 support what is predicted by our theoretical results, *i.e.*, BC-PI still performs significantly worse than CQL with noisy-expert data, even though we utilized online rollouts for tuning BC-PI and report the best hyperparameters found.

## 6 DISCUSSION

We sought to understand if offline RL is at all preferable over running BC, even provided with expert or near-expert data. While in the worst case, both approaches attain similar performance on expert data, additional assumptions on the environment can provide certain offline RL methods with an advantage. We also show that running RL on noisy-expert, suboptimal data attains more favorable guarantees compared to running BC on expert data for the same task, using equal amounts of data. Empirically, we observe that offline-tuned offline RL can outperform BC on various practical problem domains, with different kinds of expert policies. While our work is an initial step towards understanding when RL presents a favorable approach, there is still plenty of room for further investigation. Our theoretical analysis can be improved to handle function approximation. Understanding if offline RL is preferred over BC for other real-world data distributions is also important. Finally, our work focuses on analyzing cases where we might expect offline RL to outperform BC. An interesting direction is to understand cases where the opposite holds; such analysis would further contribute to this discussion.

ACKNOWLEDGEMENTS

We thank Dibya Ghosh, Yi Su, Xinyang Geng, Tianhe Yu, Ilya Kostrikov and Michael Janner for informative discussions, Karol Hausman for providing feedback on an early version of this paper, and Bo Dai for answering some questions pertaining to Ren et al. [53]. AK thanks George Tucker for informative discussions. We thank the members of RAIL at UC Berkeley for their support and suggestions. We thank anonymous reviewers for feedback on an early version of this paper. This research is funded in part by the DARPA Assured Autonomy Program, the Office of Naval Research, an EECS departmental fellowship, and in part by compute resources from Google Cloud.

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

# Appendices

## A  PSEUDOCODE FOR ALGORITHMS

---

**Algorithm 1** Conservative Offline RL Algorithm

---

**Require:** Offline dataset $\mathcal{D}$, discount factor $\gamma$, and confidence level $\delta$

1: Compute $n(\mathbf{s}, \mathbf{a})$ from $\mathcal{D}$, and estimate $\widehat{r}(\mathbf{s}, \mathbf{a})$, $\widehat{P}(s'|\mathbf{s}, \mathbf{a})$, $\forall (\mathbf{s}, \mathbf{a}) \in \mathcal{S} \times \mathcal{A}$

2: Initialize $\widehat{Q}(\mathbf{s}, \mathbf{a}) \leftarrow 0, \widehat{V}(\mathbf{s}) \leftarrow 0, \forall (\mathbf{s}, \mathbf{a})$

3: **for** $i = 1, 2, \ldots, m$ **do**

4:     Calculate $b(\mathbf{s}, \mathbf{a})$ as:

$$b(\mathbf{s}, \mathbf{a}) \leftarrow \sqrt{\frac{\mathbb{V}(\widehat{P}(\mathbf{s}, \mathbf{a}), \widehat{V}) \log(|\mathcal{S}||\mathcal{A}|m/\delta))}{(n(\mathbf{s}, \mathbf{a}) \wedge 1)}} + \sqrt{\frac{\widehat{r}(\mathbf{s}, \mathbf{a}) \log(|\mathcal{S}||\mathcal{A}|m/\delta)}{(n(\mathbf{s}, \mathbf{a}) \wedge 1)}} + \frac{\log(|\mathcal{S}||\mathcal{A}|m/\delta)}{(n(\mathbf{s}, \mathbf{a}) \wedge 1)}$$

5:     Calculate $\widehat{\pi}^*(\mathbf{s})$ as:

$$\widehat{Q}(\mathbf{s}, \mathbf{a}) \leftarrow \widehat{r}(\mathbf{s}, \mathbf{a}) - b(\mathbf{s}, \mathbf{a}) + \gamma \widehat{P}(\mathbf{s}, \mathbf{a}) \cdot \widehat{V}$$
$$\widehat{V}(\mathbf{s}) \leftarrow \max_a \widehat{Q}(\mathbf{s}, \mathbf{a})$$
$$\widehat{\pi}^*(\mathbf{s}) \leftarrow \arg\max_a \widehat{Q}(\mathbf{s}, \mathbf{a})$$

6: Return $\widehat{\pi}^*$

---

**Algorithm 2** Policy-Constraint Offline RL Algorithm

---

**Require:** Offline dataset $\mathcal{D}$, discount factor $\gamma$, and threshold $b$

1: Compute $n(\mathbf{s}, \mathbf{a})$ from $\mathcal{D}$, and estimate $\widehat{r}(\mathbf{s}, \mathbf{a})$, $\widehat{P}(s'|\mathbf{s}, \mathbf{a})$, $\widehat{\mu}(\mathbf{s}, \mathbf{a})$, $\forall (\mathbf{s}, \mathbf{a}) \in \mathcal{S} \times \mathcal{A}$

2: Compute $\zeta(\mathbf{s}, \mathbf{a}) \leftarrow \mathbb{I}\{\widehat{\mu}(\mathbf{s}, \mathbf{a}) \geq b\}, \forall (\mathbf{s}, \mathbf{a})$

3: Initialize $\widehat{\pi}^*(\mathbf{a}|\mathbf{s}) \leftarrow \frac{1}{|\mathcal{A}|}, \widehat{Q}_\zeta^{\widehat{\pi}^*}(\mathbf{s}, \mathbf{a}) \leftarrow 0, \widehat{V}_\zeta^{\widehat{\pi}^*}(\mathbf{s}) \leftarrow 0, \forall (\mathbf{s}, \mathbf{a})$

4: **for** $\ell = 1, 2, \ldots, k$ **do**

5:     **for** $i = 1, 2, \ldots, m$ **do**

6:         Update $\widehat{Q}_\zeta^{\widehat{\pi}^*}(\mathbf{s}, \mathbf{a}), \widehat{V}_\zeta^{\widehat{\pi}^*}(\mathbf{s})$ as:

$$\widehat{Q}_\zeta^{\widehat{\pi}^*}(\mathbf{s}, \mathbf{a}) \leftarrow \widehat{r}(\mathbf{s}, \mathbf{a}) + \gamma \widehat{P}(\mathbf{s}, \mathbf{a}) \cdot \widehat{V}_\zeta^{\widehat{\pi}^*}$$
$$\widehat{V}_\zeta^{\widehat{\pi}^*}(\mathbf{s}) \leftarrow \sum_{\mathbf{a}} \widehat{\pi}^*(\mathbf{a}|\mathbf{s}) \zeta(\mathbf{s}, \mathbf{a}) \cdot \widehat{Q}(\mathbf{s}, \mathbf{a})$$

7:     Compute $\widehat{\pi}^*$ as:

$$\widehat{\pi}^* \leftarrow \arg\max_\pi \mathbb{E}_{\mathbf{s} \sim \mathcal{D}} \left[ \mathbb{E}_{\mathbf{a} \sim \pi'} \left[ \zeta(\mathbf{s}, \mathbf{a}) \cdot \widehat{Q}_\zeta^\pi(\mathbf{s}, \mathbf{a}) \right] \right]$$

8: Return $\widehat{\pi}^*$.

---

## B  PROOFS

### B.1  PROOF OF THEOREM 4.1

Let $\pi_\beta$ be the behavior policy that we fit our learned policy $\widehat{\pi}_\beta$ to. Recall that the BC algorithm we analyze fits $\widehat{\pi}_\beta$ to choose actions according to the empirical dataset distribution for states that appear in dataset $\mathcal{D}$, and uniformly at random otherwise. We have

$$\mathbb{E}_{\mathcal{D}} \left[ J(\pi^*) - J(\widehat{\pi}_\beta) \right] \leq J(\pi^*) - J(\pi_\beta) + \mathbb{E}_{\mathcal{D}} \left[ J(\pi_\beta) - J(\widehat{\pi}_\beta) \right]$$

The following lemma from Rajaraman et al. [49] bounds the suboptimality from performing BC on a (potentially stochastic) expert, which we adapt below factoring in bounded returns of trajectories from Condition 3.2.

**Lemma B.1** (Theorem 4.4, Rajaraman et al. [49]). *The policy returned by BC on behavior policy $\pi_\beta$ has expected error bounded as*

$$\mathbb{E}_{\mathcal{D}}\left[J(\pi_\beta) - J(\widehat{\pi}_\beta)\right] \leq \frac{SH\log N}{N},$$

*where $\pi_\beta$ could be stochastic.*

Using Lemma B.1, we have $\mathbb{E}_{\mathcal{D}}\left[J(\pi_\beta) - J(\widehat{\pi}_\beta)\right] \leq SH\iota/N$. What remains is bounding the suboptimality of the behavior policy, which we can upper-bound as

$$\begin{aligned}
J(\pi^*) - J(\pi_\beta) &\leq \sum_{t=0}^{\infty}\sum_{\mathbf{s}} \gamma^t \mathbb{P}\left(s_t = s\right) \mathbb{E}_{\pi_\beta(\cdot|s)}\left[\mathbb{I}\{\mathbf{a} \neq \pi_t^*(\mathbf{s})\}\right] \\
&= \frac{1}{2}\sum_{t=0}^{\infty}\sum_{\mathbf{s}} \gamma^t d_t^*(\mathbf{s}) \sum_{\mathbf{a}} |\pi_\beta(\mathbf{a}|\mathbf{s}) - \mathbb{I}\{\mathbf{a} = \pi_t^*(\mathbf{s})\}| \\
&= \frac{1}{2}\sum_{t=0}^{\infty}\sum_{(\mathbf{s},\mathbf{a})} \gamma^t |d_t^*(\mathbf{s})\pi_\beta(\mathbf{a}|\mathbf{s}) - d_t^*(\mathbf{s},\mathbf{a})| \\
&\leq \frac{C^* - 1}{2}H \sum_{(\mathbf{s},\mathbf{a})} \mu(\mathbf{s},\mathbf{a}) \\
&= \frac{(C^* - 1)H}{2},
\end{aligned}$$

where we use the definition of $C^*$ in Condition 3.1. Taking the sum of both terms yields the desired result.

## B.2 PROOF OF THEOREM 4.2

In this section, we proof the performance guarantee for the conservative offline RL algorithm detailed in Algorithm 1. Recall that the algorithm we consider builds upon empirical value iteration but subtracts a penalty during each $Q$-update. Specifically, we initialize $Q_0(\mathbf{s},\mathbf{a}) = 0, V_0(\mathbf{s}) = 0$ for all $(\mathbf{s},\mathbf{a})$. Let $n(\mathbf{s},\mathbf{a})$ be the number of times $(\mathbf{s},\mathbf{a})$ appeared in $\mathcal{D}$, and let $\widehat{r}(\mathbf{s},\mathbf{a})$, $\widehat{P}(\mathbf{s},\mathbf{a})$ be the empirical estimates of their reward and transition probabilities. Then, for each iteration $i \in [m]$:

$$\begin{aligned}
\widehat{Q}_i(\mathbf{s},\mathbf{a}) &\leftarrow \widehat{r}(\mathbf{s},\mathbf{a}) - b_i(\mathbf{s},\mathbf{a}) + \gamma\widehat{P}(\mathbf{s},\mathbf{a})\cdot\widehat{V}_{i-1}, \quad \text{for all } \mathbf{s},\mathbf{a}, \\
\widehat{V}_i(\mathbf{s}) &\leftarrow \max\{\widehat{V}_{t-1}(\mathbf{s}), \max_a \widehat{Q}_i(\mathbf{s},\mathbf{a})\}, \quad \text{for all } s,
\end{aligned}$$

In our algorithm we define the penalty function as

$$b_i(\mathbf{s},\mathbf{a}) \leftarrow \sqrt{\frac{\mathbb{V}(\widehat{P}(\mathbf{s},\mathbf{a}),\widehat{V}_{i-1})\iota}{(n(\mathbf{s},\mathbf{a})\wedge 1)}} + \sqrt{\frac{\widehat{r}(\mathbf{s},\mathbf{a})\iota}{(n(\mathbf{s},\mathbf{a})\wedge 1)}} + \frac{\iota}{(n(\mathbf{s},\mathbf{a})\wedge 1)},$$

where we let $\iota$ to capture all poly-logarithmic terms. As notation, we drop the subscript $i$ to denote the final $\widehat{Q}$ and $\widehat{V}$ at iteration $m$, where $m = H\log N$. Finally, the learned policy $\widehat{\pi}^*$ satisfies $\widehat{\pi}^*(\mathbf{s}) \in \arg\max_a \widehat{Q}(\mathbf{s},\mathbf{a})$ for all $s$, if multiple such actions exist, then the policy samples an action uniformly at random.

### B.2.1 TECHNICAL LEMMAS

**Lemma B.2** (Bernstein's inequality). *Let $X, \{X_i\}_{i=1}^n$ be i.i.d random variables with values in $[0,1]$, and let $\delta > 0$. Then we have*

$$\mathbb{P}\left(\left|\mathbb{E}\left[X\right] - \frac{1}{n}\sum_{i=1}^n X_i\right| > \sqrt{\frac{2\text{Var}\left[X\right]\log(2/\delta)}{n}} + \frac{\log(2/\delta)}{n}\right) \leq \delta.$$

**Lemma B.3** (Theorem 4, Maurer & Pontil [39])**.** *Let $X, \{X_i\}_{i=1}^n$ with $n \geq 2$ be i.i.d random variables with values in $[0, 1]$. Define $\bar{X} = \frac{1}{n} \sum_{i=1}^n X_i$ and $\widehat{\mathrm{Var}}(X) = \frac{1}{n} \sum_{i=1}^n (X_i - \bar{X})^2$. Let $\delta > 0$. Then we have*

$$\mathbb{P}\left( \left| \mathbb{E}\left[ X \right] - \frac{1}{n} \sum_{i=1}^n X_i \right| > \sqrt{\frac{2\widehat{\mathrm{Var}}(\bar{X}) \log(2/\delta)}{n-1}} + \frac{7 \log(2/\delta)}{3(n-1)} \right) \leq \delta \,.$$

**Lemma B.4** (Lemma 4, Ren et al. [53])**.** *Let $\lambda_1, \lambda_2 > 0$ be constants. Let $f : \mathbb{Z}_{\geq 0} \to \mathbb{R}$ be a function such that $f(i) \leq H$, $\forall i$ and $f(i)$ satisfies the recursion*

$$f(i) \leq \sqrt{\lambda_1 f(i+1)} + \lambda_1 + 2^{i+1} \lambda_2 \,.$$

*Then, we have that $f(0) \leq 6(\lambda_1 + \lambda_2)$.*

### B.2.2 PESSIMISM GUARANTEE

The first thing we want to show is that with high probability, the algorithm provides pessimistic value estimates, namely that $\widehat{V}_i(\mathbf{s}) \leq V^*(\mathbf{s})$ for all $t \in [T]$ and $\mathbf{s} \in \mathcal{S}$. To do so, we introduce a notion of a "good" event, which occurs when our empirical estimates of the MDP are not far from the true MDP. We define $\mathcal{E}_1$ to be the event where

$$\left| (\widehat{P}(\mathbf{s}, \mathbf{a}) - P(\mathbf{s}, \mathbf{a})) \cdot \widehat{V}_i \right| \leq \sqrt{\frac{\mathbb{V}(\widehat{P}(\mathbf{s}, \mathbf{a}), \widehat{V}_i)\iota}{(n(\mathbf{s}, \mathbf{a}) \wedge 1)}} + \frac{\iota}{(n(\mathbf{s}, \mathbf{a}) \wedge 1)} \tag{3}$$

holds for all $i \in [m]$ and $(\mathbf{s}, \mathbf{a}) \in \mathcal{S} \times \mathcal{A}$. We also define $\mathcal{E}_2$ to be the event where

$$|\widehat{r}(\mathbf{s}, \mathbf{a}) - r(\mathbf{s}, \mathbf{a})| \leq \sqrt{\frac{\widehat{r}(\mathbf{s}, \mathbf{a})\iota}{(n(\mathbf{s}, \mathbf{a}) \wedge 1)}} + \frac{\iota}{(n(\mathbf{s}, \mathbf{a}) \wedge 1)} \tag{4}$$

holds for all $(\mathbf{s}, \mathbf{a})$.

We want to show that the good event $\mathcal{E} = \mathcal{E}_1 \cap \mathcal{E}_2$ occurs with high probability. The proof mostly follows from Bernstein's inequality in Lemma B.2 . Note that because $\widehat{P}(\mathbf{s}, \mathbf{a})$, $\widehat{V}_i$ are not independent, we cannot straightforwardly apply Bernstein's inequality. We instead use the approach of Agarwal et al. [1] who, for each state $s$, partition the range of $\widehat{V}_i(\mathbf{s})$ within a modified $s$-absorbing MDP to create independence from $\widehat{P}$. The following lemma from Agarwal et al. [1] is a result of such analysis, and is slightly modified below to account for bounded returns of trajectories, *i.e.*, $\widehat{V}_i(\mathbf{s}) \leq 1$:

**Lemma B.5** (Lemma 9, Agarwal et al. [1])**.** *For any iteration $t$, state-action $(\mathbf{s}, \mathbf{a}) \in \mathcal{S} \times \mathcal{A}$ such that $n(\mathbf{s}, \mathbf{a}) \geq 1$, and $\delta > 0$, we have*

$$\mathbb{P}\left( \left| (\widehat{P}(\mathbf{s}, \mathbf{a}) - P(\mathbf{s}, \mathbf{a})) \cdot \widehat{V}_i \right| > \sqrt{\frac{\mathbb{V}(\widehat{P}(\mathbf{s}, \mathbf{a}), \widehat{V}_i)\iota}{n(\mathbf{s}, \mathbf{a})}} + \frac{\iota}{n(\mathbf{s}, \mathbf{a})} \right) \leq \delta \,.$$

Using this, we can show that $\mathcal{E}$ occurs with high probability:

**Lemma B.6.** $\mathbb{P}\left( \mathcal{E} \right) \geq 1 - 2|\mathcal{S}||\mathcal{A}|m\delta$.

*Proof.* For each $i$ and $(\mathbf{s}, \mathbf{a})$, if $n(\mathbf{s}, \mathbf{a}) \leq 1$, then equation 3 and equation 4 hold trivially. For $n(\mathbf{s}, \mathbf{a}) \geq 2$, we have from Lemma B.5 that

$$\mathbb{P}\left( \left| (\widehat{P}(\mathbf{s}, \mathbf{a}) - P(\mathbf{s}, \mathbf{a})) \cdot \widehat{V}_i \right| > \sqrt{\frac{\mathbb{V}(\widehat{P}(\mathbf{s}, \mathbf{a}), \widehat{V}_i)\iota}{n(\mathbf{s}, \mathbf{a})}} + \frac{\iota}{n(\mathbf{s}, \mathbf{a})} \right) \leq \delta \,.$$

Similarly, we can use Lemma B.3 to derive

$$\mathbb{P}\left( |\widehat{r}(\mathbf{s}, \mathbf{a}) - r(\mathbf{s}, \mathbf{a})| > \sqrt{\frac{\widehat{r}(\mathbf{s}, \mathbf{a})\iota}{n(\mathbf{s}, \mathbf{a})}} + \frac{\iota}{n(\mathbf{s}, \mathbf{a})} \right)$$

$$\leq \mathbb{P}\left( |\widehat{r}(\mathbf{s}, \mathbf{a}) - r(\mathbf{s}, \mathbf{a})| > \sqrt{\frac{\widehat{\mathrm{Var}}(\widehat{r}(\mathbf{s}, \mathbf{a}))\iota}{2(n(\mathbf{s}, \mathbf{a}) - 1)}} + \frac{\iota}{2(n(\mathbf{s}, \mathbf{a}) - 1)} \right) \leq \delta \,,$$

where we use that $\widehat{\mathrm{Var}}(\widehat{r}(\mathbf{s}, \mathbf{a})) \leq \widehat{r}(\mathbf{s}, \mathbf{a})$ for $[0, 1]$ rewards, and with slight abuse of notation, let $\iota$ capture all constant factors. Taking the union bound over all $i$ and $(\mathbf{s}, \mathbf{a})$ yields the desired result. $\square$

Now, we can prove that our value estimates are indeed pessimistic.

**Lemma B.7** (Pessimism Guarantee). *On event $\mathcal{E}$, we have that $\widehat{V}_i(\mathbf{s}) \leq V^{\widehat{\pi}^*}(\mathbf{s}) \leq V^*(\mathbf{s})$ for any iteration $i \in [m]$ and state $\mathbf{s} \in \mathcal{S}$.*

*Proof.* We aim to prove the following for any $i$ and $s$: $\widehat{V}_{i-1}(\mathbf{s}) \leq \widehat{V}_i(\mathbf{s}) \leq V^{\widehat{\pi}^*}(\mathbf{s}) \leq V^*(\mathbf{s})$. We prove the claims one by one.

$\widehat{V}_{i-1}(\mathbf{s}) \leq \widehat{V}_i(\mathbf{s})$: This is directly implied by the monotonic update of our algorithm.

$\widehat{V}_i(\mathbf{s}) \leq V^{\widehat{\pi}^*}(\mathbf{s})$: We will prove this via induction. We have that this holds for $\widehat{V}_0$ trivially. Assume it holds for $t - 1$, then we have

$$V^{\widehat{\pi}^*}(\mathbf{s}) \geq \mathbb{E}_{a \sim \widehat{\pi}^*(\cdot|s)} \left[ r(\mathbf{s}, \mathbf{a}) + \gamma P(\mathbf{s}, \mathbf{a}) \cdot \widehat{V}_{i-1} \right]$$

$$\geq \mathbb{E}_a \left[ \widehat{r}(\mathbf{s}, \mathbf{a}) - b_i(\mathbf{s}, \mathbf{a}) + \gamma \widehat{P}(\mathbf{s}, \mathbf{a}) \cdot \widehat{V}_{t-1} \right] +$$

$$\mathbb{E}_a \left[ b_i(\mathbf{s}, \mathbf{a}) - (\widehat{r}(s, a) - r(\mathbf{s}, \mathbf{a})) - \gamma(\widehat{P}(\mathbf{s}, \mathbf{a}) - P(\mathbf{s}, \mathbf{a})) \cdot \widehat{V}_{i-1} \right]$$

$$\geq \widehat{V}_t(\mathbf{s}) \,,$$

where we use that

$$b_i(\mathbf{s}, \mathbf{a}) = \sqrt{\frac{\mathbb{V}(\widehat{P}(\mathbf{s}, \mathbf{a}), \widehat{V}_{i-1})\iota}{(n(\mathbf{s}, \mathbf{a}) \wedge 1)}} + \sqrt{\frac{\widehat{r}(\mathbf{s}, \mathbf{a})\iota}{(n(\mathbf{s}, \mathbf{a}) \wedge 1)}} + \frac{\iota}{(n(\mathbf{s}, \mathbf{a}) \wedge 1)}$$

$$\geq (\widehat{r}(s, a) - r(\mathbf{s}, \mathbf{a})) + \gamma(\widehat{P}(\mathbf{s}, \mathbf{a}) - P(\mathbf{s}, \mathbf{a})) \cdot \widehat{V}_{i-1}$$

under event $\mathcal{E}$.

Finally, the claim of $V^{\widehat{\pi}^*}(\mathbf{s}) \leq V^*(\mathbf{s})$ is trivial, which completes the proof of our pessimism guarantee. $\square$

### B.2.3 VALUE DIFFERENCE LEMMA

Now, we are ready to derive the performance guarantee from Theorem 4.2. The following lemma is a bound on the estimation error of our pessimistic $Q$-values.

**Lemma B.8.** *On event $\mathcal{E}$, the following holds for any $i \in [m]$ and $(\mathbf{s}, \mathbf{a}) \in \mathcal{S} \times \mathcal{A}$:*

$$Q^*(\mathbf{s}, \mathbf{a}) - \widehat{Q}_i(\mathbf{s}, \mathbf{a}) \leq \gamma P(\mathbf{s}, \mathbf{a}) \cdot (Q^*(\cdot; \pi^*) - \widehat{Q}_{i-1}(\cdot; \pi^*)) + 2b_i(\mathbf{s}, \mathbf{a}) \,, \tag{5}$$

*where $f(\cdot; \pi)$ satisfies $f(s; \pi) = \sum_{\mathbf{a}} \pi(\mathbf{a}|\mathbf{s}) f(\mathbf{s}, \mathbf{a})$.*

*Proof.* We have,

$$Q^*(\mathbf{s}, \mathbf{a}) - \widehat{Q}_i(\mathbf{s}, \mathbf{a})$$

$$= r(\mathbf{s}, \mathbf{a}) + \gamma P(\mathbf{s}, \mathbf{a}) \cdot V^* - (\widehat{r}(\mathbf{s}, \mathbf{a}) - b_i(\mathbf{s}, \mathbf{a}) + \gamma \widehat{P}(\mathbf{s}, \mathbf{a}) \cdot \widehat{V}_{t-1})$$

$$= b_i(\mathbf{s}, \mathbf{a}) + r(\mathbf{s}, \mathbf{a}) - \widehat{r}(\mathbf{s}, \mathbf{a}) + \gamma P(\mathbf{s}, \mathbf{a}) \cdot (V^* - \widehat{V}_{t-1}) + \gamma(P(\mathbf{s}, \mathbf{a}) - \widehat{P}(\mathbf{s}, \mathbf{a})) \cdot \widehat{V}_{t-1}$$

$$\leq \gamma P(\mathbf{s}, \mathbf{a}) \cdot (V^* - \widehat{V}_{t-1}) + 2b_i(\mathbf{s}, \mathbf{a})$$

$$\leq \gamma P(\mathbf{s}, \mathbf{a}) \cdot (Q^*(\cdot; \pi^*) - \widehat{Q}_{t-1}(\cdot; \pi^*)) + 2b_i(\mathbf{s}, \mathbf{a}) \,.$$

The first inequality is due by definition of $\mathcal{E}$ and the second is because $\widehat{V}_{t-1} \geq \max_a \widehat{Q}_{t-1}(\cdot, a) \geq \widehat{Q}_i(\cdot, \pi^*)$. $\qquad\square$

By recursively applying Lemma B.8, we can derive the following value difference lemma:

**Lemma B.9** (Value Difference Lemma). *On event $\mathcal{E}$, at any iteration $i \in [m]$, we have*

$$J(\pi^*) - J(\widehat{\pi}^*) \leq \gamma^i + 2 \sum_{t=1}^{i} \sum_{(\mathbf{s},\mathbf{a})} \gamma^{i-t} d_{i-t}^*(\mathbf{s}, \mathbf{a}) b_t(\mathbf{s}, \mathbf{a}), \tag{6}$$

*where $d_t^*(\mathbf{s}, \mathbf{a}) = \mathbb{P}(s_t = \mathbf{s}, \mathbf{a}_t = a; \pi^*)$.*

*Proof.* We have,

$$J(\pi^*) - J(\widehat{\pi}^*) = \mathbb{E}_\rho \left[ V^*(\mathbf{s}) - V^{\widehat{\pi}^*}(\mathbf{s}) \right] \leq \mathbb{E}_\rho \left[ V^*(\mathbf{s}) - \widehat{V}_i(\mathbf{s}) \right] \leq \rho(Q^*(\cdot; \pi^*) - \widehat{Q}_i(\cdot; \pi^*))$$

where we use Lemma B.7 in the first inequality. As shorthand, let $P^\pi \in \mathbb{R}^{(\mathcal{S} \times \mathcal{A}) \times (\mathcal{S} \times \mathcal{A})}$ where $P^\pi(\mathbf{s}, \mathbf{a}, s', a') = P(s'|\mathbf{s}, \mathbf{a})\pi(a'|s')$ be the transition matrix for policy $\pi$. Now, we can apply Lemma B.8 recursively to derive

$$\begin{aligned}
\rho^{\pi^*}(Q^* - \widehat{Q}_i) &\leq \rho^{\pi^*} \left( \gamma P^{\pi^*}(Q - \widehat{Q}_{i-1}) + 2b_i \right) \\
&\leq \rho^{\pi^*} \left( \gamma P^{\pi^*} \left( \gamma P^{\pi^*}(Q^* - \widehat{Q}_{i-2}) + 2b_{i-1} \right) + 2b_i \right) \\
&\leq \ldots \\
&\leq \rho^{\pi^*}(\gamma P^{\pi^*})^i(Q^* - \widehat{Q}_0) + 2 \sum_{t=1}^{i} \rho^{\pi^*}(\gamma P^{\pi^*})^{i-t} b_t \\
&\leq \gamma^i \mathbf{1} + 2 \sum_{t=1}^{i} \gamma^{i-t} d_{i-t}^* b_t
\end{aligned}$$

where we use that $d_t^* = \rho^{\pi^*}(P^{\pi^*})^t$. This yields the desired result. $\qquad\square$

Now, we are ready to bound the desired quantity $\mathsf{SubOpt}(\widehat{\pi}^*) = \mathbb{E}_\mathcal{D}[J(\pi^*) - J(\widehat{\pi}^*)]$. We have

$$\begin{aligned}
\mathbb{E}_\mathcal{D}[J(\pi^*) - J(\widehat{\pi}^*)] &= \mathbb{E}_\mathcal{D}\left[ \sum_s \rho(\mathbf{s})(V^*(\mathbf{s}) - V^{\widehat{\pi}^*}(\mathbf{s})) \right] \tag{7} \\
&= \underbrace{\mathbb{E}_\mathcal{D}\left[ \mathbb{I}\{\bar{\mathcal{E}}\} \sum_s \rho(\mathbf{s})(V^*(\mathbf{s}) - V^{\widehat{\pi}^*}(\mathbf{s})) \right]}_{:=\Delta_1} \\
&\quad + \underbrace{\mathbb{E}_\mathcal{D}[\mathbb{I}\{\exists \mathbf{s} \in \mathcal{S}, \; n(\mathbf{s}, \pi^*(\mathbf{s})) = 0] \sum_s \rho(\mathbf{s})(V^*(\mathbf{s}) - V^{\widehat{\pi}^*}(\mathbf{s}))\}}_{:=\Delta_2} \\
&\quad + \underbrace{\mathbb{E}_\mathcal{D}[\mathbb{I}\{\forall \mathbf{s} \in \mathcal{S}, \; n(\mathbf{s}, \pi^*(\mathbf{s})) > 0] \mathbb{I}\{\mathcal{E}\} \sum_s \rho(\mathbf{s})(V^*(\mathbf{s}) - V^{\widehat{\pi}^*}(\mathbf{s}))\}}_{:=\Delta_3}.
\end{aligned}$$

We bound each term individually. The first is bounded as $\Delta_1 \leq \mathbb{P}(\bar{\mathcal{E}}) \leq 2|\mathcal{S}||\mathcal{A}|m\delta \leq \frac{\iota}{N}$ for choice of $\delta = \frac{1}{2|\mathcal{S}||\mathcal{A}|HN}$.

### B.2.4 BOUND ON $\Delta_2$

For the second term, we have

$$
\begin{aligned}
\Delta_2 &\leq \sum_s \rho(\mathbf{s}) \mathbb{E}_{\mathcal{D}} \left[ \mathbb{I}\{n(\mathbf{s}, \pi^*(\mathbf{s})) = 0]\} \right. \\
&\leq H \sum_{\mathbf{s}} d^*(\mathbf{s}, \pi^*(\mathbf{s})) \mathbb{E}_{\mathcal{D}} \left[ \mathbb{I}\{n(\mathbf{s}, \pi^*(\mathbf{s})) = 0\} \right] \\
&\leq C^* H \sum_{\mathbf{s}} \mu(\mathbf{s}, \pi^*(\mathbf{s}))(1 - \mu(\mathbf{s}, \pi^*(\mathbf{s})))^N \\
&\leq \frac{4 C^* |\mathcal{S}| H}{9N},
\end{aligned}
$$

where we use that $\rho(\mathbf{s}) \leq H d^*(\mathbf{s}, \pi^*(\mathbf{s}))$, and that $\max_{p \in [0,1]} p(1-p)^N \leq \frac{4}{9N}$.

### B.2.5 BOUND ON $\Delta_3$

What remains is bounding the last term, which we know from Lemma B.9 is bounded by

$$
\Delta_3 \leq \frac{1}{N} + 2\mathbb{E}_{\mathcal{D}} \left[ \mathbb{I}\{\forall \mathbf{s} \in \mathcal{S}, \ n(\mathbf{s}, \pi^*(\mathbf{s})) > 0\} \sum_{t=0}^m \sum_{(\mathbf{s},\mathbf{a})} \gamma^{m-t} d^*_{m-t}(\mathbf{s}, \mathbf{a}) b_t(\mathbf{s}, \mathbf{a}) \right],
$$

where we use that $\gamma^m \leq \frac{1}{N}$ for $m = H \log N$. Recall that $b_t(\mathbf{s}, \mathbf{a})$ is given by

$$
b_t(\mathbf{s}, \mathbf{a}) = \sqrt{\frac{\mathbb{V}(\widehat{P}(\mathbf{s}, \mathbf{a}), \widehat{V}_{t-1})\iota}{n(\mathbf{s}, \mathbf{a})}} + \sqrt{\frac{\widehat{r}(\mathbf{s}, \mathbf{a})\iota}{n(\mathbf{s}, \mathbf{a})}} + \frac{\iota}{n(\mathbf{s}, \mathbf{a})}
$$

We can bound the summation of each term separately. For the third term we have,

$$
\begin{aligned}
\mathbb{E}_{\mathcal{D}} \left[ \sum_{t=1}^m \sum_{(\mathbf{s},\mathbf{a})} \gamma^{m-t} d^*_{m-t}(\mathbf{s}, \mathbf{a}) \frac{\iota}{n(\mathbf{s}, \mathbf{a})} \right] &\leq \sum_{t=0}^{m-1} \sum_{(\mathbf{s},\mathbf{a})} \gamma^t d^*_t(\mathbf{s}, \mathbf{a}) \mathbb{E}_{\mathcal{D}} \left[ \frac{\iota}{n(\mathbf{s}, \mathbf{a})} \right] \\
&\leq \sum_{\mathbf{s}} \sum_{t=0}^{\infty} \gamma^t d^*_t(\mathbf{s}, \pi^*(\mathbf{s})) \frac{\iota}{N\mu(\mathbf{s}, \pi^*(\mathbf{s}))} \\
&\leq \frac{H\iota}{N} \sum_{\mathbf{s}} \left( (1-\gamma) \sum_{t=0}^{\infty} \gamma^t d^*_t(\mathbf{s}, \pi^*(\mathbf{s})) \right) \frac{1}{\mu(\mathbf{s}, \pi^*_h(\mathbf{s}))} \\
&\leq \frac{C^* |\mathcal{S}| H\iota}{N}.
\end{aligned}
$$

Here we use Jensen's inequality and that $(1-\gamma) \sum_{t=1}^{\infty} \gamma^t d^*_t(\mathbf{s}, \mathbf{a}) \leq C^* \mu(\mathbf{s}, \mathbf{a})$ for any $(\mathbf{s}, \mathbf{a})$. For the second term, we similarly have

$$
\begin{aligned}
\mathbb{E}_{\mathcal{D}} &\left[ \sum_{t=1}^m \sum_{(\mathbf{s},\mathbf{a})} \gamma^{m-t} d^*_{m-t}(\mathbf{s}, \mathbf{a}) \sqrt{\frac{\widehat{r}(\mathbf{s}, \mathbf{a})\iota}{n(\mathbf{s}, \mathbf{a})}} \right] \\
&\leq \mathbb{E}_{\mathcal{D}} \left[ \sqrt{\sum_{t=1}^m \sum_{(\mathbf{s},\mathbf{a})} \gamma^{m-t} d^*_{m-t}(\mathbf{s}, \mathbf{a}) \frac{\iota}{n(\mathbf{s}, \mathbf{a})}} \right] \sqrt{\sum_{t=1}^m \sum_{(\mathbf{s},\mathbf{a})} \gamma^{m-t} d^*_{m-t}(\mathbf{s}, \mathbf{a}) \widehat{r}(\mathbf{s}, \mathbf{a})} \\
&\leq \sqrt{\frac{C^* |\mathcal{S}| H\iota}{N}},
\end{aligned}
$$

where we use Cauchy-Schwarz, then Condition 3.2 to bound the total estimated reward. Finally, we consider the first term of $b_t(\mathbf{s}, \mathbf{a})$

$$
\mathbb{E}_{\mathcal{D}} \left[ \sum_{t=1}^{m} \sum_{(\mathbf{s}, \mathbf{a})} \gamma^{m-t} d_{m-t}^*(\mathbf{s}, \mathbf{a}) \sqrt{\frac{\mathbb{V}(\widehat{P}(\mathbf{s}, \mathbf{a}), \widehat{V}_{t-1}) \iota}{n(\mathbf{s}, \mathbf{a})}} \right]
$$

$$
\leq \mathbb{E}_{\mathcal{D}} \left[ \sqrt{\sum_{t=1}^{m} \sum_{(\mathbf{s}, \mathbf{a})} \gamma^{m-t} d_{m-t}^*(\mathbf{s}, \mathbf{a}) \frac{\iota}{n(\mathbf{s}, \mathbf{a})}} \sqrt{\sum_{t=1}^{m} \sum_{(\mathbf{s}, \mathbf{a})} \gamma^{m-t} d_{m-t}^*(\mathbf{s}, \mathbf{a}) \mathbb{V}(\widehat{P}(\mathbf{s}, \mathbf{a}), \widehat{V}_{t-1})} \right]
$$

$$
\leq \sqrt{\frac{C^* |\mathcal{S}| H \iota}{N}} \sqrt{\sum_{t=1}^{m} \sum_{(\mathbf{s}, \mathbf{a})} \gamma^{m-t} d_{m-t}^*(\mathbf{s}, \mathbf{a}) \mathbb{V}(\widehat{P}(\mathbf{s}, \mathbf{a}), \widehat{V}_{t-1})} \,.
$$

Similar to what was done in Zhang et al. [72], Ren et al. [53] for finite-horizon MDPs, we can bound this term using variance recursion for infinite-horizon ones. Define

$$
f(i) := \sum_{t=1}^{\infty} \sum_{(\mathbf{s}, \mathbf{a})} \gamma^{m-t} d_{m-t}^*(\mathbf{s}, \mathbf{a}) \mathbb{V}(\widehat{P}(\mathbf{s}, \mathbf{a}), (\widehat{V}_{t-1})^{2^i}) \,. \tag{8}
$$

Using Lemma 3 of Ren et al. [53] for the infinite-horizon case, we have the following recursion:

$$
f(i) \leq \sqrt{\frac{C^* |\mathcal{S}| H \iota}{N} f(i+1)} + \frac{C^* |\mathcal{S}| H \iota}{N} + 2^{i+1}(\Phi + 1) \,,
$$

where

$$
\Phi := \sqrt{\frac{C^* |\mathcal{S}| H \iota}{N}} \sqrt{\sum_{t=1}^{m} \sum_{(\mathbf{s}, \mathbf{a})} \gamma^{m-t} d_{m-t}^*(\mathbf{s}, \mathbf{a}) \mathbb{V}(\widehat{P}(\mathbf{s}, \mathbf{a}), \widehat{V}_{t-1})} + \frac{C^* |\mathcal{S}| H \iota}{N} \tag{9}
$$

Using Lemma B.4, we can bound $f(0) = \mathcal{O}\left( \frac{C^* |\mathcal{S}| H \iota}{N} + \Phi + 1 \right)$. Using that for constant $c$,

$$
\Phi = \sqrt{\frac{C^* |\mathcal{S}| H \iota}{N} f(0)} + \frac{C^* |\mathcal{S}| H \iota}{N}
$$

$$
\leq \sqrt{\frac{C^* |\mathcal{S}| H \iota}{N} \left( \frac{c C^* |\mathcal{S}| H \iota}{N} + c\Phi + c \right)} + \frac{C^* |\mathcal{S}| H \iota}{N}
$$

$$
\leq \frac{c\Phi}{2} + \frac{2c C^* |\mathcal{S}| H \iota}{N} + \frac{c}{2}
$$

we have that

$$
\Phi \leq c + \frac{4c C^* |\mathcal{S}| H \iota}{N} \,.
$$

Substituting this back into the inequality for $\Phi$ yields,

$$
\Phi = \mathcal{O}\left( \sqrt{\frac{C^* |\mathcal{S}| H \iota}{N}} + \frac{C^* |\mathcal{S}| H \iota}{N} \right)
$$

Finally, we can bound

$$
\Delta_3 \leq \sqrt{\frac{C^* |\mathcal{S}| H \iota}{N}} + \frac{C^* |\mathcal{S}| H \iota}{N} \,.
$$

Combining the bounds for the three terms yields the desired result.

### B.3 PROOF OF COROLLARY 4.1

In this section, we will provide a proof of Corollary 4.1.

**Intuition and strategy:** The intuition for why offline RL can outperform BC in this setting, despite the near-expert data comes from the fact that RL can better control the performance of the policy on non-critical states that it has seen before in the data. This is not true for BC since it does not utilize reward information. Intuitively, we can partition the states into two categories:

1. critical states including those that are visited enough and those that are not visited enough in the dataset → bound this via the machinery already discussed in Appendix B.2.

2. non-critical states → offline RL can perform well here: we show that under some technical conditions, it can exponentially fast collapse at a good-enough action at such states, while BC might still choose bad actions.

We restate the definition of non-critical points below for convenience.

**Definition B.1** ((Non-critical points, restated)). *A state $\mathbf{s}$ is said to be non-critical (i.e., $\mathbf{s} \notin \mathcal{C}$) if there exists a subset $\mathcal{G}(\mathbf{s})$ of $\varepsilon$-good actions, such that,*

$$\forall \mathbf{a} \in \mathcal{G}(\mathbf{s}), \ \max_{\mathbf{a}'} Q^*(\mathbf{s}, \mathbf{a}') - Q^*(\mathbf{s}, \mathbf{a}) \leq \frac{\varepsilon}{H}, \ \ and$$

$$\forall \mathbf{a} \in \mathcal{A} \smallsetminus \mathcal{G}(\mathbf{s}), \ \max_{\mathbf{a}'} Q^*(\mathbf{s}, \mathbf{a}') - Q^*(\mathbf{s}, \mathbf{a}) \simeq \Delta(\mathbf{s}),$$

*and $\Delta(\mathbf{s}) \leq \Delta_0$, where we will define $\Delta_0$ later.*

Recall that our condition on critical states was that for any policy $\pi$ in the MDP, the expected occupancy of critical states is bounded by $p_c$ and $p_c \leq \frac{1}{H}$. We restate this condition below:

**Condition B.1** ((Occupancy of critical states is low, restated)). *For any policy $\pi$, the total occupancy of critical states in the MDP is bounded by $p_c$, i.e.,*

$$\sum_{\mathbf{s} \in \mathcal{C}} d^\pi(\mathbf{s}) \leq p_c.$$

Now, using this definition, we will bound the total suboptimality of RL separately at critical and non-critical states. At critical states, i.e., scenario (1) from the list above, we will reuse the existing machinery for the general setting from Appendix B.2. For non-critical states (2), we will utilize a stronger argument for RL that relies on the fact that $\Delta_0$ and $|\mathcal{G}(\mathbf{s})|$ are large enough, and consider policy-constraint offline RL algorithms that extract the policy via a pessimistic policy extraction step.

This policy constraint algorithm follows Algorithm 2, but in addition to using the learned Q-function for policy extraction, it uses a local pessimistic term in Step 7, i.e.,

$$\widehat{\pi}^* \leftarrow \arg\max_\pi \mathbb{E}_{\mathbf{s} \sim \mathcal{D}} \left[ \mathbb{E}_{\mathbf{a} \sim \pi} \left[ \widehat{Q}^\pi(\mathbf{s}, \mathbf{a}) - \sigma b(\mathbf{s}, \mathbf{a}) \right] \right], \tag{10}$$

where $b(\mathbf{s}, \mathbf{a})$ refers to the bonus (Equation 2) and $\sigma > 0$. This modification makes the algorithm more pessimistic but allows us to obtain guarantees when $\forall (\mathbf{s}, \mathbf{a}), n(\mathbf{s}, \mathbf{a}) \geq n_0$ as we can express the probability that the policy makes a mistake at a given state-action pair in terms of the count of the given state-action pair. Finally, as all state-action pairs are observed in the dataset, the learned Q-values, $\widehat{Q}$ can only differ from $Q^*$ in a bounded manner, i.e., $\left| \widehat{Q}(\mathbf{s}, \mathbf{a}) - Q^*(\mathbf{s}, \mathbf{a}) \right| \leq \varepsilon_0$.

**Lemma B.10** (Fast convergence at non-critical states.). *Consider a non-critical state $\mathbf{s} \in \mathcal{D}$, such that $|\mathcal{D}(\mathbf{s})| \geq n_0$ and let $\forall \mathbf{a} \in \mathcal{A}, \frac{\pi^*(\mathbf{a}|\mathbf{s})}{\mu(\mathbf{a}|\mathbf{s})} \leq \alpha < 2$, where $\mu(\mathbf{a}|\mathbf{s})$ is the conditional action distribution at state $\mathbf{s}$. Then, the probability that the policy $\widehat{\pi}_{RL}$ obtained from the modified offline RL algorithm above does not choose a good action at state $\mathbf{s}$ is upper bounded as ($c_0$ is a universal constant):*

$$\mathbb{P} \left( \widehat{\pi}_{RL}(\mathbf{s}) \notin \mathcal{G}(\mathbf{s}) \right) \leq \exp \left( -n_0 \cdot \frac{\alpha^2}{2(\alpha - 1)} \cdot \left[ \frac{|\mathcal{G}(\mathbf{s})| \sigma c_0}{n_0 \cdot (c_1 \sigma + \Delta)} - \frac{1}{\alpha} \right]^2 \right).$$

*Proof.* First note that:

$$\mathbb{P}\left[\widehat{\pi}_{\mathrm{RL}}(\mathbf{s}) \notin \mathcal{G}(\mathbf{s})\right] = \mathbb{P}\left[\exists \mathbf{a} \notin \mathcal{G}(\mathbf{s}), \text{ s.t. } \forall \mathbf{a}_g \in \mathcal{G}(\mathbf{s}), \ \ \widehat{Q}(\mathbf{s}, \mathbf{a}) - \sigma b(\mathbf{s}, \mathbf{a}) \geq \widehat{Q}(\mathbf{s}, \mathbf{a}_g) - \sigma b(\mathbf{s}, \mathbf{a}_g)\right]$$

$$\leq \mathbb{P}\left[\exists \mathbf{a} \notin \mathcal{G}(\mathbf{s}), \text{ s.t. } \forall \mathbf{a}_g \in \mathcal{G}(\mathbf{s}), \widehat{Q}(\mathbf{s}, \mathbf{a}) - Q^*(\mathbf{s}, \mathbf{a}) \geq \widehat{Q}(\mathbf{s}, \mathbf{a}_g) - Q^*(\mathbf{s}, \mathbf{a}_g) + \Delta(\mathbf{s}) - \sigma b(\mathbf{s}, \mathbf{a}_g)\right]$$

$$\leq \mathbb{P}\left[\cap_{\mathbf{a}_g \in \mathcal{G}(\mathbf{s})} \{\sigma b(\mathbf{s}, \mathbf{a}_g) \geq \Delta(\mathbf{s}) - 2\varepsilon_0\}\right]$$

$$\leq \mathbb{P}\left[\cap_{\mathbf{a}_g \in \mathcal{G}(\mathbf{s})} \{n(\mathbf{s}, \mathbf{a}) \leq n_\Delta\}\right],$$

where $n_\Delta$ corresponds to the maximum value of $n(\mathbf{s}, \mathbf{a})$ such that

$$b(\mathbf{s}, \mathbf{a}) := \frac{c_1}{\sqrt{n(\mathbf{s}, \mathbf{a}) \wedge 1}} + \frac{c_2}{n(\mathbf{s}, \mathbf{a}) \wedge 1} \geq \frac{\Delta(\mathbf{s}) - 2\varepsilon_0}{\sigma}. \tag{11}$$

We can then upper bound the probability using a Chernoff bound for Bernoulli random variables (the Bernoulli random variable here is whether the action sampled from the dataset $\mu(\mathbf{a}|\mathbf{s})$ belongs to the good set $\mathcal{G}(\mathbf{s})$ or not):

$$\mathbb{P}\left[\cap_{\mathbf{a} \in \mathcal{G}(\mathbf{s})} \{n(\mathbf{s}, \mathbf{a}) \leq n_\Delta(\mathbf{s})\}\right] \leq \mathbb{P}\left[\sum_{\mathbf{a} \in \mathcal{G}(\mathbf{s})} n(\mathbf{s}, \mathbf{a}) \leq |\mathcal{G}(\mathbf{s})| n_\Delta(\mathbf{s})\right].$$

$$\leq \mathbb{P}\left[\frac{\sum_{\mathbf{a} \in \mathcal{G}(\mathbf{s})} n(\mathbf{s}, \mathbf{a})}{n_0} \leq \frac{|\mathcal{G}(\mathbf{s})| n_\Delta(\mathbf{s})}{n_0}\right]$$

$$\leq \exp\left(-n_0 \mathrm{KL}\left(\mathrm{Bern}\left(\frac{|\mathcal{G}(\mathbf{s})| n_\Delta(\mathbf{s})}{n_0}\right) \| \mathrm{Bern}(1/\alpha)\right)\right).$$

The above expression can then be simplified using the inequality that $\mathrm{KL}(p + \varepsilon \| p) \geq \frac{\varepsilon^2}{2p(1-p)}$ if $p \geq 1/2$, which is the case here, since $p = \frac{1}{\alpha}$, and $\alpha \leq 2$. Thus, we can simplify the bound as:

$$\mathbb{P}\left[\cap_{\mathbf{a} \in \mathcal{G}(\mathbf{s})} \{n(\mathbf{s}, \mathbf{a}) \leq n_\Delta(\mathbf{s})\}\right] \leq \exp\left(-n_0 \cdot \frac{\alpha^2}{2(\alpha - 1)} \cdot \left[\frac{|\mathcal{G}(\mathbf{s})| n_\Delta(\mathbf{s})}{n_0} - \frac{1}{\alpha}\right]^2\right).$$

To finally express the bound in terms of $\Delta(\mathbf{s})$, we note that the maximum-valued solution $n_\Delta(\mathbf{s})$ to Equation 11 satisfies $n_\Delta(\mathbf{s}) \simeq \frac{\sigma}{c_1 \sigma + \Delta(\mathbf{s})}$. Substituting the above in the bound, we obtain:

$$\mathbb{P}\left[\cap_{\mathbf{a} \in \mathcal{G}(\mathbf{s})} \{n(\mathbf{s}, \mathbf{a}) \leq n_\Delta(\mathbf{s})\}\right] \leq \exp\left(-n_0 \cdot \frac{\alpha^2}{2(\alpha - 1)} \cdot \left[\frac{|\mathcal{G}(\mathbf{s})| \sigma c_0}{n_0 \cdot (c_1 \sigma + \Delta)} - \frac{1}{\alpha}\right]^2\right),$$

for some universal constants $c_0$ and $c_1$. $\qquad\square$

We will now use Lemma B.10 to prove the formal comparison of RL and BC when only a few critical states are encountered in a given trajectory (i.e., under Condition B.1).

**Theorem B.1** (Critical states). *Assume that the data distribution $\mu$ satisfies $\rho(\mathbf{s}) := \frac{d^{\pi^*}(\mathbf{s})}{\mu(\mathbf{s})} = 1$ and $C^* \leq 1 + \frac{1}{N}$. Let $\Delta(\mathbf{s}) \geq \Delta_0$ for all $\mathbf{s} \notin \mathcal{C}$. Then, under Condition B.1, for an appropriate value of $\Delta_0$ and $p_c \lesssim \frac{1}{H}$, the worst-case suboptimality incurred by conservative offline RL is upper bounded by the lower bound on performance from BC from Theorem 4.3, i.e.,*

$$\mathsf{SubOpt}(\widehat{\pi}_{\mathrm{RL})}) \lesssim \mathsf{SubOpt}(\widehat{\pi}_{\mathrm{BC}}).$$

*Proof.* For any learning algorithm that returns a policy $\widehat{\pi}^*$, the suboptimality is given by:

$$J(\pi^*) - J(\widehat{\pi}^*) = H \sum_{\mathbf{s} \in \mathcal{S}} d^{\widehat{\pi}^*}(\mathbf{s}) \left(Q^*(s; \pi^*) - Q^*(s; \widehat{\pi}^*)\right)$$

$$= H \underbrace{\sum_{\mathbf{s} \in \mathcal{C}} d^{\widehat{\pi}^*}(\mathbf{s}) \left(Q^*(s; \pi^*) - Q^*(s; \widehat{\pi}^*)\right)}_{\text{term (i): bound using techniques from Appendix B.2}} + H \underbrace{\sum_{\mathbf{s} \notin \mathcal{C}} d^{\widehat{\pi}^*}(\mathbf{s}) \left(Q^*(s; \pi^*) - Q^*(s; \widehat{\pi}^*)\right)}_{\text{term (ii): we control this term better for RL}}$$

The first term (i) corresponds to the value difference at critical states, and the second term corresponds to the value difference at non-critical states. To bound the first term, we can consider it as the value difference under a modified MDP where the advantage for all actions at all states that are non-critical is 0. One way to construct such an MDP is to take each $s \notin \mathcal{C}$ and modify the reward and transitions so that $r(\mathbf{s}, \mathbf{a}) = r(\mathbf{s}, \pi^*(\mathbf{s})), P(\mathbf{s}, \mathbf{a}) = P(\mathbf{s}, \pi^*(\mathbf{s}))$ for all actions. Let us denote the performance function for this modified MDP be $J_\mathcal{C}$, then we have

$$J(\pi^*) - J(\widehat{\pi}^*) \leq J_\mathcal{C}(\pi^*) - J_\mathcal{C}(\widehat{\pi}^*) + \varepsilon$$

The first term can be bounded as was done in Appendix B.2. Namely, we can show the following equivalent of Lemma B.9:

$$J_\mathcal{C}(\pi^*) - J_\mathcal{C}(\widehat{\pi}^*) \leq \gamma^i + 2 \sum_{t=1}^{i} \sum_{\mathbf{s} \in \mathcal{C}} \gamma^{i-t} d_{i-t}^*(\mathbf{s}, \pi^*(\mathbf{s})) b_t(\mathbf{s}, \pi^*(\mathbf{s})) .$$

Bounding this for $i = H\iota$ can be done exactly as in was done in Appendix B.2.4 and B.2.5, except with the dependence on all states $\mathcal{S}$ replaced by $\mathcal{C}$ only the critical ones. We get the following bound:

$$J_\mathcal{C}(\pi^*) - J_\mathcal{C}(\widehat{\pi}^*) \leq \sqrt{\frac{C^* p_c |\mathcal{S}| H\iota}{N}} + \frac{C^* p_c |\mathcal{S}| H\iota}{N}$$

Now, we will focus on the second term, which we will control tightly for RL using Lemma B.10. We can decompose this term into separate components for good and bad actions:

$$\mathbb{E}_\mathcal{D} [\text{term (ii)}] = \mathbb{E}_\mathcal{D} \left[ \sum_{\mathbf{s} \notin \mathcal{C}} \sum_{\mathbf{a} \in \mathcal{A}} d^{\widehat{\pi}^*}(\mathbf{s}) \widehat{\pi}^*(\mathbf{a}|\mathbf{s}) (Q^*(s; \pi^*) - Q^*(s, a)) \right]$$

$$= \mathbb{E}_\mathcal{D} \left[ \sum_{\mathbf{s} \notin \mathcal{C}} \sum_{\mathbf{a} \in \mathcal{G}(\mathbf{s})} d^{\widehat{\pi}^*}(\mathbf{s}, \mathbf{a}) (Q^*(s; \pi^*) - Q^*(s, a)) \right] + \mathbb{E}_\mathcal{D} \left[ \sum_{\mathbf{s} \notin \mathcal{C}} \sum_{\mathbf{a} \in \mathcal{A} \smallsetminus \mathcal{G}(\mathbf{s})} d^{\widehat{\pi}^*}(\mathbf{s}, \mathbf{a}) (Q^*(s; \pi^*) - Q^*(s, a)) \right]$$

We bound each term independently:

$$\mathbb{E}_\mathcal{D} \left[ \sum_{\mathbf{s} \notin \mathcal{C}} \sum_{\mathbf{a} \in \mathcal{G}(\mathbf{s})} d^{\widehat{\pi}^*}(\mathbf{s}, \mathbf{a}) (Q^*(s; \pi^*) - Q^*(s, a)) \right] \leq \mathbb{E}_\mathcal{D} \left[ \sum_{\mathbf{s} \notin \mathcal{C}} d^{\widehat{\pi}^*}(\mathbf{s}) \widehat{\pi}^*(\mathcal{G}(\mathbf{s})|\mathbf{s}) \cdot \frac{\varepsilon}{H} \right]$$

$$\mathbb{E}_\mathcal{D} \left[ \sum_{\mathbf{s} \notin \mathcal{C}} \sum_{\mathbf{a} \in \mathcal{A} \smallsetminus \mathcal{G}(\mathbf{s})} d^{\widehat{\pi}^*}(\mathbf{s}, \mathbf{a}) (Q^*(s; \pi^*) - Q^*(s, a)) \right] \leq \mathbb{E}_\mathcal{D} \left[ \sum_{\mathbf{s} \notin \mathcal{C}} d^{\widehat{\pi}^*}(\mathbf{s}) (1 - \widehat{\pi}^*(\mathcal{G}(\mathbf{s})|\mathbf{s})) (\Delta(\mathbf{s}) + \varepsilon') \right]$$

The first equation corresponds to bounding the suboptimality due to good actions. The second term above bounds the suboptimality due to bad actions. The first term is controlled as best as it can using the definition of critical states. The second term can be controlled by applying Lemma B.10. Further note that since $\forall (\mathbf{s}, \mathbf{a}), \frac{d^{\pi^*}(\mathbf{s}, \mathbf{a})}{\mu(\mathbf{s}, \mathbf{a})} \leq C^*$, and per the assumption in this theorem, $\forall \mathbf{s}, \frac{d^{\pi^*}(\mathbf{s})}{\mu(\mathbf{s})} = 1$, where we assume that $0/0 = 1$. Therefore, at each state, $\forall \mathbf{s}, \mathbf{a}, \frac{\pi^*(\mathbf{a}|\mathbf{s})}{\mu(\mathbf{a}|\mathbf{s})} \leq C^*$. Also note that in this case, we are interested in the setting where $C^* = 1 + \mathcal{O}\left(\frac{1}{N}\right)$, and as a result, $\frac{1}{C^*} \approx \frac{N}{N+1}$. Therefore, the upper bound for term (ii) is given by:

$$\mathbb{E}_\mathcal{D} \left[ \sum_{\mathbf{s} \notin \mathcal{C}} d^{\widehat{\pi}^*}(\mathbf{s}) (1 - \widehat{\pi}^*(\mathcal{G}(\mathbf{s})|\mathbf{s})) \Delta(\mathbf{s}) \right]$$

$$\lesssim \mathbb{E}_\mathcal{D} \left[ \sum_{\mathbf{s} \notin \mathcal{C}} d^{\widehat{\pi}^*}(\mathbf{s}) \cdot \Delta(\mathbf{s}) \cdot \exp\left( -n_0 \cdot \frac{(N+1)^2}{2(N)} \cdot \left[ \frac{|\mathcal{G}(\mathbf{s})| c_0 \sigma}{n_0 \cdot (c_1 \sigma + \Delta(\mathbf{s}))} - \frac{N+1}{N} \right]^2 \right) \right]$$

$$\lesssim (1 - p_c) \, \Delta_0 \exp\left( -n_0 \cdot \frac{(N+1)^2}{2(N)} \cdot \left[ \frac{|\mathcal{G}(\mathbf{s})| c_0 \sigma}{n_0 \cdot (c_1 \sigma + \Delta_0)} - \frac{N+1}{N} \right]^2 \right) := f_{\text{RL}}(N+1, \Delta_0),$$

where $1 - p_c$ appears from the condition of bounded critical states (Definition 4.1). On the other hand, the corresponding term for BC grows as $\frac{1}{N+1}$, and is given by:

$$\mathbb{E}_{\mathcal{D}} \left[ \sum_{\mathbf{s} \notin \mathcal{C}} d^{\widehat{\pi}^*}(\mathbf{s}) \left(1 - \widehat{\pi}^*(\mathcal{G}(\mathbf{s})|\mathbf{s})\right) \Delta(\mathbf{s}) \right] \lesssim (1 - p_c) \cdot \max_{\mathbf{s} \notin \mathcal{C}} \Delta(\mathbf{s}) \cdot \frac{1}{N+1} := g_{\mathrm{BC}}(N+1, \Delta_0)$$

The bound for BC matches the information-theoretic lower-bound from Theorem 4.3, implying that this bound is tight for BC.

We will now compare the bounds for RL and BC by noting that the function $h(N, \Delta) := \frac{f_{\mathrm{RL}}(N+1, \Delta)}{g_{\mathrm{BC}}(N+1, \Delta)}$ can be set to $\leq \frac{1}{\sqrt{H}}$ since $f$ is an exponential function of $-N$, and $g$ decays linearly in $N$, by controlling $\Delta_0$ and $|\mathcal{G}(\mathbf{s})|$ are implicitly defined as a function of $N$. Per condition (Condition B.1), if $\Delta \geq \Delta_0$, then $h(N, \Delta) = \mathcal{O}(H)$. This means that the suboptimalities of RL and BC compare as follows:

$$\frac{\mathsf{SubOpt}(\widehat{\pi}_{\mathrm{RL}})}{\mathsf{SubOpt}(\widehat{\pi}_{\mathrm{BC}})} = \frac{\sqrt{p_c \frac{C^*|\mathcal{S}|H\iota}{N}} + \frac{C^* p_c |\mathcal{S}|H\iota}{N} + (1 - p_c) \cdot \blacksquare \cdot \frac{1}{\sqrt{H}}}{p_c \frac{H}{N} + (1 - p_c)\blacksquare}.$$

By setting $p_c = \frac{1}{H}$, we get

$$\mathsf{SubOpt}(\widehat{\pi}_{\mathrm{BC}}) \gtrsim \mathsf{SubOpt}(\widehat{\pi}_{\mathrm{RL}})\sqrt{H} \gtrsim \mathsf{SubOpt}(\widehat{\pi}_{\mathrm{RL}}).$$

$\square$

## B.4 PROOF OF COROLLARY 4.2

The proof of Corollary 4.2 is a slight modification of the one for Theorem 4.2. For brevity, we will point out the parts of the proof that change, and simply defer to the proof in Appendix B.2 for parts that are similar. Recall the decomposition for suboptimality in equation 7, which we restate below:

$$\mathbb{E}_{\mathcal{D}} \left[ J(\pi^*) - J(\widehat{\pi}^*) \right] = \underbrace{\mathbb{E}_{\mathcal{D}} \left[ \mathbb{I}\{\bar{\mathcal{E}}\} \sum_s \rho(\mathbf{s})(V^*(\mathbf{s}) - V^{\widehat{\pi}^*}(\mathbf{s})) \right]}_{\Delta_1}$$

$$+ \underbrace{\mathbb{E}_{\mathcal{D}} \left[ \mathbb{I}\{\exists \mathbf{s} \in \mathcal{S}, \ n(\mathbf{s}, \pi^*(\mathbf{s})) = 0\} \sum_s \rho(\mathbf{s})(V^*(\mathbf{s}) - V^{\widehat{\pi}^*}(\mathbf{s})) \right\}}_{\Delta_2}$$

$$+ \underbrace{\mathbb{E}_{\mathcal{D}} \left[ \mathbb{I}\{\forall \mathbf{s} \in \mathcal{S}, \ n(\mathbf{s}, \pi^*(\mathbf{s})) > 0\} \mathbb{I}\{\mathcal{E} \sum_s \rho(\mathbf{s})(V^*(\mathbf{s}) - V^{\widehat{\pi}^*}(\mathbf{s}))\}\}}_{\Delta_3}.$$

$\Delta_1$ is bounded by $\frac{\iota}{N}$ as before.

### B.4.1 BOUND ON $\Delta_2$

The bound for $\Delta_2$ changes slightly from Appendix B.2.4 due to accounting for the lower-bound on $\mu(\mathbf{s}, \mathbf{a}) \geq b \geq \frac{\log H}{N}$. We have

$$\Delta_2 \leq \sum_s \rho(\mathbf{s})\mathbb{E}_{\mathcal{D}} \left[ \mathbb{I}\{n(\mathbf{s}, \pi^*(\mathbf{s})) = 0\} \right]$$

$$\leq H \sum_\mathbf{s} d^*(\mathbf{s}, \pi^*(\mathbf{s}))\mathbb{E}_{\mathcal{D}} \left[ \mathbb{I}\{n(\mathbf{s}, \pi^*(\mathbf{s})) = 0\} \right]$$

$$\leq H \sum_\mathbf{s} d^*(\mathbf{s}, \pi^*(\mathbf{s}))\mathbb{I}\{d^*(\mathbf{s}, \pi^*(\mathbf{s})) \leq \frac{b}{H}\} + H \sum_\mathbf{s} d^*(\mathbf{s}, \pi^*(\mathbf{s}))\mathbb{E}_{\mathcal{D}} \left[ \mathbb{I}\{n(\mathbf{s}, \pi^*(\mathbf{s})) = 0\} \right]$$

$$\leq |\mathcal{S}|c + C^* H \sum_\mathbf{s} \mu(\mathbf{s}, \pi^*(\mathbf{s}))(1 - \mu(\mathbf{s}, \pi^*(\mathbf{s})))^N$$

$$\leq |\mathcal{S}|b + \frac{C^*|\mathcal{S}|\iota}{N},$$

where we use that $\rho(\mathbf{s}) \leq H d^*(\mathbf{s}, \pi^*(\mathbf{s}))$, and that

$$\max_{p \in [\frac{\log H}{N}, 1]} p(1 - p)^N \leq \frac{\log H}{N} \left(1 - \frac{\log H}{N}\right)^N \leq \frac{\log H}{HN}.$$

### B.4.2 BOUND ON $\Delta_3$

Due to the lower bound on $\mu(\mathbf{s}, \mathbf{a}) \geq b$, we can instead bound,

$$
\begin{aligned}
\mathbb{E}_{\mathcal{D}} \left[ \sum_{t=1}^{m} \sum_{(\mathbf{s}, \mathbf{a})} \gamma^{m-t} d_{m-t}^*(\mathbf{s}, \mathbf{a}) \, \frac{\iota}{n(\mathbf{s}, \mathbf{a})} \right] &\leq \sum_{t=0}^{m-1} \sum_{(\mathbf{s}, \mathbf{a})} \gamma^t d_t^*(\mathbf{s}, \mathbf{a}) \mathbb{E}_{\mathcal{D}} \left[ \frac{\iota}{n(\mathbf{s}, \mathbf{a})} \right] \\
&\leq \sum_{\mathbf{s}} \sum_{t=0}^{\infty} \gamma^t d_t^*(\mathbf{s}, \pi^*(\mathbf{s})) \, \frac{\iota}{N \mu(\mathbf{s}, \pi^*(\mathbf{s}))} \\
&\leq \mathbb{I}\{d^*(\mathbf{s}, \mathbf{a}) \leq \frac{b}{H}\} H \sum_{\mathbf{s}} \left( (1-\gamma) \sum_{t=0}^{\infty} \gamma^t d_t^*(\mathbf{s}, \pi^*(\mathbf{s})) \right) + \\
&\quad \frac{H\iota}{Nc} \sum_{\mathbf{s}} \left( (1-\gamma) \sum_{t=0}^{\infty} \gamma^t d_t^*(\mathbf{s}, \pi^*(\mathbf{s})) \right) \\
&\leq b + \frac{H\iota}{bN}.
\end{aligned}
$$

The analysis for bounding $\Delta_3$ proceeds exactly as in Appendix B.2.5 but using the new bound. Namely, we end up with the recursion

$$
f(i) \leq \sqrt{\frac{H\iota}{bN} + b} \, \sqrt{f(i+1)} + \frac{H\iota}{bN} + b + 2^{i+1}(\Phi + 1),
$$

where

$$
\Phi := \sqrt{\frac{H\iota}{bN} + b} \, \sqrt{\sum_{t=1}^{m} \sum_{(\mathbf{s}, \mathbf{a})} \gamma^{m-t} d_{m-t}^*(\mathbf{s}, \mathbf{a}) \mathbb{V}(\widehat{P}(\mathbf{s}, \mathbf{a}), \widehat{V}_{t-1})} + \frac{H\iota}{Nb} + b.
$$

Using Lemma B.4 and proceeding as in Appendix B.2.5 yields the bound

$$
\Delta_3 \leq \sqrt{\frac{H\iota}{bN}} + \frac{H\iota}{bN} + \sqrt{b\iota}.
$$

Combining the new bounds for $\Delta_2, \Delta_3$ results in the bound in the Corollary 4.2.

### B.5 PROOF OF THEOREM 4.4

The proof of Theorem 4.4 builds on analysis by Agarwal et al. [2] that we apply to policies with a softmax parameterization, which we define below.

**Definition B.2** (Softmax parameterization). *For a given $\theta \in \mathbb{R}^{|\mathcal{S}| \times |\mathcal{A}|}$, $\pi_\theta(\mathbf{a}|\mathbf{s}) = \frac{\exp(\theta_{\mathbf{s}, \mathbf{a}})}{\sum_{\mathbf{a}'} \exp(\theta_{\mathbf{s}, \mathbf{a}'})}$.*

We consider generalized BC algorithms that perform advantage-weighted policy improvement for $k$ improvement steps. A BC algorithm with $k$-step policy improvement is defined as follows:

**Definition B.3** (BC with $k$-step policy improvement). *Let $\widehat{A}^k(\mathbf{s}, \mathbf{a})$ denote the advantage of action $\mathbf{a}$ at state $\mathbf{s}$ under a given policy $\widehat{\pi}_k$, where the policy $\widehat{\pi}^k(\mathbf{a}|\mathbf{s})$ is defined via the recursion:*

$$
\widehat{\pi}^{k+1}(\mathbf{a}|\mathbf{s}) := \widehat{\pi}^k(\mathbf{a}|\mathbf{s}) \frac{\exp(\eta H \widehat{A}^k(\mathbf{s}, \mathbf{a}))}{\mathbb{Z}_k(\mathbf{s})},
$$

*starting from $\widehat{\pi}^0(\mathbf{a}|\mathbf{s}) = \widehat{\pi}_\beta$. Then, BC with $k$-step policy improvement returns $\widehat{\pi}^k$.*

This advantage weighted update is utilized in practical works such as Brandfonbrener et al. [6], which first estimates the Q-function of the behavior policy using the offline dataset, i.e, $\widehat{Q}^0(\mathbf{s}, \mathbf{a})$, and then computes $\widehat{\pi}^1$ as the final policy returned by the algorithm. To understand the performance difference between multiple values of $k$, we first utilize essentially Lemma 5 from Agarwal et al. [2], which we present below for completeness:

**Lemma B.11** (Lower bound on policy improvement in the empirical MDP, $\widehat{M}$)**.** *The iterates* $\widehat{\pi}^k$ *generated by* $k$-*steps of policy improvement, for any initial state distributions* $\rho_0(\mathbf{s})$ *satisfy the following lower-bound on improvement:*

$$\widehat{J}(\widehat{\pi}^{k+1}) - \widehat{J}(\widehat{\pi}^k) := \mathbb{E}_{\mathbf{s}_0 \sim \rho_0} \left[ \widehat{V}^{\widehat{\pi}_{k+1}}(\mathbf{s}_0) \right] - \mathbb{E}_{\mathbf{s}_0 \sim \rho_0} \left[ \widehat{V}^{\widehat{\pi}_k}(\mathbf{s}_0) \right] \geq \frac{1}{\eta H} \mathbb{E}_{\mathbf{s}_0 \sim \rho_0} \log \mathbb{Z}_t(\mathbf{s}_0). \quad (12)$$

*Proof.* We utilize the performance difference lemma in the empirical MDP to show this:

$$\begin{aligned}
\widehat{J}(\widehat{\pi}^{k+1}) - \widehat{J}(\widehat{\pi}^k) &= H \mathbb{E}_{\mathbf{s} \sim d^{\widehat{\pi}_{k+1}}} \left[ \sum_{\mathbf{a}} \widehat{\pi}_{k+1}(\mathbf{a}|\mathbf{s}) \widehat{A}^k(\mathbf{s}, \mathbf{a}) \right] \\
&= \frac{1}{\eta} \mathbb{E}_{\mathbf{s} \sim d^{\widehat{\pi}_{k+1}}} \left[ \sum_{\mathbf{a}} \widehat{\pi}^{k+1}(\mathbf{a}|\mathbf{s}) \log \frac{\widehat{\pi}^{k+1}(\mathbf{a}|\mathbf{s}) \mathbb{Z}_k(\mathbf{s})}{\widehat{\pi}^k(\mathbf{a}|\mathbf{s})} \right] \\
&= \frac{1}{\eta} \mathbb{E}_{\mathbf{s} \sim d^{\widehat{\pi}_{k+1}}} \left[ D_{\text{KL}}(\widehat{\pi}^{k+1}(\cdot|\mathbf{s}) || \widehat{\pi}^k(\cdot|\mathbf{s})) \right] + \frac{1}{\eta} \mathbb{E}_{\mathbf{s} \sim d^{\widehat{\pi}_{k+1}}} \left[ \log \mathbb{Z}_k(\mathbf{s}) \right] \\
&\geq \frac{1}{\eta} \mathbb{E}_{\mathbf{s} \sim d^{\widehat{\pi}_{k+1}}} \left[ \log \mathbb{Z}_k(\mathbf{s}) \right].
\end{aligned}$$

Finally, note that the final term $\log \mathbb{Z}_t(\mathbf{s})$ is always positive because of Jensen's inequality, and the fact that the expected advantage under a given policy is $0$ for any MDP. $\square$

Utilizing Lemma B.11, we can then lower bound the total improvement of the learned policy in the actual MDP as:

$$J(\widehat{\pi}^k) - J(\widehat{\pi}^l) \geq \underbrace{J(\widehat{\pi}^k) - \widehat{J}(\widehat{\pi}^k)}_{(a)} + \underbrace{\widehat{J}(\widehat{\pi}^k) - \widehat{J}(\widehat{\pi}^l)}_{(b)} - \underbrace{J(\widehat{\pi}^l) - \widehat{J}(\widehat{\pi}^l)}_{(c)}$$

$$\geq \frac{1}{\eta} \sum_{j=l}^{k} \mathbb{E}_{\mathbf{s} \sim d^{\widehat{\pi}_{j+1}}} \left[ \log \mathbb{Z}_j(\mathbf{s}) \right] - \sqrt{\frac{C^* H \iota}{N}}$$

where the $\sqrt{C^* H \iota / N}$ guarantee for terms (a) and (c) arises under the conditions studied in Section 4.3.

**Interpretation of Theorem 4.4.** Theorem 4.4 says that if atleast $k$ many updates can be made to the underlying empirical MDP, $\widehat{M}$, such that each update is non-trivially lower-bounded, i.e., $\mathbb{E}_{\mathbf{s} \sim d^{\widehat{\pi}}_{k+1}} \left[ \log \mathbb{Z}_k(\mathbf{s}) \right] \geq c_0 > 0$, then the performance improvement obtained by $k$-steps of policy improvement is bounded below by $kc_0/\eta - \mathcal{O}(\sqrt{H/N})$. This result indicates that if $k = \mathcal{O}(H)$ many high advantage policy updates are possible in a given empirical MDP, then the methods with that perform $\mathcal{O}(H)$ steps of policy improvement will attain higher performance than the counterparts that perform only one update.

This is typically the case in maze navigation-style environments, where $\mathcal{O}(H)$ many possible high-advantage updates are possible on the empirical MDP, especially by "stitching" parts of suboptimal trajectories to obtain a much better trajectory. Therefore, we expect that in offline RL problems where stitching is possible, offline RL algorithms will attain an improved performance compared to one or a few-steps of policy improvement.

## C GUARANTEES FOR POLICY-CONSTRAINT OFFLINE RL

In this section, we analyze a policy-constraint offline algorithm [34] that constrains the policy to choose a safe set of actions by explicitly preventing action selection from previously unseen, low-density actions. The algorithm we consider builds upon the MBS-PI algorithm from Liu et al. [36], which truncates Bellman backups and policy improvement steps from low-density, out-of-support state-action pairs. The algorithm is described in detail in Algorithm 2, but we provide a summary below. Let $\widehat{\mu}(\mathbf{s}, \mathbf{a})$ denote the empirical state-action distribution and choose a constant $b$. Then, let

$\zeta(\mathbf{s}, \mathbf{a}) = 1\{\widehat{\mu}(\mathbf{s}, \mathbf{a}) \geq b\}$ be the indicator of high-density state-action tuples. The algorithm we analyze performs the following update until convergence:

$$\widehat{Q}_\zeta^\pi(\mathbf{s}, \mathbf{a}) \leftarrow \widehat{r}(\mathbf{s}, \mathbf{a}) + \gamma \sum_{(\mathbf{s}', \mathbf{a}')} \widehat{P}(\mathbf{s}'|\mathbf{s}, \mathbf{a})\pi(\mathbf{a}'|\mathbf{s}')\zeta(\mathbf{s}', \mathbf{a}') \cdot \widehat{Q}_\zeta^\pi(\mathbf{s}', \mathbf{a}'), \quad \text{for all } (\mathbf{s}, \mathbf{a}),$$

$$\widehat{\pi} \leftarrow \arg\max_\pi \mathbb{E}_{\mathbf{s} \sim \mathcal{D}} \left[ \mathbb{E}_{\mathbf{a} \sim \pi'} \left[ \zeta(\mathbf{s}, \mathbf{a}) \cdot \widehat{Q}_\zeta^\pi(\mathbf{s}, \mathbf{a}) \right] \right],$$

In order to derive performance guarantees for this generic policy-constraint algorithm, we define the notion of a $\zeta-$covered policy following Liu et al. [36] in Definition C.1. The total occupancy of all out-of-support state-action pairs (i.e., $(\mathbf{s}, \mathbf{a})$ such that $\zeta(\mathbf{s}, \mathbf{a}) = 0$) under a $\zeta$-covered policy is bounded by a small constant $U$, which depends on the threshold $b$. Let $\pi_\zeta^*$ denote the best performing $\zeta$-covered policy.

**Definition C.1** ($\zeta$-covered). $\pi$ *is called* $\zeta$-covered if $\sum_{(\mathbf{s}, \mathbf{a})}(1 - \zeta(\mathbf{s}, \mathbf{a}))d^\pi(\mathbf{s}, \mathbf{a}) \leq (1 - \gamma)U(b)$.

Equipped with this definition C.1, Lemma C.1 shows that the total value estimation error of any given $\zeta-$covered policy, $\pi$, $|J(\pi) - \widehat{J}_\zeta(\pi)|$ is upper bounded in expectation over the dataset

**Lemma C.1** (Value estimation error of a $\zeta$-covered policy). *For any given $\zeta$-covered policy $\pi$, under Condition 3.2, the estimation error $|J(\pi) - \widehat{J}_\zeta(\pi)|$ is bounded as:*

$$\mathbb{E}_\mathcal{D} \left[ \left| J(\pi) - \widehat{J}_\zeta(\pi) \right| \right] \lesssim \sqrt{\frac{C^*|\mathcal{S}|H\iota}{N}} + \frac{C^*|\mathcal{S}|H\iota}{N} + U(b) \tag{13}$$

*Proof.* To prove this lemma, we consider the following decomposition of the policy performance estimate:

$$\left| J(\pi) - \widehat{J}_\zeta(\pi) \right|$$

$$= \sum_{t=0}^\infty \sum_{(\mathbf{s}, \mathbf{a})} \gamma^t d_t^\pi(\mathbf{s}, \mathbf{a}) \left[ \sum_{(\mathbf{s}', \mathbf{a}')} \left( \widehat{P}(\mathbf{s}'|\mathbf{s}, \mathbf{a})\zeta(\mathbf{s}', \mathbf{a}') - P(\mathbf{s}'|\mathbf{s}, \mathbf{a}) \right) \cdot \widehat{Q}^\pi(\mathbf{s}', \mathbf{a}') \right]$$

$$= \underbrace{\sum_{t=0}^\infty \sum_{(\mathbf{s}, \mathbf{a})} \gamma^t d_t^\pi(\mathbf{s}, \mathbf{a}) \sum_{(\mathbf{s}', \mathbf{a}')} (\widehat{P}(\mathbf{s}'|\mathbf{s}, \mathbf{a}) - P(\mathbf{s}'|\mathbf{s}, \mathbf{a})) \cdot \zeta(\mathbf{s}', \mathbf{a}') \cdot \pi(\mathbf{a}'|\mathbf{s}') \cdot \widehat{Q}(\mathbf{s}', \mathbf{a}')}_{\Delta_1 : \text{bound using concentrability and variance recursion}}$$

$$+ \underbrace{\sum_{t=0}^\infty \gamma^t d_t^\pi(\mathbf{s}, \mathbf{a}) \sum_{(\mathbf{s}', \mathbf{a}')} P(\mathbf{s}'|\mathbf{s}, \mathbf{a}) \cdot (1 - \zeta(\mathbf{s}', \mathbf{a}')) \cdot \pi(\mathbf{a}'|\mathbf{s}') \cdot \widehat{Q}^\pi(\mathbf{s}', \mathbf{a}')}_{\Delta_2 : \text{bias due to leaving support; upper bounded due to } \zeta\text{-cover}}$$

To bound the inner summation over $(\mathbf{s}', \mathbf{a}')$ in term (a), we can apply Lemma B.5 since $\widehat{P}(\mathbf{s}'|\mathbf{s}, \mathbf{a})$ and $\zeta(\mathbf{s}', \mathbf{a}')$ are not independent, to obtain a horizon-free bound. Finally, we use Condition 3.1 to bound the density ratios, in expectation over the randomness in dataset $\mathcal{D}$, identical to the proof for the conservative lower-confidence bound method from before. Formally, using Lemma B.5, we get, with high probability $\geq 1 - \delta$:

$$\forall (\mathbf{s}, \mathbf{a}) \text{ s.t. } n(\mathbf{s}, \mathbf{a}) \geq 1, \left| \left( \widehat{P}(\mathbf{s}, \mathbf{a}) - P(\mathbf{s}, \mathbf{a}) \right) \cdot \widehat{V}_\zeta^\pi \right| \leq \sqrt{\frac{\mathbb{V}(\widehat{P}(\mathbf{s}, \mathbf{a}), \widehat{V}_\zeta^\pi)\iota}{n(\mathbf{s}, \mathbf{a})}} + \frac{\iota}{n(\mathbf{s}, \mathbf{a})},$$

where we utilized the fact that $\widehat{V}_\zeta^\pi \leq \widehat{V}^\pi \leq 1$ due to Condition 3.2. For bounding $\Delta_2$, we note that this term is bounded by the definition of $\zeta$-covered policy:

$$\Delta_2 \leq \sum_{t=0}^\infty \gamma^t(1 - \gamma)U(b) \leq U(b). \tag{14}$$

Thus, the overall policy evaluation error is given by:

$$\left| J(\pi) - \widehat{J}_\zeta(\pi) \right| \lesssim \sum_{t=0}^\infty \gamma^t d_t^\pi(\mathbf{s}, \mathbf{a}) \left[ \sqrt{\frac{\mathbb{V}(\widehat{P}(\mathbf{s}, \mathbf{a}), \widehat{V}_\zeta^\pi)\iota}{n(\mathbf{s}, \mathbf{a})}} + \frac{\iota}{n(\mathbf{s}, \mathbf{a})} \right] + U(b). \tag{15}$$

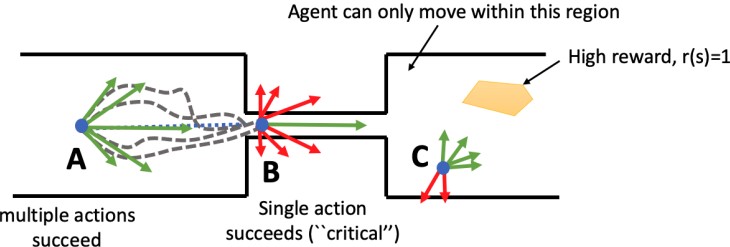

Figure 4: **(Figure 1 restated) Illustration showing the intuition behind critical points in a navigation task.** The agent is supposed to navigate to a high-reward region marked as the yellow polygon, without crashing into the walls. For different states, **A**, **B** and **C** that we consider, the agent has a high volume of actions that allow it to reach the goal at states **A** and **C**, but only few actions that allow it to do so at state **B**. States around **A** and **C** are not critical, and so this task has only a small volume of critical states (i.e., those in the thin tunnel).

Equation 15 mimics the $\Phi$ term in equation 9 that is bounded in Section B.2.5, with an additional offset $U(b)$. Hence, we can reuse the same machinery to show the bound in expectation over the randomness in the dataset, which completes the proof. □

Using Lemma C.1, we can now that the policy constraint algorithm attains a favorable guarantee when compared to the best policy that is $\zeta$-covered:

**Theorem C.1** (Performance of policy-constraint offline RL). *Under Condition 3.2, the policy $\widehat{\pi}^*$ incurs bounded suboptimality against the best $\zeta$-covered policy, with high probability $\geq 1 - \delta$:*

$$\mathbb{E}_{\mathcal{D}}\left[J(\pi_\zeta^*) - J(\widehat{\pi}^*)\right] \lesssim \sqrt{\frac{C^*|\mathcal{S}|H\iota}{N}} + \frac{C^*|\mathcal{S}|H\iota}{N} + 2U(b).$$

To prove this theorem, we use the result of Lemma C.1 for the fixed policy, that is agnostic of the dataset, and then again use the recursion as before to bound the value of the data-dependent policy. The latter uses Lemma B.5 and ends up attaining a bound previously found in Appendix B.2.5, which completes the proof of this Theorem. When the term $U(b)$ is small, such that $U(b) \leq \mathcal{O}(H^{0.5-\varepsilon})$ for $\varepsilon > 0$, then we find that the guarantee in Theorem C.1 matches that in Theorem 4.2, modulo a term that grows slower in the horizon than the other terms in the bound. If $U(b)$ is indeed small, then all properties that applied to conservative offline RL shall also follow for policy-constraint algorithms.

**Note on the bound.** We conjecture that it is possible to get rid of the $U(b)$ term, under certain assumptions on the support indicator $\zeta(\mathbf{s}, \mathbf{a})$, and by relating the values of $\zeta(\mathbf{s}, \mathbf{a})$ and $\zeta(\mathbf{s}', \mathbf{a}')$, at consecutive state-action tuples. For example, if $\zeta(\mathbf{s}', \mathbf{a}') = 1 \implies \zeta(\mathbf{s}, \mathbf{a}) = 1$, then we can derive a stronger guarantee.

# D    INTUITIVE ILLUSTRATIONS OF THE CONDITIONS ON THE ENVIRONMENT AND PRACTICAL GUIDELINES FOR VERIFYING THEM

In this section, we present intuitive illustrations of the various conditions we study and discuss practical guidelines that allow a practitioner to verify whether they are likely to hold for their problem domain. We focus on Conditions 4.1 and 4.2, as Condition 3.2 is satisfied in very common settings such as learning from sparse rewards obtained at the end of an episode, indicating success or failure.

## D.1    CONDITION 4.1

To provide intuition behind when this condition is satisfied, we first provide an example of a sample navigation domain in Figure 4, to build examples of critical states. As shown in the figure, the task is to navigate to the high-reward yellow-colored region. We wish to understand if states marked as **A**, **B** and **C** are critical or not. The region where the agent can move is very wide in the neighborhood around state **A**, very narrow around state **B** and wide again around state **C**. In this case, as marked on the figure if the agent executes actions shown via green arrows at the various states, then it is likely to

sill finish the task, while if it executes the actions shown via red arrows it is likely not on track to reach the high-reward region.

Several actions at states **A** and **C** allow the agent to still reach the goal, without crashing into the walls, while only one action at state **B** allows so, as shown in the figure. Thus, state **B** is a "critical state" as per Definition **??**. But since most of the states in the wide regions of the tunnel are non-critical, this navigation domain satisfies Condition 4.1.

### D.2  CONDITION 4.2

Now we discuss the intuition behind why offline RL run on a form of noisy-expert data can outperform BC on expert data. As shown in Figure 5, the task is to navigate from the **Start** to the **Goal**. In this case, when BC is trained using only expert trajectories, and the environment consists of a stochastic dynamics, then running the BC policy during evaluation may diverge to low reward regions as shown in the second row, second column of Figure 5. On the other hand, if RL is provided with some noisy-expert data that visits states around expert trajectories which might eventually lead to low-rewarding states, then an effective offline RL method should be able to figure out how to avoid such states and solve the task successfully.

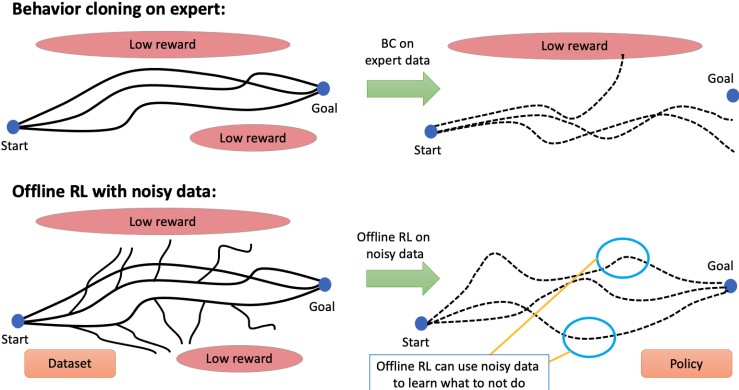

Figure 5:  (Figure 2 restated) **Illustration showing the intuition behind suboptimal data in a simple navigation task.** BC trained on expert data (data composition is shown on the left) may diverge away from the expert and find a poor policy that does not solve the task. On the other hand, if instead of expert data, offline RL is provided with noisy expert data that sometimes ventures away from the expert distribution, RL can use this data to learn to stay on the course to the goal.

Condition 4.2 requires that state-action tuples with $d^*(\mathbf{s}, \mathbf{a}) \geq b/H$ under the expert policy have high-enough density $\mu(\mathbf{s}, \mathbf{a}) \geq b$ under the data distribution. There are two ways to attain this condition: **(1)** the expert policy already sufficiently explores the state ($d^*(\mathbf{s}, \mathbf{a}) \geq b$), for example, when the environment is mostly deterministic or when trajectories are cyclic (e.g., in locomotion tasks discussed below), or, **(2)** the offline dataset is noisy such that states that are less-frequently visited by the expert are explored more in the data. The latter can be satisfied when the agent is provided with "negative data", i.e., failed trajectories, starting from states in an optimal trajectory as shown in Figure 5. Such counterfactual trajectories allow the agent to observe more outcomes starting from a state in the optimal trajectory. And running offline RL on them will enable the agent to learn what *not* to do, as illustrated in Figure 5. On the other hand, BC cannot use this negative data.

This condition is satisfied in the following practical problems:

- **Robotics**: In robotics noisy-expert data may not be directly available via demonstrations but can be obtained by a practitioner by running some limited autonomous data collection using noisy scripted policies (e.g., Kalashnikov et al. [21] and Kalashnikov et al. [23] run partly trained RL policies for noisy collection for running offline RL). Another simple way to obtain such data is to first run standard BC on the offline demonstrations, then store the rollouts obtained from the BC policy when evaluating it, and then run offline RL on the entire dataset (of expert demonstrations + suboptimal/noisy BC evaluation rollouts).

- **Autonomous driving and robotics:** In some domains such as autonomous driving and robotics, rolling out short counterfactual trajectories from states visited in the offline data, and labeling it with failures is a common strategy [5, 51]. A practitioner could therefore satisfy this condition by tuning the amount of counterfactual data they augment to training.

- **Recommender systems**: In recommender systems, Chen et al. [9] found that it was practical to add additional stochasticity while rolling out a policy for data collection which could be controlled, enabling the practitioner to satisfy this condition.

- **Trajectories consist of cyclic states (e.g., robot locomotion, character animation in computer graphics):** This condition can be satisfied in several applications such as locomotion (e.g., in character animation in computer graphics [46], or open-world robot navigation [57]) where the states observed by the agent are repeated over the course of the trajectory. Therefore a state with sufficient density under the optimal policy may appear enough times under $\mu$.

### D.3 Practical Guidelines For Verifying These Conditions

These conditions listed above and discussed in the paper can be difficult to check for in practice; for example, it is non-obvious how to quantitatively compute the volume of critical states, or how well-explored the dataset is. However, we believe that a practitioner who has sufficient domain-specific knowledge has enough intuition to qualitatively reason about whether the conditions hold. We believe that such practitioners can answer the following questions about their particular problem domain:

- Does there only exist a large fraction of states along a trajectory where either multiple good actions exist, or it is easy to recover from suboptimal actions?

- Is the offline dataset collected from a noisy-expert policy, and if not, can the dataset be augmented using perturbed or simulated trajectories?

If either of those questions can be answered positively, our theoretical and empirical results show that it is favorable to use offline RL algorithms, even over collecting expert data and using BC. At least, offline RL algorithms should be tried on the problem under consideration. Hence, the goal of our contributions is not to provide a rigid set of guidelines, but rather provide practical advice to the ML practitioner. We would like to highlight that such non-rigid guidelines exist in general machine learning, beyond RL. For example, in supervised learning, tuning the architecture of a deep neural network depends heavily on domain knowledge, and choosing kernel in a kernel machine again depends on the domain.

## E  Experimental Details

In this section we provide a detailed description of the various tasks used in this paper, and describe the data collection procedures for various tasks considered. We discuss the details of our tasks and empirical validation at the following website: https://sites.google.com/view/shouldirunrlorbc/home.

### E.1  Tabular Gridworld Domains

The gridworld domains we consider are described by $10 \times 10$ grids, with a start and goal state, and walls and lava placed in between. We consider a sparse reward where the agent earns a reward of $1$ upon reaching the goal state; however, if the agent reaches a lava state, then its reward is $0$ for the rest of the trajectory. The agent is able to move in either of the four direction (or choose to stay still); to introduce stochasticity in the transition dynamics, there is a $10\%$ chance that the agent travels in a different direction than commanded.

The exact three gridworlds we evaluate on vary in the number of critical points encountered per trajectory. We model critical states as holes in walls through which the agent must pass; if the agent chooses a wrong action at those states, it veers off into a lava state. The exact three gridworlds we evaluate on are: (a) "Single Critical" with one critical state per trajectory, (b) "Multiple Critical" with three critical states per trajectory, and (c) "Cliffwalk", where every state is critical [56]. The renderings of each gridworld are in Figure 6.

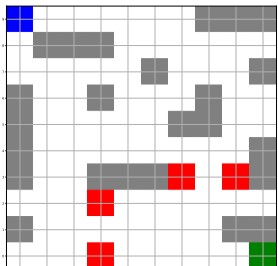 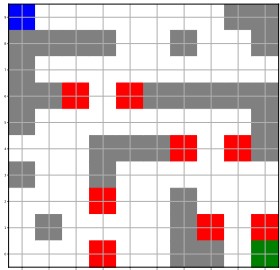 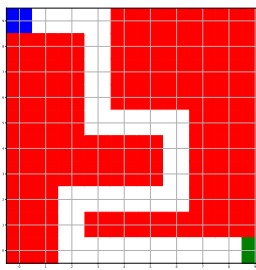

Figure 6: Renderings of three gridworld domains we evaluate on, where states are colored as: Start:blue, Goal:green, Lava:red, Wall:grey, and Open:white. The domains have varying number of critical points. *Left*: Single Critical. *Middle*: Multiple Critical. *Right*: Cliffwalk.

### E.2    MULTI-STAGE ROBOTIC MANIPULATION DOMAINS

**Overview of domains.** These tasks are taken from Singh et al. [60]. The robotic manipulation simulated domains comprise of a 6-DoF WidowX robot that interacts with objects in the environment. There are three tasks of interest, all of which involve a drawer and a tray. The objective of each task is to remove obstructions of the drawer, open the drawer, pick an object and place it in a tray. The obstructions of the drawer were varied giving rise to three different domains — **open-grasp** (no obstruction of the drawer), **close-open-grasp** (an open top drawer obstructs the bottom drawer), **pick-place-open-grasp** (an object obstructs the bottom drawer).

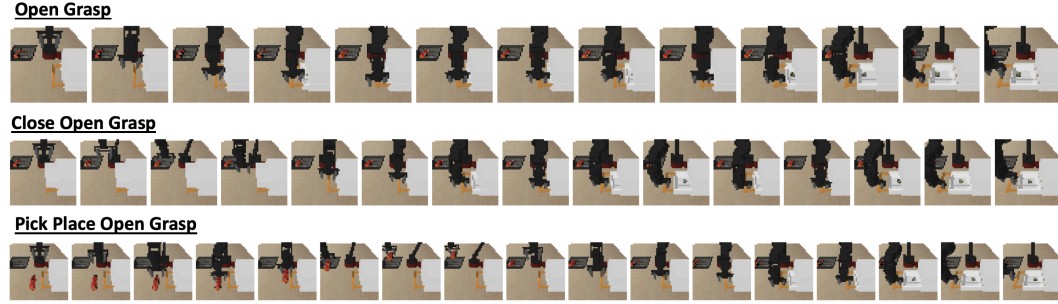

Figure 7: Filmstrip of the three tasks that we stufy for robotic manipulation – open-grasp, close-open-grasp and pick-place-open-grasp.

**Reward function.** For all the three tasks considered, a reward of $+1$ is provided when the robot is successfully able to open the drawer of interest (bottom drawer in close-open-grasp and pick-place-open-grasp; the only drawer in open-grasp) and is able to grasp the object inside it. If the robot fails at doing so, it gets no reward.

**Dataset composition.** For each task, we collected a dataset comprising of 5000 trajectories. For our experiments where we utilize expert data, we used the (nearly)-expert scripted policy for collecting trajectories and discarded the ones that failed to succeed. Thus the expert data attains a 100% success rate on this task. For our experiments with suboptimal data, which is used to train offline RL, we ran a noisy version of this near-expert scripted policy and collected 5000 trajectories. The average success rate in the suboptimal data is around 40-50% in both opening and closing the drawers with, 70% success rate in grasping objects, and a 70% success rate in place those objects at random locations in the workspace.

### E.3 ANTMAZE DOMAINS

**Overview of the domain.** This task is based on the antmaze-medium and antmaze-large environments from Fu et al. [14]. The goal in this environment is to train an 8-DoF quadraped ant robot to successfully navigate to a given, pre-specifcied target location in a maze. We consider two different maze layouts provided by Fu et al. [14]. We believe that this domain is well-suited to test BC and RL methods in the presence of multiple critical points, and is representative of real-world navigation scenarios.

**Scripted policies and datasets.** We utilize the scripted policies provided by Fu et al. [14] to generate two kinds of expert datasets: first, we generate trajectories that actually traverse the path from a given default start location to the target goal location that we consider for evaluation, and second, we generate trajectories that go from multiple random start positions in the maze to the target goal location in the maze. The latter has a wider coverage and a different initial state distribution compared to what we will test these algorithms on. We collected a dataset of 500k transitions, which was used by both BC and offline RL.

**Reward functions.** In this task, we consider a sparse binary reward $r(\mathbf{s}, \mathbf{a}) = +1$, if $|\mathbf{s}' - \mathbf{g}| \leq \varepsilon = 0.5$ and 0 otherwise. This reward is only provided at the end of a trajectory. This reward function is identical to the one reported by D4RL [14], but the dataset composition in our case comes from an expert policy.

### E.4 ADROIT DOMAINS

**Overview of the domain.** The Adroit domains [50, 14] involve controlling a 24-DoF simulated Shadow Hand robot tasked with hammering a nail (hammer), opening a door (door), twirling a pen (pen) or picking up and moving a ball (relocate). This domain presents itself with narrow data distributions, and we utilize the demonstrations provided by Rajeswaran et al. [50] as our expert dataset for this task. The environments were instantiated via D4RL, and we utilized the environments marked as: `hammer-human-longhorizon`, `door-human-longhorizon`, `pen-human-longhorizon` and `relocate-human-longhorizon` for evaluation.

**Reward functions.** We directly utilize the data from D4RL [14] for this task. However, we modify the reward function to be used for RL. While the D4RL adroit domains provide a dense reward function, with intermediate bonuses provided for various steps, we train offline RL using a binary reward function. To compute this binary reward function, we first extract the D4RL dataset for these tasks, and then modify the reward function as follows:

$$r(\mathbf{s}, \mathbf{a}) = +1 \quad \text{if } r_{\text{D4RL}}(\mathbf{s}, \mathbf{a}) \geq 70.0 \quad \text{(hammer-human)} \tag{16}$$

$$r(\mathbf{s}, \mathbf{a}) = +1 \quad \text{if } r_{\text{D4RL}}(\mathbf{s}, \mathbf{a}) \geq 9.0 \quad \text{(door-human)} \tag{17}$$

$$r(\mathbf{s}, \mathbf{a}) = +1 \quad \text{if } r_{\text{D4RL}}(\mathbf{s}, \mathbf{a}) \geq 47.0 \quad \text{(pen-human)} \tag{18}$$

$$r(\mathbf{s}, \mathbf{a}) = +1 \quad \text{if } r_{\text{D4RL}}(\mathbf{s}, \mathbf{a}) \geq 18.0 \quad \text{(relocate-human)} \tag{19}$$

The constant thresholds for various tasks are chosen in a way that only any transition that actually activates the flag `goal_achieved=True` flag in the D4RL Adroit environments attains a reward +1, while other transitions attain a reward 0. We evaluate the performance of various algorithms on this new sparse reward that we consider for our setting.

### E.5 ATARI DOMAINS

We utilized 7 Atari games which are commonly studied in prior work [27, 28]: ASTERIX, BREAKOUT, SEAQUEST, PONG, SpaceInvaders, Q*BERT, ENDURO for our experiments. We do not modify the Atari domains, directly utilize the sparse reward for RL training and operate in the stochastic Atari setting with sticky actions for our evaluations. For our experiments, we extracted datasets of different qualities from the DQN-Replay dataset provided by Agarwal et al. [3]. The DQN-Replay dataset is stored as 50 buffers consisting of sequentially stored data observed during training of an online DQN agent over the course of training.

**Expert data.** To obtain expert data for training BC and RL algorithms, we utilized all the data from buffer with id 49 (i.e., the last buffer stored). Since each buffer in DQN-Replay consists of 1M transition samples, all algorithms training on expert data learn from 1M samples.

| Domain / Behavior Policy | Task/Data Quality | BC | Naïve CQL | Tuned CQL |
|---|---|---|---|---|
| **7 Atari games (RL policy)** | Pong, Expert | $109.78 \pm 2.93$ | $102.03 \pm 4.43$ | $105.84 \pm 2.22$ |
| | Breakout, Expert | $75.59 \pm 21.59$ | $71.22 \pm 27.55$ | $94.77 \pm 27.02$ |
| | Asterix, Expert | $41.10 \pm 9.5$ | $44.81 \pm 12.0$ | $80.19 \pm 20.7$ |
| | SpaceInvaders, Expert | $40.88 \pm 4.17$ | $45.27 \pm 7.32$ | $54.15 \pm 2.96$ |
| | Q*bert, Expert | $121.48 \pm 9.06$ | $105.83 \pm 23.17$ | $98.52 \pm 18.62$ |
| | Enduro, Expert | $78.67 \pm 3.98$ | $141.53 \pm 18.79$ | $127.02 \pm 10.53$ |
| | Seaquest, Expert | $63.15 \pm 9.47$ | $64.03 \pm 27.67$ | $85.28 \pm 21.28$ |

Table 3: Per-game results for the Atari domains with expert data. Note that while naïve CQL does not perform much better than BC (it performs similarly as BC), tuned CQL with the addition of the DR3 regularizer performs much better.

| Task | BC-PI | CQL |
|---|---|---|
| Pong | $100.03 \pm 5.01$ | $94.48 \pm 8.39$ |
| Breakout | $25.99 \pm 1.98$ | $86.92 \pm 13.74$ |
| Asterix | $29.77 \pm 5.33$ | $157.54 \pm 37.94$ |
| SpaceInvaders | $31.45 \pm 1.96$ | $63.7 \pm 16.18$ |
| Q*bert | $106.06 \pm 8.63$ | $88.72 \pm 20.41$ |
| Enduro | $68.56 \pm 0.23$ | $148.97 \pm 12.3$ |
| Seaquest | $22.51 \pm 2.23$ | $124.95 \pm 43.86$ |

Table 4: Comparing the performance of BC-PI and offline RL on noisy-expert data. Observe that in general, offline RL significantly outperforms BC-PI.

**Noisy-expert data.** For obtaining noisy-expert data, analogous to the gridworld domains we study, we mix data from the optimal policy (buffer 49) with an equal amount of random exploration data drawn from the initial replay buffers in DQN replay (buffers 0-5). i.e. we utilize 0.5M samples form buffer 49 in addition to 0.5M samples sampled uniformly at random from the first 5 replay buffers.

## F  TUNING AND HYPERPARAMETERS

In this section, we discuss our tuning strategy for BC and CQL used in our experiments.

**Tuning CQL.** We tuned CQL offline, using recommendations from prior work [30]. We used default hyperparameters for the CQL algorithm (Q-function learning rate = 3e-4, policy learning rate = 1e-4), based on prior works that utilize these domains. Note that prior works do not use the kind of data distributions we use, and our expert datasets can be very different in composition compared to some of the other medium or diverse data used by prior work in these domains. In particular, with regards to the hyperaprameter $\alpha$ in CQL that trades off conservatism and the TD error objective, we used $\alpha = 0.1$ for all Atari games (following Kumar et al. [28]), and $\alpha = 1.0$ for the robotic manipulation domains following [60]. For the Antmaze and Adroit domains, we ran CQL training with multiple values of $\alpha \in \{0.01, 0.1, 0.5, 1.0, 5.0, 10.0, 20.0\}$, and then picked the smallest $\alpha$ that did not lead to eventually divergent Q-values (either positively or negatively) with more (1M) gradient steps. Next, we discuss how we regularized the Q-function training and performed policy selection on the various domains.

- **Detecting overfitting and underfitting**: Following Kumar et al. [30], as a first step, we detect whether the run is overfitting or underfitting, by checking the trend in Q-values. In our experiments, we found that Q-values learned on Adroit domains exhibited a decreasing trend throughout training, from which we concluded it was overfitting. On the Antmaze and Atari experiments, Q-values continued to increase and eventually stabilized, indicating that the run might be underfitting (but not overfitting).

- **Correcting for overfitting and policy selection**: As recommended, we applied a capacity decreasing regularizer to correct for overfitting, by utilizing dropout on every layer of the Q-function. We ran with three values of dropout parobability, $p \in \{0.1, 0.2, 0.4\}$, and found that $0.4$ was the most effective in alleviating the monotonically decreasing trend in Q-values, so used that for our results.

Then, we performed policy checkpoint selection by picking the earliest checkpoint that appears after the peak in the Q-values for our evaluation.

- **Correcting for underfitting:** In the Atari and Antmaze domains, we observed that the Q-values exhibited a stable, convergent trend and did not decrease with more training. Following Kumar et al. [30], we concluded that this resembled underfitting and utilized a capacity-increasing regularizer (DR3 regularizer [29]) for addressing this issue. We used identical hyperparameter for the multiplier ($\beta$) on this regularizer term for both Atari and Antmaze, $\beta = 0.03$ and did not tune it.

**Tuning BC.** In all domains, we tested BC with different network architectures. On the antmaze domain, we evaluated two feed-forward policy architectures of sizes $(256, 256, 256)$ and $(256, 256, 256, 256, 256, 256)$ and picked the one that performed best online. ON Adroit domains, we were not able to get a tanh-Gaussian policy, typically used in continuous control to work well, since it overfitted very quickly giving rise to worse-than-random performance and therefore, we switched to utilizing a Gaussian policy network with hidden layer sizes $(256, 256, 256, 256)$, and a learned, state-dependent standard deviation. To prevent overfitting in BC, we applied a strong dropout regularization of $p = 0.2$ after each layer for Adroit domains. On Atari and the manipulation domains, we utilized a Resnet architecture borrowed from IMPALA [10], but without any layer norm.

**Tuning BC-PI.** Our BC-PI method is implemented by training a Q-function via SARSA, i.e., $Q(\mathbf{s}, \mathbf{a}) \leftarrow r(\mathbf{s}, \mathbf{a}) + \gamma Q(\mathbf{s}', \mathbf{a}')$, where $(\mathbf{s}', \mathbf{a}')$ is the state-action pair that appears next in the trajectory after $(\mathbf{s}, \mathbf{a})$ in the dataset using the following Bellman error loss function to train $Q_\theta$:

$$\mathcal{L}(\theta) = \frac{1}{|\mathcal{D}|} \sum_{\mathbf{s}, \mathbf{a}, \mathbf{s}', \mathbf{a}' \sim \mathcal{D}} \left( Q_\theta(\mathbf{s}, \mathbf{a}) - (r(\mathbf{s}, \mathbf{a}) + \gamma \bar{Q}_{\bar{\theta}}(\mathbf{s}', \mathbf{a}'))\right)^2,$$

and then performing advantage-weighted policy extraction: $\pi(\mathbf{a}|\mathbf{s}) \propto \widehat{\pi_\beta}(\mathbf{a}|\mathbf{s}) \cdot \exp(A(\mathbf{s}, \mathbf{a})/\eta)$ on the offline dataset $\mathcal{D}$. The loss function for this policy extraction step, following Peng et al. [47] is given by:

$$\pi_\phi \leftarrow \max_{\pi_\phi} \sum_{\mathbf{s}, \mathbf{a}} \log \pi_\phi(\mathbf{a}|\mathbf{s}) \cdot \exp\left( \frac{Q_\theta(\mathbf{s}, \mathbf{a}) - V(\mathbf{s})}{\eta} \right),$$

where the value function was given by $V(\mathbf{s}) = \sum_{\mathbf{a}'} \pi_\beta(\mathbf{a}'|\mathbf{s}) Q_\theta(\mathbf{s}, \mathbf{a}')$ and $\pi_\beta$ is a learned model of the behavior policy, as done in the implementation of Brandfonbrener et al. [6]. This model of the behavior policy is trained according to the tuning protocol for BC, and is hence well-tuned.

**What we tuned:** We tuned the temperature hyperparameter $\eta$ using multiple values spanning various levels of magnitude: $\{0.005, 0.05, 0.1, 0.5, 1.0, 3.0\}$ and additionally tried two different clippings of the advantage values $A(\mathbf{s}, \mathbf{a}) := Q(\mathbf{s}, \mathbf{a}) - V(\mathbf{s})$ between $[-10, 2]$ and $[-10, 4]$. The Q-function architecture is identical to tuned CQL, and the policy and the model of the behavior policy both utilize the architecture used by our BC baseline.

**Our observations:** We summarize our observations below:

- We found that in all the runs the temporal difference (TD) error for SARSA was in the range of $[0.001, 0.003]$, indicating that the SARSA Q-function is well behaved.

- The optimal hyperparameters that lead to the highest average performance across all games is $\eta = 0.005$, and the advantage clipping between $[-10, 2]$. We find a huge variation in the performance of a given $\eta$, but we used a single hyperparameter $\eta$ across all games, in accordance with the Atari evaluation protocols [40], and as we did for all other baselines. For example, while on some games such as Qbert, BC-PI improves quite a lot and attains 118% normalized return, on Enduro it attains only 78% and on SpaceInvaders it attains 31.7%.

- These results perhaps indicate the need for per-state tuning of $\eta$, i.e., utilizing $\eta(\mathbf{s})$, however, this is not covered in our definition of BC-PI from Section 4.4, and so we chose to utilize a single $\eta$ across all states. Additionally, we are unaware of any work that utilizes per-state $\eta(\mathbf{s})$ values for exponentiated-advantage weighted policy extraction.

# G    REGARDING OFFLINE RL LOWER BOUNDS

Here we address the connection between recently published lower-bounds for offline RL by Zanette [71] and Wang et al. [64] and our work.

Zanette [71] worst-case analysis does address function approximation even under realizability and closure. However, this analysis concerns policy evaluation and applies to policies that go out of the support of the offline data. This does not necessarily prove that any conservative offline RL algorithm (i.e., one that prevents the learned policy from going out of the support of the offline dataset) would suffer from this issue during policy learning, just that there exist policies for which evaluation would be exponentially bad. Offline RL can exactly prevent the policy from going out of the support of the dataset in Zanette [71]'s counterexample since the dynamics and reward functions are deterministic. This policy can still improve over behavior cloning by stitching overlapping trajectories, similar to the discussion of one-step vs multi-step PI in Theorem 4.4. Thus, while the lower-bound applies to off-policy evaluation of some target policies, it doesn't apply to evaluation of *every* policy in the MDP, and specifically **does not** apply to in-support policies, which modern offline RL algorithms produce, in theory and practice. Certainly, their lower bound would apply to an algorithm aiming to estimate $Q^*$ from the offline data, but that's not the goal of pessimistic algorithms that we study. Finally, our practical results indicates that offline RL methods can be made to perform well in practice despite the possibility of any such lower bound.

The lower bound in Wang et al. [65] only applies to non-pessimistic algorithms that do not tackle distributional shift. In fact, Section 5, Theorem 5.1 in Wang et al. [64] provides an upper bound for the performance of offline policy evaluation with low distribution shift. The algorithms analyzed in our paper are pessimistic and guarantee low distribution shift between the learned policy and the dataset, and so the lower bound does not apply to the algorithms we analyze.

# H    DIAGNOSTIC EXPERIMENTS ON A GRIDWORLD

We first evaluate tabular versions of the BC and offline RL methods analyzed in Section 4.1 on sparse-reward $10 \times 10$ gridworlds environments [13]. Complete details about the setup can be found in Appendix E.1. On a high-level, we consider three different environments, each with varying number of critical states, from "Single Critical" with exactly one, to "Cliffwalk" where every state is critical and veering off yields zero reward. The methods we consider are: naive BC (BC), conservative RL (RL-C), policy-constraint RL (RL-PC), and generalized BC with one-step and k-step policy improvement (BC-PI, BC-kPI). In the left plot of Figure 8, we show the return (normalized by return of the optimal policy) across all the different environments for optimal data ($C^* = 1$) and data generated from the optimal policy but with a different initial state distribution ($C^* > 1$ but $\pi_\beta(\cdot|\mathbf{s}) = \pi^*(\cdot|\mathbf{s})$). As expected from our discussion in Section 4.2, BC performs best under $C^* = 1$, but RL-C and RL-PC performs much better when $C^* > 1$; also BC with one-step policy improvement outperforms naive BC for $C^* > 1$, but does not beat full offline RL. In Figure 8, we vary $C^*$ by interpolating the dataset with one generated by a random policy, where $\alpha$ is the proportion of random data. RL performs much better over all BC methods, when the data supporting our analysis in Section 4.3. Finally, BC with multiple policy improvement steps performs better than one step when the data is noisy, which validates Theorem 4.4.

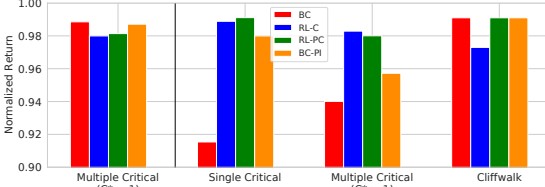
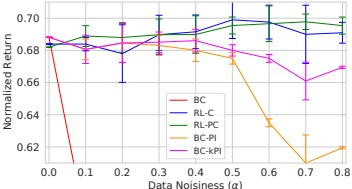

Figure 8: **Offline RL vs BC on gridworld domains.** *Left*: We compare offline RL and BC on three different gridworlds with varying number of critical points for expert and near-expert data. *Right*: Taking the "Multiple Critical" domain, we examine the effect of increasing the noisiness of the dataset by interpolating it with one generated by a random policy, and show that RL improves drastically with increased noise over BC.

