# OpenReview forum: "Should I Run Offline Reinforcement Learning or Behavioral Cloning?"
_ICLR.cc/2022/Conference — ICLR 2022 Poster_

### Official Review · Reviewer_4pq1 · 2021-11-01

**Correctness:** 4
**Technical Novelty And Significance:** 4
**Empirical Novelty And Significance:** 3
**Recommendation:** 8
**Confidence:** 3

**Main Review:**

### Pros
- The quality and clarity of the writing are remarkable. The assumptions are clearly stated and transparent to the reader. The discussion around the theoretical results is well conducted and the insights are informative to the reader.
- The empirical evaluation on a tabular gridworld domain is appreciated as it removes the influence of function approximators on the results. The 3 types of girdworld instances provide the right settings to confront the theoretical results of section 4.
    - On a side note, I appreciate the care of using the suggestions introduced in [1] to report the empirical results of RL experiments.
    - I also appreciate the fact that the authors didn't shy away from highlighting potential discrepancies in their results due to the various tricks introduced in the literature to tune offline RL algorithms.

### Questions
- I would ask the authors to clarify the need for condition 3.2 as it seems that some of their results could be derived without it, e.g. theorem 4.1. How would the results be affected if we were to remove this condition?

### Minor Comments
- The space taken by the discussion around the theoretical results doesn't leave much space for the analysis of the experiments. As a reader, it is hard to digest and understand whether the results fully align with the theory.
- Add an explanation of the terms for theorem 4.2, like it is done for theorem 4.1.
1. Rishabh Agarwal, Max Schwarzer, Pablo Samuel Castro, Aaron Courville, and Marc G Bellemare. Deep reinforcement learning at the edge of the statistical precipice. *Advances in Neural Information Processing Systems*, 2021b.

**Summary Of The Paper:**

The paper provides an attempt to help practitioners answer the question "For what type of environments and datasets should we prefer Offline RL over Behavioural Cloning". To do so, the authors extend the previous work of [1] studying this problem for contextual bandits to MDPs. These theoretical results enable them to draw some conclusions on when we can expect Offline RL to have an edge on Behavioural cloning. To name a few:

- When the dataset provided is not optimal, offline RL can scale more favourable with the horizon especially for environments with horizon-independent returns or with a low volume of critical states,i.e. states for which there is no significant advantage to select one action over another.
- For long horizon tasks again, offline RL trained on noisy data displaying higher coverage can outperform BC trained on the same amount of expert demonstrations.

These theoretically grounded insights are then empirically validated on a tabular gridworld domain, after which they validate their findings on high dimensional offline RL problems (continuous control and navigation, and some atari games). On the tabular gridworld, the experiments strongly agree with their theoretical results. When turning to deep RL, the experiments mostly aligned although the various tuning mechanisms introduced in previous work to train efficiently offline RL algorithms introduce a few discrepancies.

1. Paria Rashidinejad, Banghua Zhu, Cong Ma, Jiantao Jiao, and Stuart Russell. Bridging offline reinforcement learning and imitation learning: A tale of pessimism. *arXiv preprint arXiv:2103.12021*,

**Summary Of The Review:**

Overall, I consider this paper to be a great contribution to the research community. The question investigated is important for the research in offline RL and practitioners. The theoretical analysis is well conducted and meaningful insights can be drawn from it. The experimental results are adequately reported and support the findings.

---

> ### Author Response · Authors · 2021-11-18
> **Author Response**
>
> We would like to thank the reviewer for the detailed review, and for a positive assessment of the paper. We would like address comments that the reviewer had below:
>
> > How would the results be affected if we were to remove [Condition 3.2]?
>
> Our bounds for BC and offline RL would be affected as follows. For BC, we use Condition 3.2 to bound the worst-case suboptimality by a constant 1, rather than H that is done in prior works. This is why our bound saves a O(H) factor. For offline RL, our usage of Condition 3.2 is more nuanced. In particular, we use Condition 3.2 in conjunction with a variance recursion technique similarly used in [Ren et al. 2021](https://arxiv.org/pdf/2103.14077.pdf) and [Zhang et al., 2021](https://arxiv.org/pdf/2009.13503.pdf) to shave off a O(H^2) factor from existing analysis of a similar VI-LCB algorithm ([Rashidinejad et al. 2021](https://arxiv.org/pdf/2103.12021.pdf)). This is important as we are able to derive matching guarantees for BC and offline RL, whereas prior work often had offline RL bounds with worse scaling in horizon H. However, we believe that Condition 3.2 is only necessary because of our analysis techniques and perhaps can be relaxed with more refined analysis.
>
>
> > Minor comments.
>
> We agree with the reviewer that those particular aspects of the paper (interpretation of empirical results, and of the bounds in each theorem) should have a more detailed discussion. In the revised version of our paper, we have reorganized the content in the paper to make the practical takeaways of our work more clear which is reorganized in terms of Practical Insights 4.1-4.4.

---

### Official Review · Reviewer_b23W · 2021-11-01

**Correctness:** 3
**Technical Novelty And Significance:** 2
**Empirical Novelty And Significance:** 2
**Recommendation:** 6
**Confidence:** 3

**Main Review:**

Strengths
1. The paper studies an interesting problem: when an offline RL algorithm has advantages over a simple BC algorithm. I think the problem is in the interests of many RL practitioners.
2. The paper provides a good overview on related works.

Weaknesses (and questions)
1.	The paper proposes several sufficient conditions that VI-LCB could have a better worst-case sub-optimality gap. However, it does not provide practitioners a clear guidance on when they should run BC and when they should run VI-LCB, which I think is one of the main purpose of the paper. How should we check if the sufficient conditions are satisfied in real world problems? Some of the parameters are even unknown (e.g., b in Corollary 4.2).
2.	I find it difficult to compare these upper bounds. Is it possible that some algorithms get larger bounds simply because the analysis is not tight?  Can we say that algorithm A is better than algorithm B because an upper bound (of the sup-optimality gap) for A is lower than an upper bound for B?
3.	Some conditions are not well-justified. The paper studies some sufficient conditions that VI-LCB outperforms BC, however, some conditions seems a little arbitrary to me (e.g., condition 3.2, condition 4.1 and condition 4.2). I guess the paper has some specific applications in mind and makes these conditions based on these applications. However, this makes the results less clear and less general. For example, how do the results extend without condition 3.2? do we expect to just see an extra H term in all bounds without the condition? If the paper considers some specific applications, the paper should clearly mention it instead of saying the conclusion holds in general.
4.	The paper studies a particular type of offline RL algorithm based on VI with LCB style penalty. However, the title and the paper seems to suggest that this particular algorithm can represent the class of offline RL algorithms? I don’t see how study one algorithm can generalize to a conclusion for offline RL algorithms?
5.	I am not sure what is the novelty and significance of the theoretical results. It seems the theoretical results are mostly borrowed from existing literature with some modifications. I want to make sure I am not missing something important there. Can you explain more about the novelty or significance for the theoretical results?

Minor questions and comments:
1. In the empirical section, does CQL lie in the VI-LCB framework? If not, why is it used in the empirical section while all theoretical results are for VI-LCB.
2. VI-LCB with Bernstein style penalty is also used in [1], which is not cited in this paper.

[1] Xie et al. (2021) “Policy Finetuning: Bridging Sample-Efficient Offline and Online Reinforcement Learning”


**Summary Of The Paper:**

The paper studies the sufficient conditions that VI-LCB would have a better worst-case sub-optimality gap than the gap of a BC algorithm, under expert data and noisy data. Moreover, the paper provide an empirical study to validate the theoretical results.

**Summary Of The Review:**

I am leaning towards a recommendation to reject the paper since I think the paper does not clearly answer the core question: whether should I run BC or offline RL for several reasons mentioned in the Main Review section. However, I am open to change my scores after author response.

----
After the author response, most of my concerns were addressed. I think the paper provides good empirical and theoretical understandings of the comparison between BC and pessimistic VI, so I raised my score to a 6.

---

> ### Author Response · Authors · 2021-11-12
> **Author Response (Part 1 of 3): Addressing the main points: our goal, answering the core question**
>
> Thank you for your constructive feedback. *We have updated the paper (in red) to address various concerns about core takeaways, novelty, contributions, and practitioner guidelines. We have toned down any claims that might appear over-general in the revised paper. Please let us know if your concerns are addressed, we are happy to address any remaining concerns.* First, we would like to clarify what the main goal of our work is, how the paper attempts to answer the core question, what the practical takeaways of our paper are, and how we have revised the paper to make these more explicit. **Please let us know** if this addresses your concerns about the “core question” and your comment regarding “sufficient conditions that VI-LCB could have a better worst-case sub-optimality gap”.
>
> **Our Goal:** We would like to stress that our goal is not to theoretically improve existing bounds in offline RL, nor to simply analyze VI-LCB under a new set of conditions. We are not just trying to show sufficient conditions that VI-LCB could have a better worst-case sub-optimality gap. The goal of our work is to answer: **“under what environment and dataset conditions can offline RL outperform BC with an equal amount of expert data, even when BC is a natural choice?”** Since a number of recent works claim that BC should be a viable substitute for offline RL in various cases (see, [Mandlekar et al. CoRL 2021](https://openreview.net/forum?id=JrsfBJtDFdI), [Florence et al. CoRL 2021](https://openreview.net/forum?id=rif3a5NAxU6), [Hahn et al. NeurIPS 2021](https://arxiv.org/abs/2110.09470) for empirical studies), we believe that it is of interest to the community to describe situations where the opposite is expected to be true.
>
> **Answering the core question:** We have improved the exposition of Abstract, Section 1, and the technical sections in the revised paper to make it clear how the paper answers the core question. We show that a strong offline RL algorithm (e.g., VI-LCB used in our analysis) can outperform running BC on an equal amount of expert data in environments under certain conditions, and then discuss how these conditions are quite reasonable in a variety of applications. We empirically verify that the takeaways (now shown in blue boxes in Section 4: **Practical Insights 4.1-4.4** in the revised paper) apply in practice on modern deep offline RL algorithms, and show how existing tuning procedures can enable practitioners to use offline RL in a way that already outperforms BC on expert or near-expert data. We believe this conclusion is both novel and significant. To highlight the practical takeaways more explicitly, we now present these implications in boxes (**Practical Insights 4.1-4.4**), along with intuition to verify these guidelines in practice in **Appendix D**.
>
> We thank you for bringing the Xie et al. 2021 paper to our notice that we were not aware of. We have now added a discussion of it in the related work section.
>
> ___
>
> We now answer the individual points raised by the reviewer.
>
> > **I find it difficult to compare these upper bounds.**
>
> We would like to clarify that in all major theoretical results we do not compare one upper bound on suboptimality against another.  In Corollary 4.1 and 4.2, we compare the worst-case performance for RL ("upper bound’") to the best possible performance of BC (i.e., the information-theoretic lower bound for BC) on expert data presented in Theorem 4.3.
> And in Theorem 4.4, where we compare the impact of multiple steps of policy improvement performance, we again do not compare two upper bounds, but rather explicitly bound the performance difference between multiple steps and one step of policy improvement (i.e., $J(\widehat{\pi}^k) - J(\widehat{\pi}^1) \geq \text{our bound in Theorem 4.4}$).
> Thus, we believe we avoid comparing upper bounds on suboptimality for any two algorithms in our analysis, thus avoiding this issue.

---

> > ### Author Response · Authors · 2021-11-12
> > **Author Response (Part 2 of 3): Conditions we study and practical implications**
> >
> > > **Some of the conditions are not well-justified.**
> >
> > We agree that the conditions we list may not hold generally in all problem domains and we have toned down the claims of the generality of these conditions in the revised version of the paper both in the introduction as well as the motivation. We now explicitly and prominently discuss applications where our analysis is the most relevant in **Section 4** and **Appendix D**.  ***We are also happy to additionally change the title of the paper to reduce indications of over-generality if the reviewer sees fit.**
> >
> > We discuss some of the individual conditions one by one:
> >
> > - **Condition 3.2:** One very common setting that satisfies this condition is when the task has a sparse reward at the end indicating success or failure. This is very common in domains such as robotics or navigation, where the agent might receive a signal upon successful completion of the task (e.g.,  Kalashnikov et al. 2018, 2020).
> >
> > - **Condition 4.1:** We discussed some practical scenarios where Condition 4.1 holds in the paper. We’ve further revised the paper to expand this discussion and added a figure **Figure 3** in **Appendix D** to build intuition illustratively. Intuitively, Condition 4.1 holds in states where multiple good actions exist. We give two example domains: **(1)** robotic manipulation and **(2)** navigation. In manipulation tasks such as grasping, if the robot is not near the object, it can take many different actions while still picking up the object in the end; this is because unless the object breaks, actions taken by the robot are typically reversible allowing for recovery from suboptimal actions. There are only a few “critical states” right at the time when the robot grasps the object, where the robot should be careful to not drop the object such that it breaks. In navigation, there may exist multiple paths that end at the same goal, particularly in large, unobstructed areas. For example, while navigating through a wide tunnel, the exact direction the agent takes may not matter so much as multiple directions will take the agent out of the tunnel. There are a few “critical states” (e.g., doorways) where taking a specific action is important, but most states are not critical points.
> >
> > - **Condition 4.2:** First we would like to note that the condition of a softmax-optimal expert demonstrator is a standard assumption in [Maxent inverse reinforcement learning work](https://www.aaai.org/Papers/AAAI/2008/AAAI08-227.pdf). Second, while we agree that in practice, not all datasets will be noisy-expert, however, we argue that in several real-world applications, collecting noisy-expert data is not a severe bottleneck. In many robotics applications, it is practical to augment expert demonstrations using counterfactual actions or by running autonomous scripted policies, which large-scale systems such as [QT-Opt](https://arxiv.org/abs/1806.10293) and [MT-Opt](https://arxiv.org/abs/2104.08212) do. These autonomous policies may still fail, and attain poor reward, but are not so expensive to collect, and will allow us to improve performance better than just cloning the expert demonstrations because they increase coverage. Additionally in self-driving, as shown in Bojarski et al. 2016, it is beneficial to perturb the vehicle location to create new trajectories for RL, which may be non-expert but can help offline RL by increasing coverage (Condition 4.2). We also demonstrate an intuitive example for this intuition in **Appendix D, Figure 4**.
> >
> > ___
> >
> >
> > > **How should we check if the sufficient conditions are satisfied in real-world problems?**
> >
> > We have added a discussion of this in **Appendix D**. We agree that the conditions listed earlier can be difficult to exactly quantify. However, we believe that an ML practitioner who has sufficient domain-specific knowledge has enough intuition to qualitatively reason about whether the conditions hold. We believe that such practitioners can answer the following questions about their particular problem domain:
> >
> > - Does there only exist a large fraction of states along a trajectory where either multiple good actions exist, or it is easy to recover from suboptimal actions?
> >
> > - Is the offline dataset collected from a noisy expert, and if not, can the dataset be augmented using perturbed or simulated trajectories?
> >
> > If either of those questions can be answered positively, our theoretical and empirical results show that it is favorable to use offline RL algorithms, even over collecting expert data and using BC. Hence, the goal of our theoretical contributions is not to provide a rigid set of guidelines, but rather provide practical advice to the ML practitioner. We would like to highlight that such non-rigid guidelines exist in general machine learning, beyond RL. For example, in supervised learning, tuning the architecture of a deep neural network depends heavily on domain knowledge, and choosing a kernel in a kernel machine again depends on the domain.

---

> > > ### Author Response · Authors · 2021-11-12
> > > **Author Response (Part 3 of 3): Novelty, Do conclusions apply to other algorithms**
> > >
> > > > **Novelty concerns**
> > >
> > > **Theory:** We have now updated the paper to clearly highlight the novel contributions in theory and practice (changes in red in Section 4). While the algorithms we analyze exist in prior work, our work does not simply compare bounds from these prior works. We highlight the novel components of the main theoretical results below:
> > >
> > > - **Theorem 4.2:** Our analysis in Theorem 4.2 improves over the existing analysis in Rashidinejad et al. 2021 by using a Bernstein bonus and the recursion technique to obtain an $\mathcal{O}(\sqrt{H})$ bound compared to $\mathcal{O}(H^{2.5})$, and avoids an additional factor of $|\mathcal{S}|$. These improvements are key in allowing us to show that offline RL can perform better than BC; the bounds in Rashidinejad et al. 2021 directly do not show this.
> > >
> > > - **Corollary  4.2:** We compare the performance of BC with expert/near-expert data to RL with noisy data and show that for an equal amount of data, RL can do better. Such a comparison is not present in prior works, and is perhaps also surprising, and practically informative for a practitioner, especially when they are deciding what data to collect to attain the best performance on their problem. Just comparing the upper bounds from BC and offline RL from prior works naively would in fact indicate the opposite, meaning that our analysis is non-trivial. We have added the practical takeaway for this in Practical Insight 4.3.
> > >
> > > - **Theorem 4.4:** We utilize tools from prior work to prove this theorem, but we characterize the performance gain obtained by considering multiple rounds of policy improvement compared to one round of policy improvement. We are unaware of any prior work showing this benefit in the context of offline RL, and in fact, recent work has shown conflicting opinions (see discussion in [Kostrikov et al. 2021](https://arxiv.org/abs/2110.06169), Section 5.1 comparing to Brandfonbrener et al. 2021).
> > >
> > > In addition, to our knowledge, the analysis of algorithms under the practically-relevant condition of critical states in Corollary 4.1 is novel.
> > >
> > > **Empirical Results:** We have now updated the paper to clearly highlight the novel takeaways (Practical Insights 4.1-4.4) and related them to our experiments. While the specific algorithms used for offline learning and offline tuning are not novel (since we are not trying to propose a new algorithm), to our knowledge, no prior work shows that running some kind of offline RL (either on noisy data or expert data) can outperform cloning an expert dataset for the same task, given equal amounts of data. In fact, results in D4RL would indicate the opposite (compare BC with expert data vs the best offline RL algorithm for medium-expert data in Table 2 in the D4RL paper). The fact that offline tuning procedures applied to offline RL algorithms on noisy-expert datasets allow us to outperform BC is a conclusion that, to the best of our knowledge, has not been demonstrated before.
> > >
> > > ___
> > >
> > > > **I don’t see how study one algorithm can generalize to a conclusion for offline RL algorithms?**
> > >
> > > The goal of our paper is to examine if, under reasonable conditions, there exist offline RL algorithms that can outperform directly cloning the expert data for a given task. Since we just want to show that it is possible, we consider representative algorithms for offline RL and BC that obtain good theoretical or empirical performance. That doesn’t mean other offline RL algorithms don’t enjoy these benefits, as our aim is simply to provide a proof of existence. On the theory side, we study a modified version of VI-LCB (Rashidinejad et al. 2021) that uses the Bernstein-style penalty to achieve better suboptimality guarantees. This algorithm serves as a representative method from the class of approaches that penalize value functions (e.g., CQL (Kumar et al. 2020), BRAC (Wu et al. 2019)). There does exist a theory-practice gap between generalizing VI-LCB to neural networks, but we do use an algorithm from the same family (i.e., CQL) for our experiments, which works well. Both algorithms are similar in that they both estimate a pessimistic / conservative / lower-bound value function. Given our objective, we believe it is appropriate to use the best-known algorithm of each class.
> > > **We have clarified this objective in the revised paper by revising any unclear claims that might have appeared overly general and if the reviewer still finds the title of the paper misleading, we are willing to change the title if there are any specific suggestions.**

---

> > > > ### Comment · Reviewer_b23W · 2021-11-18
> > > > **Response to authors**
> > > >
> > > > Thanks for the detailed response. I am generally happy with the revised paper which does not seem to make unclear and over-generalizing claims. I think it might also be a good idea to change the title to make it less misleading (this is completely up to the authors to decide, and won't change my evaluation).  In general, I think the paper provides good empirical and theoretical understandings of the comparison between BC and pessimistic VI, so I will raise my score accordingly.
> > > >
> > > > I do have some questions:
> > > > 1. how do the results extend without condition 3.2. The dependence on the horizon plays an important role in the paper so I am wondering if condition 3.2 hides some negative results.
> > > > 2. In Corollary 4.2, it seems like $b$ has to be $H/N$ (up to a constant scaling) to get the upper bound to be $\tilde O(\sqrt{H})$. Otherwise, if $b$ is just a constant, we get $O(H/N)$. Do I miss something here? If not, what does $b=H/N$ mean? Is it a practical assumption?

---

> > > > > ### Author Response · Authors · 2021-11-22
> > > > > **Author Response: Thanks! Answering the Questions**
> > > > >
> > > > > Thank you for the reply, and we are glad that we addressed your concerns. We would be grateful if you could raise your score to reflect our discussion.
> > > > >
> > > > > We have updated the title to *“When Should Offline Reinforcement Learning be Preferred Over Behavioral Cloning?”* to better reflect the contributions of the paper, and are happy to revise it further on the reviewers’ suggestions.
> > > > >
> > > > > We answer your questions below:
> > > > >
> > > > > > **How do the results extend without condition 3.2?**
> > > > >
> > > > > For BC, we use Condition 3.2 to bound the worst-case suboptimality by a constant 1, rather than $H$ that is done in prior works. This is why our bound saves a $O(H)$ factor. Without this condition, we would obtain an additional $H$ factor. For offline RL, our usage of Condition 3.2 is more nuanced. In particular, we use Condition 3.2 in conjunction with Bernstein bonuses and a variance recursion technique (e.g., [Ren et al. 2021](https://arxiv.org/pdf/2103.14077.pdf)) to shave off a $O(H^2)$ factor from the existing analysis of the VI-LCB algorithm ([Rashidinejad et al. 2021](https://arxiv.org/pdf/2103.12021.pdf)). This is important as we are able to derive matching/better guarantees for offline RL compared to BC, whereas prior work often had offline RL bounds with worse scaling in horizon $H$. It might be possible to derive similar guarantees without needing Condition 3.2, perhaps with a relaxed analysis, and we are not aware of any information-theoretic lower-bound that prohibits an offline RL algorithm with such guarantees without needing Condition 3.2.
> > > > >
> > > > > That said, condition 3.2 is already satisfied in many practical problems (e.g., sparse reward problems) which we considered as motivating examples in the paper.
> > > > >
> > > > > ___
> > > > >
> > > > > > **Questions about Corollary 4.2.**
> > > > >
> > > > > This is a practical condition and we have added more discussion of this in **Appendix D.2** (in magenta). Theoretically, you are correct that if b were just a constant, the bound for offline RL would not improve upon $O(H)$. For generality, we state our result in terms of variable b, which can take on any values in $[\log H/N, 1)$. However, this bound also holds when b satisfies  $b = O(H^{\alpha} / N)$ for some constant $\alpha < 1$. We only listed one such b (in fact, $b = O(\sqrt{H} / N)$ would also lead to $O(\sqrt{H})$ suboptimality). Further, in contrast to prior theoretical works such as [Ren et al. 2021](https://arxiv.org/pdf/2103.14077.pdf), this assumption only applies to states visited by the optimal policy and is, therefore, weaker than assuming coverage over all states.
> > > > >
> > > > > **Practically**, we list several cases where this condition is easy to satisfy in **Appendix D.2** (in magenta) in the revision. Briefly, some examples of where this is satisfied are:
> > > > >
> > > > > 1. In several instances in robotics and navigation domains, it is possible to simulate counterfactual data (e.g., [Rao et al. 2020](https://arxiv.org/abs/2006.09001)) and use this data as failed data. This data augmentation allows a practitioner to tune how well-explored the offline dataset should be. One simple way to satisfy this condition is to roll out short counterfactual trajectories from the infrequent states visited by an expert. Similarly, in the domain of recommender systems, [Chen et al. 2019](https://arxiv.org/abs/1812.02353) found that it was practical to add additional stochasticity while rolling out a policy for data collection and this can provide a way for the practitioner to satisfy this condition.
> > > > >
> > > > > 2. In many locomotion applications (e.g., [Shah et al 2021](https://arxiv.org/abs/2012.09812)), the trajectory of states taken by the robot exhibits some cyclicity; for example, a robot learning to walk will repeat the same movements when taking each step. There, a particular state can appear multiple times in a trajectory, and the states covered by the optimal policy attain a density of more than $H^{\alpha}/N$.
> > > > >
> > > > > ___
> > > > >
> > > > > *Please let us know if this answers your questions.*

---

### Official Review · Reviewer_9b7d · 2021-11-02

**Correctness:** 3
**Technical Novelty And Significance:** 3
**Empirical Novelty And Significance:** 3
**Recommendation:** 8
**Confidence:** 5

**Main Review:**

**Strengths**

Offline RL approaches are of quite great interest because of potentially easier real world applications. But as the paper points out in the extensive related work, there are a lot of conflicting results in the literature when it comes to comparison with plain old behavior cloning. To this end, the paper functions as fairly clear exposition of the current state of offline RL literature and proposes a formalization to allow comparison between these approaches. This formalization leads to a better characterization for the conditions for offline RL methods to outperform behavior cloning (and the importance of planning horizon and critical states).

The paper performs experiments on a wide number of domains which allows the readers to make their own inferences for the conditions for offline RL successes.

**Weaknesses**

It's unclear how much bearing the theory has on the actual empirical experiments as the authors themselves emphasize the importance of tuning offline RL. Tuning techniques seem domain dependent and exactly what features of the domain lead to particular tuning choices remains unexplored. Nevertheless, since the tuning methods are offline, it's not a major issue. Overall, some way to separate the [state] representation learning problem from the best action problem is likely important to really figure out the problem which the theory in the paper largely ignores.

The bigger issue is whether the "noisy-expert" data is a good proxy for real-world offline RL applications. While the paper cites a few examples of suboptimal data collection in Sec. 4.3 for most real world applications, the data is actually seldom noisy in the same way. Often a more realisitic assumption is that there are different suboptimal demonstrators (consider different human operators with different skill levels) who perform the task in different ways but their policies can't be simply described as noisy versions of each other (or the highest skill level). Moreover, there are often many safety constraints that simply would not allow running certain kind of behavior policies. Authors might find [1] an interesting read. Adroit results in Table 1 kind of point towards something similar (BC and CQL are not that different in performance).

[1] Vladislav Kurenkov, Sergey Kolesnikov. "Showing Your Offline Reinforcement Learning Work: Online Evaluation Budget Matters". AAAI 2022.


**Summary Of The Paper:**

Offline RL approaches are of quite great interest because of potentially easier real world applications. But as the paper points out in the extensive related work, there are a lot of conflicting results in the literature when it comes to comparison with plain old behavior cloning. To this end, the paper functions as fairly clear exposition of the current state of offline RL literature and proposes a formalization to allow comparison between these approaches. This formalization leads to a better characterization for the conditions for offline RL methods to outperform behavior cloning (and the importance of planning horizon and critical states).

The paper performs experiments on a wide number of domains which allows the readers to make their own inferences for the conditions for offline RL successes.


**Summary Of The Review:**

While some claims about offline RL doing better than BC as "surprising" are not necessarily something I agree with, overall it's an interesting paper and allows the community to make progress towards better understanding of the tradeoffs between different policy learning mechanisms and would recommend acceptance.

---

> ### Author Response · Authors · 2021-11-18
> **Author Response**
>
> Thank you for your constructive feedback and a positive assessment of our work. Thank you for pointing us to the interesting paper (Kurenkov and Kolesnikov, 2022), which we have now cited and discussed in our submission. We have updated the paper to discuss some examples of cases where noisy-expert data can be obtained in the real-world or where noisy-expert data can be easily collected, and present some examples below.
>
> ___
>
>
> > **whether noisy-expert data is a good proxy for real-world applications**
>
> We agree that this is an important question and extending our analysis to a mixture of suboptimal experts is interesting for future work. Additionally, the reviewer’s suggestion of handling safety constraints is also important. That said, we list some domains, where either **(1)** noisy-expert data is not an unreasonable assumption, or **(2)** when the demonstrations are not noisy, collecting noisy-expert data is not a severe bottleneck.
>
> A number of works in MaEnt IRL literature assume that the demonstrator is not exactly optimal, but rather only Boltzmann-rational ([Ziebart et al., 2008](https://www.aaai.org/Papers/AAAI/2008/AAAI08-227.pdf)). These works argue that this is especially true for many robotics applications where the offline dataset consists of human demonstrations (see [1-3] for papers that assume that human demonstrations are only noisily-optimal, either for activity forecasting or for performing inverse optimal control).
>
> [1] Huang et al., 2015, “Approximate MaxEnt Inverse Optimal Control and its Application for Mental Simulation of Human Interactions”. AAAI 2015.
>
> [2] Finn et al., 2016, “Guided Cost Learning: Deep Inverse Optimal Control via Policy Optimization”. ICML 2016.
>
> [3] Rhinehart et al., 2017, “First-Person Activity Forecasting with Online Inverse Reinforcement Learning”. ICCV 2017.
>
> Second, while not all datasets will be noisy-expert, we provide some examples of real-world problem domains where noisy-expert data can be easily collected:
>
> **Robotics:** in robotics noisy-expert data may not be directly available via demonstrations but can be obtained by a practitioner by running some limited autonomous data collection using noisy scripted policies (e.g., [QT-Opt](https://arxiv.org/abs/1806.10293) and [MT-Opt](https://arxiv.org/abs/2104.08212) run partly trained RL policies for noisy collection for running offline RL).
>
>
> **Recommender systems**: In recommender systems, policies are themselves often noisy and stochastic: see [Chen et al. 2018](https://arxiv.org/abs/1812.02353) which deployed noisy policies on Youtube, and found that serving either stochastic or deterministic policies in their recommender system led to similar ViewTime, This means that a logged dataset collected by such stochastic policies will likely be noisy as well, and can be used for offline RL.
>
>
> **Autonomous driving:** Offline datasets in autonomous driving can be augmented with negative samples and labeled with a ‘failure’ reward signal, which can allow us to take a near-expert or mixture of experts dataset and convert it to noisy before feeding it into the offline RL algorithm ([Bojarski et al. 2016](https://arxiv.org/abs/1604.07316)).
>
>
> ___
>
> > **state representation learning problem and the best action selection problem**
>
> We agree that decoupling the state-representation learning problem and the best action selection problem, and understanding how to perform state representation learning optimally is a very interesting open question. We believe that extending our analysis to incorporate state representation learning is important for future work, as our current analysis only considers tabular domains. However, since naively comparing existing suboptimality bounds in even tabular RL (e.g., Rashidinejad et al. 2021) would show that BC outperforms offline RL, in this work, we attempted to understand if there exist conditions under which offline RL would be better though BC is typically preferred. Since it was not clear to us if such conditions exist even in the tabular setting, as no prior works study this, we tackled the tabular setting as a first step. Though we ignore error due to representation learning via function approximation, we still study the impact of overestimation and distribution shift, which we believe is the core problem with offline RL algorithms.

---

> > ### Comment · Reviewer_9b7d · 2021-11-25
> > **Thanks for the response and revisions**
> >
> > While I am not changing my positive score, I believe the current revised version is much better with clear takeaways as a reader!

---

### Official Review · Reviewer_QvCh · 2021-11-02

**Correctness:** 4
**Technical Novelty And Significance:** 3
**Empirical Novelty And Significance:** 3
**Recommendation:** 6
**Confidence:** 4

**Main Review:**

### Strengths

1. The writing is clear throughout. The paper clearly explains its goals and is well organized.
2. The proofs seem to be generally correct.
3. The empirical analysis is relatively thorough and well documented. It seems to generally follow best practices with respect to tuning hyperparameters, reporting confidence intervals, and even reporting IQM on the Atari experiments.

### Weaknesses

1. I'm somewhat skeptical of the usefulness of the question the paper is trying to answer. First, from a practitioners perspective it is not clear why I have to choose between BC and offline RL. I can always run both and use some sort of validation procedure to decide which policy to deploy. Second, it seems to also be a false choice since offline RL algorithms often contain a hyperparameter that reduced the algorithm to BC if set to an extreme value, and in this sense BC is strictly contained in the offline RL algorithm. Third, the paper does not seem to consider the standard cases where BC is desirable and offline RL may not even apply. For example, BC is often used in problems where there is no obvious reward function (e.g. in many robotics problems), but we can get demonstrations of the desired behavior (e.g. from tele-operating). Finally, I am worried that the paper is arguing against a bit of a strawman. I am not aware of anyone making the argument that BC should be preferred to offline RL when a reward function and sufficient data are available.
2. The theory only considers tabular representations and as a result avoids the central issues of offline RL. By limiting to the tabular case, the analysis does not have to deal with the notorious deadly triad. Wang et al., 2021 and Zanette, 2020 (cited in the paper) prove exponential lower bounds on sample complexity in the case of linear function approximation. Importantly the results depend on errors due to the function approximation. Empirically, Brandfonbrener et al., 2021 (also cited in the paper) bears out some of these concerns caused by amplifying approximation errors in offline RL. By assuming away this key part of the problem, it is not surprising to see positive results for offline RL.
3. I am somewhat worried about novelty. On the theory side, the positive results largely exist in prior work and are even more general since guarantees exist beyond the tabular setting already. The comparison of the results across algorithms may be novel, but the results themselves do not seem to be a major contribution. On the empirical side, the sorts of results presented already exist even in the original paper that proposed the D4RL benchmark, showing that BC outperforms CQL sometimes and then in the paper proposing tuning methods for CQL demonstrating the improved performance from tuning. The emphasis on this comparison may be novel, but the results themselves do not seem to be.
4. I couldn't find any details at all about the BC-PI implementation used in the experiments. Could the authors comment on how this algorithm was implemented? This is especially interesting to me because it seems that a naive algorithm that attempted to filter out the bottom half of state-actions from the dataset would work very well on a dataset that is just half random and half expert data, so I'm wondering if there is something strange going on with the implementation.

**Summary Of The Paper:**

The paper considers a setting where we are given access to a dataset of expert or noisy-expert data collected from some MDP and need to decide whether to use either behavior cloning (BC) or offline RL. It conducts a theoretical analysis in a tabular setting showing that offline RL will recover a better policy than BC when the data is sufficiently suboptimal and has sufficient coverage. Finally it conducts an empirical analysis that confirms the results in some diagnostic gridworld tasks and shows that a tuned variant of the CQL offline RL algorithm outperforms BC in several larger-scale tasks.

**Summary Of The Review:**

While the paper is generally clear and technically sound, I found several serious issue with it and recommend rejection at this time. Specifically, I am skeptical of the usefulness of the central question, worried that the theory is assuming away the interesting part of the problem, worried about novelty, and had a few more minor concerns about the experimental setup.

---

> ### Author Response · Authors · 2021-11-12
> **Author Response (Part 1 of 3): Usefulness of the Question**
>
> Thank you for your constructive comments. We believe that the question our paper tackles is quite relevant to a significant segment of the community, and we reference several other recent works below that make various claims about the relative merits between RL and BC that suggest there is a lively debate to be had on this topic, which we believe our paper contributes to. We detail this point below along with our responses to your other questions below and have updated the paper (in red) to address concerns about the usefulness of the question (**Section 1**), detailed the key takeaways explicitly (**Practical Insights 4.1-4.4 in Section 4**), addressed novelty concerns (**Section 4**), and relationship with function approximation (**Appendix G**) and added experimental setup details in **Appendix F**. **We would appreciate it if you could tell us if these answers and the revised paper address your concerns.**
>
>
> ___
>
> > **Skeptical of the usefulness of the question the paper is trying to answer.**
>
> The key question we study is under what conditions will offline RL algorithms perform better than BC for a given task, even when BC is provided with expert data. A number of recent works claim that BC should be a viable substitute for offline RL in various cases (see, [Mandlekar et al. CoRL 2021](https://openreview.net/forum?id=JrsfBJtDFdI), [Florence et al. CoRL 2021](https://openreview.net/forum?id=rif3a5NAxU6), [Hahn et al. NeurIPS 2021](https://arxiv.org/abs/2110.09470), [Spencer et al. 2021](https://arxiv.org/pdf/2102.02872.pdf) for empirical studies), suggesting that in many cases practitioners should just use BC both with optimal and suboptimal data. We believe that it is of interest to the community to describe situations where the opposite is expected to be true. Our aim is to understand whether we should expect the conclusions of these prior works to be true in general, or if this may be an artifact of other factors, such as tuning or the problem structure. From a practitioners’ perspective, our analysis of this question indicates that trying offline RL should be a good idea, even in scenarios where expert or near-expert data is available and BC is considered to be a natural choice. We have updated **Section 1** to make this more clear.
>
> To study this key question, we first aimed to theoretically understand if there exist any conditions under which offline RL can outperform BC with expert data. After we were able to determine some conditions where offline RL should outperform BC even with optimal or near-optimal data, our goal in the empirical section was to verify our results in real-world tasks. In doing so, we also wanted to understand if, via careful offline tuning, offline RL methods can perform well. Overall, contrary to the recent papers cited above, our paper is able to theoretically and empirically identify realistic scenarios--which are representative of manipulation and navigation, where offline RL can outperform BC, even when the latter is given expert data. To make the key takeaways more clear, we have now presented the key takeaways from our analysis more prominently in **Practical Insights 4.1-4.4, in Section 4**, including implications of these takeaways for a practitioner.
>
> ___
>
> > **BC vs offline RL is a false choice. I can always run both and use some sort of validation procedure to decide which policy to deploy; There are problems where BC may not apply and offline RL may be more natural and vice versa.**
>
> We of course agree that there is no downside to trying both offline RL and BC, and taking the best one (though doing so with offline validation may be hard, as this is an open problem). However, we believe it is still important for the community to understand the circumstances where we might expect benefit from offline RL because in most cases, practitioners likely wouldn't even think to try offline RL when presented with near-optimal or optimal expert data, because this is the classic setting for using BC, and there is no prior work that indicates that offline RL should **ever** be preferred under these conditions. By showing (both theoretically and empirically) that offline RL may be preferred in some such settings, we believe our paper makes a valuable contribution. Indeed, this is also why we believe our analysis is useful even without function approximation -- the aim is to show there exist settings that would typically be addressed with BC where offline RL is preferred, not that it is **always** preferred.

---

> > ### Author Response · Authors · 2021-11-12
> > **Author Response (Part 2 of 3): Function Approximation and Lower Bounds**
> >
> > > **Theory avoids central issues (function approximation, bootstrapping) with offline RL. Throwing away the part of the problem, it is not surprising to see positive results for offline RL.**
> >
> > We agree that function approximation and bootstrapping amplify the problems in training offline RL algorithms, and extending our analysis to function approximation is important future work. However, despite the good performance of deep RL in practice, showing convergence and good performance in deep RL under function approximation, in general, is an open problem (e.g., [Du et al. 2020](https://arxiv.org/abs/1910.03016)). On the other hand, offline RL is known to suffer from overestimation and distributional shift even without function approximation (see [Laroche et al. 2019](http://proceedings.mlr.press/v97/laroche19a/laroche19a.pdf), [Kumar et al. 2019](https://arxiv.org/abs/1906.00949),  [Kidambi et al. 2020](https://arxiv.org/abs/2005.05951), [Rashidinejad et al. 2021](https://arxiv.org/abs/2103.12021)) and several recent papers in offline RL study theoretical guarantees for offline RL in tabular settings (see **references [1-4]** below for papers accepted recently at **NeurIPS 2021** which primarily analyze tabular offline RL). It is not at all clear that offline RL should outperform BC despite the overestimation issue due to distribution shift. In fact, even offline bandit and model-based optimization problems (e.g., [Kumar et al. 2020](https://proceedings.neurips.cc/paper/2020/hash/373e4c5d8edfa8b74fd4b6791d0cf6dc-Abstract.html), [Trabucco et al. 2021](https://arxiv.org/abs/2107.06882)) suffer from overestimation issues arising due to distribution shift even though no bootstrapping is present. Thus, we believe, comparing offline RL and BC along the distribution shift aspect is still valuable, without function approximation.
> >
> > In fact, naively comparing upper bounds in previous theoretical works on offline RL (for example, comparing Theorem 6 in Rashidinejad et al. 2021 and Theorem 4.2 in Rajaraman et al. 2021)  would, in fact, imply that BC would outperform offline RL. Thus, it is an open problem whether offline RL can outperform BC at all in settings where BC is applicable (i.e., expert or near-expert data), whether with or without function approximation. Certainly analyzing this question with function approximation is an interesting avenue for future work, but due to the above reasons, we believe that studying this question theoretically in the tabular setting is a natural and non-trivial first step, and our empirical results (with function approximation) lend credence to the possibility that the conclusion may still hold.
> >
> >
> > [1] Rashidinejad et al. 2021, “Bridging Offline Reinforcement Learning and Imitation Learning: A Tale of Pessimism”. NeurIPS 2021
> >
> > [2] Xie et al. 2021, “Policy Finetuning: Bridging Sample-Efficient Offline and Online Reinforcement Learning”. NeurIPS 2021.
> >
> > [3] Yin et al. 2021. “Near-Optimal Offline Reinforcement Learning via Double Variance Reduction”. NeurIPS 2021.
> >
> > [4] Yin et al. 2021. “Optimal Uniform OPE and Model-based Offline Reinforcement Learning in Time-Homogeneous, Reward-Free, and Task-Agnostic Settings”. NeurIPS 2021.
> >
> > ___
> >
> > > **Lower bounds: Wang et al. 2020, Zanette 2020**
> >
> > We would like to note that the lower bound in Wang et al. 2020 only applies to non-conservative algorithms. In fact, Section 5, Theorem 5.1 in Wang et al. 2020 provides an upper bound for offline policy evaluation with low distribution shift. The algorithms analyzed in our paper are conservative and attain prevent the learned policy from going out-of-distribution, and so the lower bound in Wang et al. 2020 does not directly apply.
> >
> > Zanette 2020's worst-case analysis does apply to very general function approximation scenarios, even under realizability and closure. However, this analysis concerns policy evaluation and applies to policies that go out of the support of the offline data. This does not necessarily prove that any conservative offline RL algorithm (i.e., one that prevents the learned policy from going out of the support of the offline dataset) would suffer from this issue during policy learning, just that there exist policies for which evaluation would be exponentially bad. Offline RL can exactly prevent the policy from going out of the support of the dataset in Zanette’s counterexample since the dynamics and reward functions are deterministic. This policy can still improve over behavior cloning by stitching overlapping trajectories, similar to the discussion of one-step vs multi-step PI in Theorem 4.4. Thus, while the lower bound applies to off-policy evaluation of ***some*** target policies, it doesn't apply to off-policy evaluation of ***every*** policy in the MDP and specifically does not apply to in-support policies, which modern offline RL algorithms produce, in theory, and practice.
> >
> > We have added this discussion to **Appendix G** in the paper.

---

> > > ### Author Response · Authors · 2021-11-12
> > > **Author Response (Part 3 of 3): Novelty concerns, experiment details**
> > >
> > > > **Novelty concerns**
> > >
> > > **Theory:** We have now updated the paper to clearly highlight the novel contributions in theory and practice (changes in red in Section 4). While the algorithms we analyze exist in prior work, our work does not simply compare bounds from these prior works. We highlight the novel components of the main theoretical results below:
> > >
> > > - **Theorem 4.2:** Our analysis in Theorem 4.2 improves over the existing analysis in Rashidinejad et al. 2021 by using a Bernstein bonus and the recursion technique to obtain an $\mathcal{O}(\sqrt{H})$ bound compared to $\mathcal{O}(H^{2.5})$, and avoids an additional factor of $|\mathcal{S}|$ by utilizing the s-absorbing state technique. These improvements are key in allowing us to show that offline RL can perform better than BC; the bounds in Rashidinejad et al. 2021 directly do not show this.
> > >
> > > - **Corollary  4.2:** We compare the performance of BC with expert/near-expert data to RL with noisy data and show that for an equal amount of data, RL can do better. Such a comparison is not present in prior works, and is perhaps also surprising, and practically informative for a practitioner, especially when they are deciding what data to collect to attain the best performance on their problem. Just comparing the upper bounds from BC and offline RL from prior works naively would in fact indicate the opposite, meaning that our analysis is non-trivial. We have added the practical takeaway for this in Practical Insight 4.3.
> > >
> > > - **Theorem 4.4:** We utilize tools from prior work to prove this theorem, but we characterize the performance gain obtained by considering multiple rounds of policy improvement compared to one round of policy improvement. We are unaware of any prior work showing this benefit in the context of offline RL, and in fact, recent work has shown conflicting opinions (see discussion in [Kostrikov et al. 2021](https://arxiv.org/abs/2110.06169), Section 5.1 comparing to Brandfonbrener et al. 2021).
> > >
> > > In addition, to our knowledge, the analysis of algorithms under the practically-relevant condition of critical states in Corollary 4.1 is novel.
> > >
> > > **Empirical Results:** We have now updated the paper to clearly highlight the novel takeaways (Practical Insights 4.1-4.4) and related them to our experiments. While the specific algorithms used for offline learning and offline tuning are not novel (since we are not trying to propose a new algorithm), to our knowledge, no prior work shows that running some kind of offline RL (either on noisy data or expert data) can outperform cloning an expert dataset for the same task, given equal amounts of data. In fact, results in D4RL would indicate the opposite (compare BC with expert data vs the best offline RL algorithm for medium-expert data in Table 2 in the D4RL paper). The fact that offline tuning procedures applied to offline RL algorithms on noisy-expert datasets allow us to outperform BC is a conclusion that, to the best of our knowledge, has not been demonstrated before.
> > >
> > > ___
> > >
> > > > **BC-PI Details**
> > >
> > > We have now updated the paper to include details about the BC-PI implementation used in our experiments in **Appendix F** in red. We utilize SARSA to train an estimate of the Q-function of the behavior policy and then extract a policy via a weighted regression objective. We sweep over multiple values of the hyperparameter $\eta$ (the temperature hyperparameter during policy extraction).
> > >
> > > Note that our experiments do not use half random and half expert data, but rather expert data and very few random exploration trajectories generated via epsilon-greedy exploration from early buffers of DQN-Replay (Agarwal et a. 2020) dataset. We suspect that the performance of this algorithm is worse than BC because the task is stochastic dynamics (Atari games) and utilizing exponentiated weights for policy extraction reduces the sample size for training the policy. We are happy to tune any hyperparameters or reimplement a different version of BC-PI if the reviewer has any suggestions.

---

> > > > ### Comment · Reviewer_QvCh · 2021-11-17
> > > > **reviewer response**
> > > >
> > > > Thanks for the detailed response. I think the response and updates resolve some of the issue I raised (specifically on the theory side), but I still have concerns (mostly on the empirical/practical side). I will raise my score to a 5 to reflect my higher opinion of the theoretical contribution.
> > > >
> > > > **Usefulness of the central question**: I think this is still an issue. The authors seem to agree that the most practical approach is to run both algorithms and choose the better one by validation. This is *especially* true given the emphasis the authors place on *tuning* the offline RL algorithms. Since BC can be seen as a special case of the offline RL algorithm (where the hyperparameter that controls how much the policy can deviate from the behavior is set to an extreme value), the tuning procedure is essentially comparing against BC and automating the decision of how much to deviate from BC. I agree that there is still some value in understanding when exactly we might expect offline RL with weaker regularization/constraints to outperform BC, but the claims about the practical usefulness of the contribution of this paper are overstated.
> > > >
> > > > **Theory**: Thanks for providing the references on tabular offline RL. While I still think that the function approximation/extrapolation issues are presenting a more fundamental issue for offline RL and the positive results in the tabular setting are not surprising, I agree that this is still a good starting point at least. And thanks for clarifying more directly the improvement of the analysis over prior work. I think this theoretical contribution is the strong point of the paper now.
> > > >
> > > > **Empirical results/novelty**: I disagree that this is the first paper showing good empirical performance of offline RL on expert data. Take for example the D4RL paper. In table 2, all the adroit tasks consist of demonstration or expert data where BC would be applicable, however BC only achieves the best performance on 2 out of 12 datasets (pen-cloned and relocate-expert). Often the gap between BC and offline RL is large, e.g on pen-expert or door-expert. And these D4RL baselines are even without the tuning procedure emphasized in this paper. But I agree that this idea of using offline RL on expert data was not emphasized or explored in much detail by prior work.
> > > >
> > > > **BC-PI details**: The added appendix raises more issues about this implementation and does not seem to be sufficient. First, the details in the appendix suggest that exponentiated Q values were used, while prior work uses exponentiated *advantage* estimates [e.g. Brandfonbrener et al. 2021]. This should make a substantial difference since the exponential will upweight positive values (advantages) and downweight negative values (advantages). Second, for numerical stability, prior work has clipped the weights, but this doesn't seem to have been done in this implementation and could also cause serious problems.  Finally, none of the hyperparameters are reported on the google site that reports hyperparameters for the other algorithms in the paper. Also, as a point of reference Gulcehre et al [2021] find that a similar BC-PI algorithm works quite well on Atari, so I am not sure that I buy the idea that this algorithm won't work on Atari.
> > > >
> > > > Gulcehre, C., Colmenarejo, S. G., Wang, Z., Sygnowski, J., Paine, T., Zolna, K., ... & de Freitas, N. (2021). Regularized behavior value estimation. *arXiv preprint arXiv:2103.09575*.
> > > >
> > > > **Final comment on story/tone of paper**: Looking back at the paper again, I think that the way the story is told is still one-sided and this is part of what is causing issues for me. While I understand that the paper is trying to differentiate itself from prior work that emphasizes the strength of BC by instead emphasizing the weakness of BC, I think it could benefit from a more even hand (i.e. by also presenting scenarios where BC outperforms offline RL).

---

> > > > > ### Author Response · Authors · 2021-11-18
> > > > > **Further Discussion (Part 1 of 2): Concerns about experimental results**
> > > > >
> > > > > Thanks for responding to our responses. We are glad that the reviewer thinks the paper is strong on theoretical contributions. We would like to answer the remaining questions below -- in summary, we are rerunning BC-PI per the reviewer’s suggestions, and are happy to change the title of the paper to reflect the tone/story we study and revise claims appropriately. We also have some requests for clarification and it would be great if the reviewer can provide these clarifications.
> > > > >
> > > > > **BC-PI details:** We are now running the method that incorporates the reviewer’s suggestion of utilizing advantages and clipping, and we will tune it over multiple values of the temperature hyperparameter for the advantage weights and the clipping threshold. We will update the reviewer and the paper with the status of the results in a few days.
> > > > >
> > > > > We believe there are key differences in the setting of Gulcehre et al. and what we study. First, note that the dataset composition we study (noisy-expert) is very different from the one used in Gulcehre et al. (full replay buffer data, which is not suitable for BC). Second, note that Figure 5 of Gulcehre et al. shows that one-step of policy improvement using a SARSA Q-function (BC-PI in our paper and BVE in Gulcehre et al.) performed worse than a naive Double DQN. To obtain good performance they apply ranking regularization on the Q-function (R-BVE), which makes the Q-function conservative by pushing down OOD Q-values, in a similar fashion as CQL. So, their algorithm does not exactly fit what we call BC-PI, but if the reviewer suggests, we can implement and evaluate it.
> > > > >
> > > > > ___
> > > > >
> > > > > **Empirical results/novelty:** We have revised our claims in the introduction (Section 1) and discussed the fact that results in D4RL may indicate that offline RL outperforms BC, although D4RL does not discuss this in depth (Section 2). That said, some papers have shown the opposite, that tuning BC on the D4RL adroit tasks can make it better compared to D4RL results. For example, observe the Explicit BC and Implicit BC numbers in Table 2 of [Florence et al. 2021](https://openreview.net/forum?id=rif3a5NAxU6) are higher than those in D4RL, indicating that D4RL Adroit results may not be fully-tuned. Thus it seems fair to say that it’s an open question how BC and RL compare on such tasks, and that there is disagreement among prior works.
> > > > >
> > > > > One of the differences we found was that Gaussian policies overfit much less on Adroit compared to tanh-Gaussian policies and hence perform better. [D4RL evaluation code]([https://github.com/rail-berkeley/d4rl_evaluations](https://github.com/rail-berkeley/d4rl_evaluations)) uses tanh-Gaussian policies for BC, which may be lead to suboptimal results. We think that comparing offline RL and BC in a well-tuned manner is more informative.

---

> > > > > > ### Author Response · Authors · 2021-11-18
> > > > > > **Further Discussion (Part 2 of 2): Concerns about tone and practical usefulness**
> > > > > >
> > > > > > **Final comment on tone/story:** We agree with the reviewer that we offer insights on one side of the offline RL vs BC debate, namely when offline RL outperforms BC and we have updated the discussion to acknowledge this limitation. We do this simply because some recent theoretical and empirical results, such as [Rashidinejad et al. 2021](https://arxiv.org/abs/2103.12021) and [Mandlekar et al. 2021](https://openreview.net/forum?id=JrsfBJtDFdI)  focus on the other side. We are happy to change the title of the paper to “When does Offline RL Outperform BC on Expert Data?” or “A Study of When Offline RL Outperforms BC”, to more adequately reflect the content and the story of the paper. In the revised version of the paper, our abstract and introduction are already focused towards answering the questions listed above and we have updated the discussion section to acknowledge that we address only one-side of the story. **Please let us know if this addresses the concern.**
> > > > > >
> > > > > > ___
> > > > > >
> > > > > > **Usefulness of the central question:** We agree with the reviewer that offline RL vs BC could be cast as a tuning problem. However, we think that our work still has considerable value for two reasons. First, this tuning problem is not solved, and evidently it is not very easy. For example, as discussed in [DOPE](https://openreview.net/forum?id=kWSeGEeHvF8), general off-policy evaluation is very difficult even with the best recent algorithms, and while there is some work on various offline tuning strategies (such as Kumar et al. CoRL 2021), there is no single and broadly applicable approach that works across the board. Second, and perhaps more importantly, the hypothesis put forward in several recent works ([Mandlekar et al. CoRL 2021](https://openreview.net/forum?id=JrsfBJtDFdI), [Florence et al. CoRL 2021](https://openreview.net/forum?id=rif3a5NAxU6), [Hahn et al. NeurIPS 2021](https://arxiv.org/abs/2110.09470))  is that RL methods may simply be unnecessary in the case where good offline data is available -- that is, we should not even **attempt** to tune how much the method performs RL, and just go with the (simpler, quicker) BC solution. Our work offers a counterpoint to this perspective, which we believe is important to this segment of the community.
> > > > > >
> > > > > > ___
> > > > > >
> > > > > > **Regarding claims about practical usefulness,** we are happy to revise any practical claims that the reviewer thinks are overstated. We have uploaded a revision in which we have removed the claim that we are first to show that offline RL can outperform BC empirically. **If the reviewer has specific claims in mind, we are happy to revise them if the reviewer can specify them.**

---

> > > > > > > ### Author Response · Authors · 2021-11-23
> > > > > > > **Author follow-up: changed title, updated BC-PI**
> > > > > > >
> > > > > > > Dear Reviewer QvCh,
> > > > > > >
> > > > > > > We wanted to give you an update on some of the aspects from our previous response to your comment:
> > > > > > >
> > > > > > > **BC-PI details:** During the rebuttal, we spent quite a bit of additional time tuning the BC-PI baseline. We believe that we've tuned to the best of our ability now, and incorporated your suggestions of utilizing exponentiated advantages and clipped advantage weights. We have updated the results in Figure 2 of the paper. The results for BC-PI are still somewhat comparable to BC, and we discuss possible reasons for this in **Appendix F, Page 32** along with our hyperparameters and observations for this method.  There could be many reasons behind this performance such as the need for a different temperature hyperparameter for advantage-weighted policy extraction for each game (which is not allowed in the standard Atari evaluation protocol typically followed for Atari evaluations since the original DQN paper ([Mnih et al. 2013](https://arxiv.org/pdf/1312.5602.pdf)), and which we also follow for all our other baselines), or even different temperature hyperparameter values across different states within the MDP.
> > > > > > >
> > > > > > > However, we think that our current results reflect the performance of this method as favorably as possible. We would be happy to compare to any other existing open-source code if you believe there is a better implementation to use or some implementation trick that we missed, but we believe we've been as complete as possible in making this baseline maximally strong. The details and the hyperparameters for this method are shown in **Appendix F, Page 32** (paragraph titled “Tuning BC-PI”, in $\textcolor{magenta}{magenta}$).
> > > > > > >
> > > > > > > **Title of the paper:** We have updated the title to “When Should Offline Reinforcement Learning be Preferred Over Behavioral Cloning?” to better reflect the contributions and the tone of the paper, and are happy to revise it further based on your suggestions.
> > > > > > >
> > > > > > > ____
> > > > > > >
> > > > > > > *Please let us know if our previous response and the answers in this response address your questions.* We are happy to answer any remaining questions.
> > > > > > >
> > > > > > > Thanks so much!

---

> > > > > > > > ### Comment · Reviewer_QvCh · 2021-11-26
> > > > > > > > **revising score after final revision**
> > > > > > > >
> > > > > > > > Thanks for the very thorough response to my comments and corresponding improvements to the paper. As a result of the comments I will raise my final score above the acceptance threshold to a 6.
> > > > > > > >
> > > > > > > > In the end, while I am still somewhat suspect of the central question, I do see that it could be useful for a subset of the RL community. Moreover, I think that the theoretical contributions are interesting and correct and the empirical evaluation is thorough while not novel. The clarity and tone of the revised version are also much improved from the original.

---

### Author Response · Authors · 2021-11-23
**Summary of Changes**

We would like to thank all the reviewers for their detailed and constructive feedback. The feedback was very useful and we have revised the paper to answer reviewers’ questions and incorporated reviewer suggestions. The updates are highlighted in red/magenta. A few major updated we have made to the paper are as follows:

1. **Changed the title of the paper**: Per the suggestions of the reviewers we have changed the title of our paper to *“When Should Offline Reinforcement Learning be Preferred over Behavioral Cloning?”*. We also believe that this better reflects the contributions of the paper. We are also happy to revise it further if the reviewers have any suggestions. We have also updated the Introduction section to better present the motivation behind this work.

2. **Added practical takeaways:** We now present the key takeaways from our analysis more clearly in Practical Insights 4.1-4.4.

3. **Improved motivation and added intuitions for conditions:** In the main paper, we provide some improved discussion on how the conditions we study arise in realistic scenarios to better motivate the practicality of our conditions. We elaborate on this discussion in Appendix D and provide various intuitive explanations of the conditions and discuss practical examples.

We have also updated various aspects in the main paper, to add references pointed out by reviewers, clarified the connection of our empirical conclusions to the results in the D4RL paper (Fu et al. 2020), incorporated changes to certain claims, updated the BC-PI results based on reviewer suggestions (**Reviewer QvCh**).

---

### Decision · Program_Chairs · 2022-01-20

**Decision:**

Accept (Poster)

**Comment:**

This paper considers helping to decide whether behavior cloning or offline RL is likely to be more effective given a particular offline dataset. The reviewers initially appreciated the importance of insights into this question around how to best leverage an existing dataset. They also had some initial concerns, due in part because the theory is restricted to tabular settings, whereas many challenges typically arise when function approximators are used, the realisticness of the assumptions over the data collection process, and a number of places where further details or clarifications would better situate and strengthen the work. The authors gave very extensive responses to the feedback which made reviewers feel much more confident about the revised paper and resulted in significantly higher scores. Though there remains many interesting areas for future work, this paper makes an interesting contribution that may be of interest to many using batch decision making data.